# Evidence for the stabilization of FeN₄ sites by Pt particles during acidic oxygen reduction

Nicolas A. Ishiki [1,2,10,13], Keyla Teixeira Santos [1,11,13], Nicolas Bibent [3], Kavita Kumar[1], Ina Reichmann [4,5], Yu-Ping Ku [4,5], Tristan Asset [3,6], Laetitia Dubau [1], Michel Mermoux[1], Hongxin Ge[7], Sandrine Berthon-Fabry[7], Viktoriia A. Saveleva[8,12], Vinod K. Paidi [8], Pieter Glatzel [8], Andrea Zitolo [9], Tzonka Mineva [3], Hazar Guesmi[3], Serhiy Cherevko [4], Edson A. Ticianelli[2], Frédéric Maillard [1] ✉ & Frédéric Jaouen [3] ✉

While Fe–N–C materials have shown promising initial oxygen reduction reaction (ORR) activity, they lack durability in acidic medium. Key degradation mechanisms include FeN₄ site demetallation and deactivation by reactive oxygen species. Here we show for mainstream Fe–N–Cs that adding 1 wt.% Pt nanoparticles via a soft polyol method results in well-defined and stable Pt/Fe–N–C hybrids. The Pt addition strongly reduces the H₂O₂ production and Fe leaching rate during ORR, while *post mortem* Mössbauer spectroscopy reveals that the highly active but unstable Fe(III)N₄ site is partially stabilized. The similar H₂O₂ electroreduction activity of Pt/Fe–N–C and Fe–N–C and other analyses point toward a long-distance electronic effect of Pt nanoparticles in stabilizing FeN₄ sites. Computational chemistry reveals that spin polarization of distant Pt atoms mitigates the structural changes of FeN₄ sites upon adsorption of oxygenated species atop Fe, especially in high-spin state.

The transition to a sustainable energy system is paramount for the coming decades. Hydrogen (H₂) is a versatile energy carrier, yet its conversion to electric power in proton-exchange-membrane fuel cells (PEMFCs) heavily relies on Pt for catalyzing the H₂ oxidation reaction and the oxygen (O₂) reduction reaction (ORR)[1,2]. Fe–N–C catalysts are promising substitutes for scarce and expensive Pt towards the ORR, but they lack long-term durability[3–6]. Research is ongoing to unravel their degradation mechanisms[7–10]. As suggested by Lefèvre and Dodelet two decades ago, hydrogen peroxide (H₂O₂) can significantly impact durability by reacting with Fe cations to form reactive oxygen species (ROS)[7,11,12]. These species oxidize the nitrogen-(N-) doped carbon matrix, decreasing the turnover frequency (TOF) of FeN₄ active sites, ultimately facilitating Fe leaching[10,13,14]. Demetallation of FeN₄ sites has been observed under inert gas but more rapidly under O₂ conditions, with subsequent precipitation as Fe oxides (FeOₓ) under oxidizing atmosphere and high temperature[7,12,15,16]. The high-spin

¹Univ. Grenoble Alpes, Univ. Savoie Mont Blanc, CNRS, Grenoble INP, LEPMI, 38000 Grenoble, France. ²Instituto de Química de São Carlos (IQSC), Universidade de São Paulo, Av. Trabalhador São-Carlense, 400, CP 780, São Carlos, SP, Brazil. ³ICGM, Univ. Montpellier, CNRS, ENSCM, 1919 route de Mende, 34293 Montpellier, France. ⁴Forschungszentrum Jülich GmbH, Helmholtz-Institute Erlangen-Nürnberg for Renewable Energy (IET-2), Cauerstraße 1, 91058 Erlangen, Germany. ⁵Friedrich-Alexander-Universität Erlangen-Nürnberg, Department of Chemical and Biological Engineering, Cauerstraße 1, 91058 Erlangen, Germany. ⁶Institut de Chimie et Procédés pour l'Energie, l'Environnement et la Santé, UMR 7515 CNRS-University of Strasbourg, 25 rue Becquerel, 67087 Strasbourg Cedex, France. ⁷MINES Paris, PSL University PERSEE - Centre procédés, énergies renouvelables et systèmes énergétiques, CS 10207 rue Claude Daunesse, 06904 Sophia Antipolis Cedex, France. ⁸ESRF, The European Synchrotron, 71 Avenue des Martyrs, 38000 Grenoble, France. ⁹Synchrotron SOLEIL, L'Orme des Merisiers, BP 48 Saint Aubin, 91192 Gif-sur-Yvette, France. ¹⁰Present address: Université Paris Cité, CNRS, ITODYS, 75013 Paris, France. ¹¹Present address: Instituto de Química, Universidade Estadual de Campinas (UNICAMP), R. Monteiro Lobato, 270, 13083-970 Campinas, SP, Brazil. ¹²Present address: Li2, joint laboratory of Blue Solutions, LEPMI and Grenoble INP, 1025 rue de la Piscine, 38610 Gières, France. ¹³These authors contributed equally: Nicolas A. Ishiki, Keyla Teixeira Santos. ✉e-mail: frederic.maillard@grenoble-inp.fr; frederic.jaouen@umontpellier.fr

Fe(III)$N_4$ site (identified as D1, by experimental and computational [57]Fe Mössbauer spectroscopy) is the most prone to demetallation, unlike the stable low- or medium-spin Fe(II)$N_4$ sites (labeled D2)[17,18].

Attempts to enhance the durability of Fe−N−C materials often involve interfacing them with Pt nanoparticles (NPs) due to the known activity of Pt towards $H_2O_2$ electroreduction[19,20]. Since Mechler et al. proposed a ROS-scavenger mechanism using low Pt content (≤ 2 wt. %)[21], various groups have explored this synergy[22–27]. However, different Pt and Fe active sites emerge in the current process for preparing such hybrids due to a high-temperature treatment used to reduce the Pt salt, leading to the formation of Pt-based alloys, Pt@$FeO_x$ core-shell structures, and single-metal atoms (SMAs) of Pt[22,24,27]. Recently, Bae et al. achieved full dispersion of 0.5 wt. % Pt as SMAs by annealing Pt salt onto Fe−N−C matrix at 250 °C[16]. They observed a reduction in Fe dissolution rates, attributed to an increased energy barrier for Fe leaching due to Pt$N_4$ sites adjacent to Fe$N_4$ sites. Conversely, their Pt/Fe−N−C catalyst did not show significantly decreased $H_2O_2$ production during ORR compared to the parent Fe−N−C, and a drop in current density was still observed during PEMFC operation. Gridin et al. synthesized catalysts with *ca.* 1 wt. % of metal NPs (Pd, Ag, Ir, Au) on Fe−N−C using polyol[28]. However, moderately improved durability was observed in PEMFC for the hybrids compared to Fe−N−C. They also prepared a Pt/Fe−N−C with *ca.* 2 wt. % Pt, but saw a 20% decline in current density after 12 h of operation. The synthesis involved physically mixing commercial 20 wt.% Pt/C with Fe−N−C, resulting in limited interaction between Fe$N_4$ SMA sites and Pt NPs. The role of Fe−N−C as a support for high-loadings of Pt NPs was also recently investigated. Xiao et al. synthesized a 30 wt.% Pt/Fe−N−C hybrid using formaldehyde to reduce the Pt salt[29]. They observed lower Pt dissolution from Pt/Fe−N−C *vs.* Pt/C, and improved electrochemical stability. Similar results were reported by others[24,30–32]. The overall ORR activity of these hybrids is however mainly attributed to Pt due to high Pt loadings.

In summary, the fundamental effect of Pt NPs in stabilizing Fe$N_4$ sites remains inconclusive. Even more important, unequivocal demonstration that Pt NPs indeed structurally stabilize Fe$N_4$ sites during ORR in acid medium has not yet been demonstrated. Here, we functionalized three different Fe−N−C catalysts with a low amount of Pt NPs (*ca.* 1 wt. %) via a soft polyol method, which does not modify the initial Fe speciation and leads to a uniform dispersion of Pt NPs. Our Pt/Fe−N−C hybrid models demonstrated enhanced electrochemical durability in rotating disk electrode (RDE) under oxygenated conditions and in PEMFC. Almost zero $H_2O_2$ production was detected in liquid electrolyte. A strongly reduced Fe dissolution rate was monitored online with a gas diffusion electrode (GDE) setup, while partial stabilization of the short-lived high-spin Fe(III)$N_4$ site was observed from [57]Fe Mössbauer spectroscopy after PEMFC operation. Based on the low ratio of Pt NPs to Fe$N_4$ sites, we propose that a long-range effect is at play in the stabilization mechanism. Computational chemistry reveals how distant Pt atoms can improve the structural stability of Fe$N_4$ sites.

## Results and discussion
### Structural and electrochemical properties of (Pt/)Fe−N−$C_{Aero}$

The aerogel-derived Fe−N−C material (Fe−N−$C_{Aero}$) was characterized in our previous works[33,34]. Transmission electron microscopy (TEM) micrograph in Fig. 1a shows its morphology. The catalyst features an Fe content of 1.25 wt. %, and a nitrogen sorption isotherm of type I at low $P/P_0$ and type II at higher $P/P_0$ (Supplementary Fig. 1a). It has a high specific surface area of 1191 $m^2$ $g^{-1}_{powder}$, a total volume of 0.51 $cm^3 g^{-1}_{powder}$ and a microporous volume of 0.46 $cm^3 g^{-1}_{powder}$ (Supplementary Table 1)[34]. [57]Fe Mössbauer spectroscopy at −268 °C confirmed the absence of Fe oxides and identified D1 and D2 doublets, with predominance of the former, and a low amount of $Fe_3C$ (Supplementary Fig. 2a and Supplementary Table 2). The D1 and D2 components are assigned to high-spin Fe(III)$N_4$ and low- or medium-spin Fe(II)$N_4$ configurations, respectively[17]. The Fe-N bond distance and Fe-N average

coordination number in Fe−N−$C_{Aero}$ were assessed from its extended X-ray absorption fine structure (EXAFS) spectrum measured at the Fe $K$-edge (Supplementary Fig. 3a and Supplementary Table 3). The experimental EXAFS spectrum was satisfactorily fitted considering a number of N and O atoms in the first coordination sphere and C atoms in the second coordination sphere while no Fe-Pt interaction was necessary to interpret the spectrum. An average Fe-N coordination number of 3.8 ± 0.3 was found with a bond distance of 2.05 ± 0.02 Å (Supplementary Table 3), confirming the Fe$N_4$ site structure. The average coordination number for Fe-O was 1.8 ± 0.2, suggesting that most of the Fe$N_4$ sites in Fe−N−$C_{Aero}$ are gas-phase accessible and adsorb either two $O_2$ molecules in end-on mode or one $O_2$ molecule in side-on mode, as previously proposed[35].

The Pt/Fe−N−$C_{Aero}$ hybrid catalyst, synthesized via a modified polyol method and then deposited onto Fe−N−$C_{Aero}$ (see Methods section), achieved homogenous Pt distribution (Fig. 1b). Scanning transmission electron microscopy measurements coupled with energy dispersive X-ray spectroscopy (STEM-X-EDS) confirmed the high dispersion of Pt NPs on Fe−N−$C_{Aero}$ (Supplementary Fig. 4). We also prepared an Fe-free aerogel using the same synthesis but without the Fe precursor, labeled N−$C_{Aero}$ and modified it similarly with Pt NPs, labeled as Pt/N−$C_{Aero}$. Pt particle size ranged from 1.0–2.5 nm in both Pt/Fe−N−$C_{Aero}$ and Pt/N−$C_{Aero}$ (Supplementary Fig. 5a–d). Inductively coupled plasma mass spectrometry (ICP-MS) measured Pt contents of 1.02 and 0.78 wt. % in Pt/N−$C_{Aero}$ and Pt/Fe−N−$C_{Aero}$, respectively, with Fe content in Pt/Fe−N−$C_{Aero}$ at 0.68 wt. %.

X-ray diffraction (XRD) measurements were conducted on N−$C_{Aero}$, Fe−N−$C_{Aero}$, and their platinized versions (Supplementary Fig. 6a). Besides graphite reflections, a minor peak at 2θ = 39.9° assigned to Pt(111) was identified in the platinized samples' diffractograms, confirming Pt NP deposition. XRD analysis also provided information on the structure of the N-doped carbon matrix. Both the interlayer spacing ($d_{002}$) and the carbon crystallite size perpendicular to the graphene layers ($Lc$) were similar for all four materials (Fig. 1c). Consistently, the $A_{D2}/A_G$ ratio (Fig. 1c) derived from Raman measurements (Supplementary Fig. 6b, c) was similar across all materials.

X-ray photoelectron spectroscopy (XPS) was conducted on N−$C_{Aero}$, Fe−N−$C_{Aero}$, Pt/N−$C_{Aero}$, Pt/Fe−N−$C_{Aero}$, and a commercial Pt/C catalyst (Supplementary Fig. 7a). Detailed analyses of the C 1$s$, N 1$s$ and Pt 4$f$ regions enabled the speciation of C, N and Pt sub-species (Supplementary Fig. 7b–d). No significant trend was observed in the relative atomic percentages of these sub-species among the materials (Supplementary Fig. 7e–g and Supplementary Tables 4–6), highlighting the effectiveness of our method in depositing Pt NPs without structural or chemical modification of the (Fe)−N−C substrates. To explore electronic interactions between Pt NPs and Fe$N_4$ sites, Fig. 1d compares the Pt 4$f$ fitting for commercial Pt/C, Pt/N−$C_{Aero}$ and Pt/Fe−N−$C_{Aero}$. A considerable shift of *ca.* 0.8 eV towards a higher binding energy was found for Pt/N−$C_{Aero}$ and Pt/Fe−N−$C_{Aero}$ compared to Pt/C. This observation aligns with the known change in the electronic density of carbon matrices when doping them with nitrogen, which results in a stronger metal-support interaction with the Pt NPs. This effect explains the shift in the binding energy values obtained and has been observed regardless of the presence, or not, of Fe$N_4$ sites in the N-doped carbon matrix[30,36].

Fe speciation and electronic states were investigated using Fe $K\alpha_1$-detected high-energy-resolution fluorescence-detected X-ray absorption near edge structure (HERFD-XANES), Fe $K\beta$ X-ray emission spectroscopy (XES) and Fe $K$-edge EXAFS spectroscopy. Both Fe−N−$C_{Aero}$ and Pt/Fe−N−$C_{Aero}$ exhibited spectra characteristic of SMAs Fe$N_4$ sites in XANES, XES and EXAFS (Fig. 1e, Supplementary Fig. 8 and Supplementary Fig. 3b, respectively)[9,17,35]. A slightly lower intensity of the white line in the XANES spectrum of Pt/Fe−N−$C_{Aero}$ hints at a slightly reduced average oxidation state of Fe[29,37]. Pt/Fe−N−$C_{Aero}$ also showed a shift towards lower energy in Fe XANES spectra pre-edge and rising

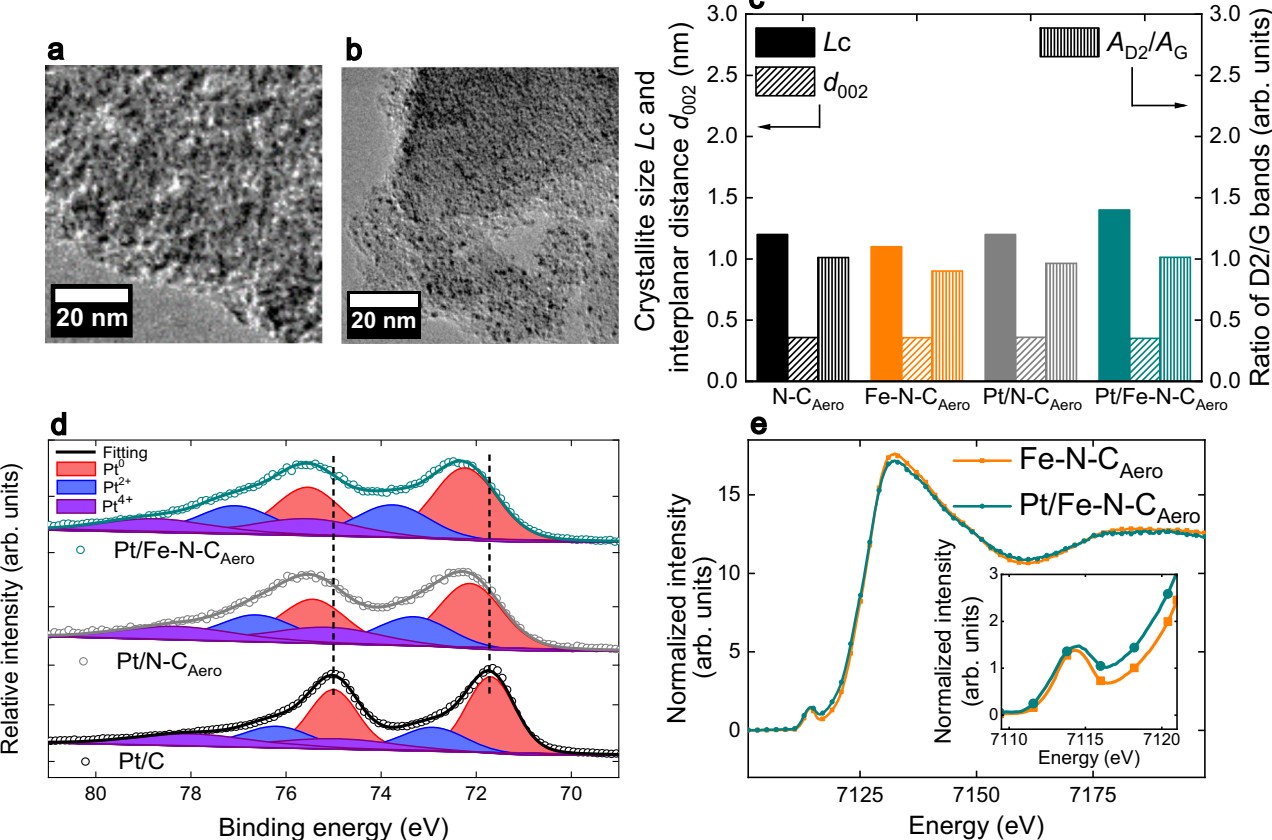

**Fig. 1 | Physical and structural characterization of synthesized aerogel and commercial catalysts.** Transmission electron microscopy micrographs of (**a**) Fe–N–C$_{Aero}$ and (**b**) Pt/Fe–N–C$_{Aero}$. **c** Carbon crystallite size perpendicular to the graphene layers (Lc) and interlayer spacing (d$_{002}$) obtained from X-ray diffraction and values of the ratio between D$_2$ and G bands (A$_{D2}$/A$_G$) from Raman spectra.

**d** X-ray photoelectron spectroscopy in the Pt 4 f region for the synthesized and the commercial Pt/C catalysts. **e** Fe Kα high-energy-resolution fluorescence-detected X-ray absorption near edge structure for Fe–N–C$_{Aero}$ and Pt/Fe–N–C$_{Aero}$. Inset graph show pre-edge and rising edge regions. Source data for (**c**–**e**) are provided as a Source Data file. These measurements were only performed once.

edge regions (inset of Fig. 1e). This suggests a subtle electronic influence of Pt NPs on FeN$_4$ sites, since these regions are associated with Fe electronic structure and local geometry[24,38]. The identical Fe Kβ XES spectra for Fe–N–C$_{Aero}$ and Pt/Fe–N–C$_{Aero}$ (Supplementary Fig. 8) confirmed that the average spin state of iron in FeN$_4$ sites was unmodified in their resting state[38]. The fitting of the Fe K-edge EXAFS spectrum of Pt/Fe–N–C$_{Aero}$ revealed the absence of Fe-Pt interaction, and Fe-N and Fe-O coordination numbers and bond distances similar to those in Fe–N–C$_{Aero}$ (Supplementary Fig. 3b and Supplementary Table 3) independently confirming the unmodified FeN$_4$ site structure in Fe–N–C$_{Aero}$ and Pt/Fe–N–C$_{Aero}$ and absence of PtFe alloys and other Pt-Fe bonds.

Following the structural investigation of as-prepared materials, their electrochemical response was examined. Cyclic voltammograms (CVs) in Ar-saturated 0.1 M H$_2$SO$_4$ electrolyte showed similar profiles for N–C$_{Aero}$, Fe–N–C$_{Aero}$, Pt/N–C$_{Aero}$ and Pt/Fe–N–C$_{Aero}$ (Fig. 2a), reinforcing the minimal changes in N-doped carbon chemistry upon Pt deposition. Notably, Pt/Fe–N–C$_{Aero}$ and Pt/N–C$_{Aero}$ exhibited small but discernible oxidation/reduction peaks below 0.1 V, attributed to underpotentially deposited H atoms onto Pt (H$_{upd}$)[39]. As Pt NPs covered by an FeO$_x$ shell (spontaneously formed upon annealing of a Pt salt and Fe–N–C at high temperature) exhibit no H$_{upd}$ region[21,26], the H$_{upd}$ features seen here indicate that a pure Pt surface is exposed, also confirmed by CO stripping (Supplementary Fig. 9).

The polarization curves recorded using an RDE setup revealed a discrete enhancement in ORR activity upon Pt NP addition onto Fe–N–C$_{Aero}$ (Fig. 2b). The calculated mass activities at 0.8 V (MA$_{@0.8V}$) confirm this (Fig. 2c) with Fe–N–C$_{Aero}$ and Pt/Fe–N–C$_{Aero}$ reaching 2.71

and 3.95 A g$^{-1}$$_{powder}$, respectively. Pt/N–C$_{Aero}$ displayed a MA$_{@0.8V}$ of only 1.19 A g$^{-1}$$_{powder}$, demonstrating that FeN$_4$ sites are the major contributors to the overall activity of Pt/Fe–N–C$_{Aero}$.

It is nevertheless of interest to assess the Pt mass activity of these composites. Assigning the ORR activity of Pt/N–C$_{Aero}$ solely to Pt leads to a MA$_{@0.9V}$ of 5.5 A g$^{-1}$$_{Pt}$ (see Supplementary Note 1). For Pt/Fe–N–C$_{Aero}$, one may assume the Pt contribution to be the difference (3.95 − 2.71) A g$^{-1}$$_{powder}$, leading to an estimated MA$_{@0.9V}$ of 5.8 A g$^{-1}$$_{Pt}$ (0.9 V is used here to ease the comparison to literature of Pt activity). These numbers are comparable, and much lower than MA$_{@0.9V}$ of state-of-art Pt/C, with values of 100–200 A g$^{-1}$$_{Pt}$[3,40]. The low MA$_{@0.9 V}$ for Pt in the present composite catalysts (with 1 wt. % Pt) may be due to the localization of the Pt particles in the micropores of Fe–N–C.

A kinetic current density plot across low overpotentials, comparing the performance of Pt/Fe–N–C$_{Aero}$ with the mathematical sum of the individual performances of Fe–N–C$_{Aero}$ and Pt/N–C$_{Aero}$, further supports a synergy between Pt NPs and FeN$_4$ sites within this hybrid catalyst (Supplementary Fig. 10). The experimental curve for Pt/Fe–N–C$_{Aero}$ shows indeed higher ORR activity than the curve obtained from the mathematical sum of the curves of Fe–N–C$_{Aero}$ and Pt/N–C$_{Aero}$, showing that the activity observed for Pt/Fe–N–C$_{Aero}$ is more than the mere sum of its FeN$_4$ sites and its Pt particles. The effect can hardly be explained by a simple change in Pt particle size between Pt/Fe–N–C$_{Aero}$ and Pt/N–C$_{Aero}$, as the average size of Pt nanoparticles is similar (1.65 ± 0.05 and 1.69 ± 0.05 nm, see Supplementary Fig. 5). This synergy could be a true kinetic effect (increased kinetic current with unmodified ORR selectivity), or a selectivity effect (the Pt addition favoring 4-electron ORR). This is discussed in more detail later.

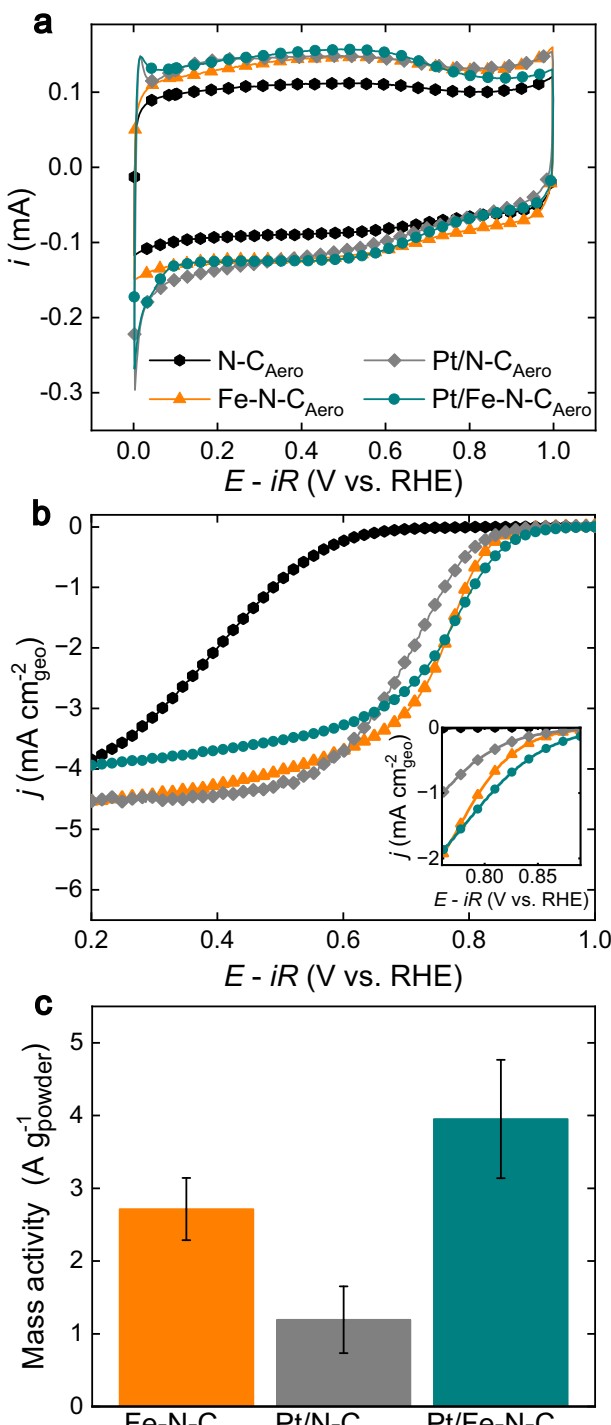

**Fig. 2 | Electrochemical evaluation of the catalysts. a** Cyclic voltammograms measured in Ar-saturated electrolyte at $v = 10\,\text{mV s}^{-1}$, (**b**) oxygen reduction reaction polarization curve measured in $O_2$-saturated $0.1\,\text{M}\,H_2SO_4$ electrolyte at $v = 2\,\text{mV s}^{-1}$ and $\omega = 1600\,\text{rpm}$, $T = 25\,°C$ with a graph inset in the kinetic region. **c** Mass activity values at 0.8 V derived after corrections for pseudocapacitive current, and $O_2$ diffusion limitation. Catalyst loading of $400\,\mu g_{\text{powder}}\,\text{cm}^{-2}_{\text{geo}}$. The error bars in (**c**) are the standard deviation obtained from at least two different measurements. The potential was corrected for iR-drop and the measured resistance was ca. $18\,\Omega$. Source data for (**a–c**) are provided as a Source Data file.

## Pt NPs enhance the durability of Fe−N−C$_{\text{Aero}}$ catalyst

To assess the beneficial role of Pt NPs on the durability of Pt/Fe−N−C$_{\text{Aero}}$, we conducted various load-cycle (LC)-ASTs in $0.1\,\text{M}\,H_2SO_4$

electrolyte, all involving 10,000 cycles. These included tests at 25 °C or 80 °C, Ar- or $O_2$-saturated electrolytes, a lower potential limit of 0.60 V and an upper potential limit (UPL) of 0.92 or 1.00 V (Supplementary Table 7). The UPL of 0.92 V was chosen to minimize carbon oxidation, aligning with recent recommendation from the United States Department of Energy (US DoE)[41]. A higher UPL of 1.00 V was also employed to promote carbon corrosion[15]. Given that both carbon oxidation kinetics and Fe demetallation rates from FeN$_4$ sites increase with temperature[16,42], ASTs were conducted at 25 or 80 °C. Insights into the adverse effect of $H_2O_2$ and ROS generated via Fenton reactions were gained by comparing results from Ar and $O_2$ gases conditions. The influence of these different ASTs on the normalized MA decay for Fe−N−C$_{\text{Aero}}$ and Pt/Fe−N−C$_{\text{Aero}}$ is presented in Fig. 3. The results obtained for Fe−N−C$_{\text{Aero}}$ (represented by orange columns in Fig. 3) indicate that the decay in MA$_{@0.8V}$ increased with higher temperature or higher UPL values, consistent with literature[15,43]. ASTs under Ar atmosphere lead to only *ca.* 20 % activity loss at 25 °C (ASTs 1-2) and to *ca.* 45-60 % activity loss at 80 °C (ASTs 3-4). The main deactivation mechanisms at play in such conditions are direct demetallation from acid-unstable FeN$_4$ sites and indirect demetallation as a result of carbon corrosion, especially for AST-4 combining high temperature and the high UPL value of 1.0 V. The presence of a fraction of intrinsically acid-unstable FeN$_4$ sites in Fe−N−C$_{\text{Aero}}$ is logical due to its last synthesis step which includes a heat-treatment at 950 °C in 10 % NH$_3$ and 90 % N$_2$. Ammonia pyrolysis is known to increase the activity of Fe−N−C catalysts via increasing their surface basicity (highly basic N-groups formed), but resulting in compromised FeN$_4$ site stability in acidic medium[35]. Surprisingly, Pt/Fe−N−C$_{\text{Aero}}$ experiences slightly more activity loss than Fe−N−C$_{\text{Aero}}$ upon ASTs 1 to 3, which may be assigned to platinum-catalyzed carbon corrosion in such mild stressing conditions. Enhanced carbon corrosion can lead to loss of FeN$_4$ sites, or simply lead to the formation of oxygen functional groups on the carbon surface and ensuing decreased TOF of FeN$_4$ sites[13]. For AST-4, the high UPL value and high temperature likely resulted in Pt surface oxidation at UPL. This is known to deactivate the ability of platinum to catalyze carbon corrosion[44], and can explain the reversed trend of relative stability of Pt/Fe−N−C$_{\text{Aero}}$ and Fe−N−C$_{\text{Aero}}$ between AST-4 and AST 1-3.

Also agreeing with former findings[7,12], ASTs conducted under $O_2$ (ASTs 5 to 8) showed a pronounced reduction in MA$_{@0.8V}$ compared to those under Ar (ASTs 1 to 4). Under the most severe conditions ($O_2$ and 80 °C, AST-7 and AST-8), ORR MA$_{@0.8V}$ losses exceeded 70 % of the initial value. Additionally, two experiments were performed to evaluate the impact of $O_2$ on the deactivation of Fe−N−C$_{\text{Aero}}$. Thin-film RDEs were placed in an oven under ambient air at 25 or 80 °C, and their MA$_{@0.8V}$ was measured before and after 17 h (average time for a 10k LC-AST), corresponding to AST-9 and AST-10, respectively. A 20-30% relative decrease in MA$_{@0.8V}$ values of Fe−N−C$_{\text{Aero}}$ was observed, in line with the recent report of Santos et al. on the aerobic ageing of Fe−N−C materials[45]. Moving to Pt/Fe−N−C$_{\text{Aero}}$, ASTs under Ar atmosphere resulted in comparable MA$_{@0.8V}$ losses to Fe−N−C$_{\text{Aero}}$ (compare orange and dark cyan columns for ASTs 1 to 4 in Fig. 3). Furthermore, no significant trend could be discerned after AST-9 and AST-10, comparing the two catalysts. However, a substantial improvement in durability was observed with Pt/Fe−N−C$_{\text{Aero}}$ for ASTs performed in $O_2$ (compare the orange and dark cyan columns for ASTs 5 to 8 in Fig. 3). The enhanced durability of Pt/Fe−N−C$_{\text{Aero}}$ over Fe−N−C$_{\text{Aero}}$ was most visible under the harshest conditions, namely AST-7 and AST-8 (80 °C, $O_2$) with a final MA$_{@0.8V}$ approximately doubled. These remarkable improvements suggest that the electronic interaction between Pt NPs and FeN$_4$ sites in the Fe−N−C matrix may be (i) mitigating $H_2O_2$/ROS formation and/or (ii) reducing Fe demetallation under oxygenated conditions, the two main degradation mechanisms for Fe−N−C materials[5,13,14,16].

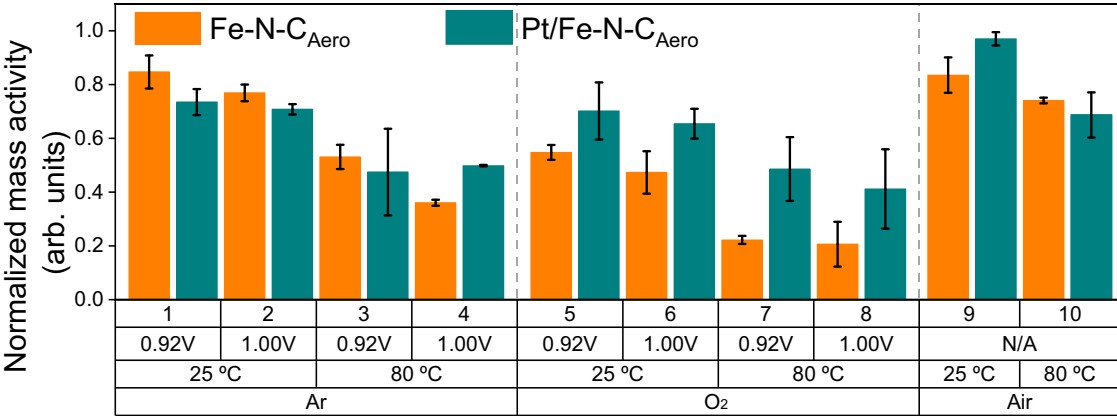

**Fig. 3 | Normalized mass activity (final/initial) of Fe−N−C$_{Aero}$ and Pt/Fe−N−C$_{Aero}$ for the ten different AST conditions.** The values were obtained from polarization curves recorded in $O_2$-saturated 0.1 M $H_2SO_4$ after pseudocapacitive current and $O_2$ diffusion limitation. $v = 2$ mV s$^{-1}$, $\omega = 1600$ rpm, $T = 25$ °C, catalyst loading of 400 $\mu g_{powder}$ cm$^{-2}_{geo}$. The normalization is made dividing the final value (after the specific AST) by the initial value (before AST) and can be found in Supplementary Table 7. The error bars are the standard deviation obtained from at least two

different measurements. The first row indicates the AST number, the second row the UPL value during AST, the third row the temperature during AST and the fourth row the gas used to saturate the electrolyte during AST. The as-measured ORR polarization curves are displayed in Supplementary Figs. 11 and 12. The potential was corrected for iR-drop and the measured resistance was ca. 18 Ω. Source data are provided as a Source Data file.

## On the role of Pt NPs in enhancing the durability of Fe−N−C$_{Aero}$

To rationalize the changes in durability between Fe−N−C$_{Aero}$ and Pt/Fe−N−C$_{Aero}$, rotating ring disk electrode (RRDE) experiments were conducted to quantify $H_2O_2$ production and the number of electrons involved in ORR (Supplementary Fig. 13). For these measurements, a thin-film with low catalyst loading was used (100 $\mu g$ cm$^{-2}_{geo}$) to minimize secondary electroreduction or disproportionation of $H_2O_2$ and, therefore, accurately reflect the material's intrinsic selectivity.

Figure 4a illustrates that Fe−N−C$_{Aero}$ produced *ca.* 30 % of $H_2O_2$ at 0.65 V, while Pt/N−C$_{Aero}$ generated only about 5 %, and $H_2O_2$ formation was nearly suppressed on Pt/Fe−N−C$_{Aero}$ ( <0.5 %). Hydrogen peroxide reduction reaction (HPRR) measurements were conducted to investigate if the much-reduced $H_2O_2$ formation on Pt/Fe−N−C$_{Aero}$ (and to a lesser extent, on Pt/N−C$_{Aero}$) stemmed from increased HPRR activity linked to Pt NPs. Both platinized catalysts exhibit a slightly higher potential at zero current density, ($E_{j=0}$, *i.e.* the potential at which the HPRR proceeds at the same rate as $H_2O_2$ electro-oxidation to $O_2$) compared to Fe−N−C$_{Aero}$ (Fig. 4b). However, the HPRR polarization curves of Pt/Fe−N−C$_{Aero}$ and Pt/N−C$_{Aero}$ did not significantly improve compared to Fe−N−C$_{Aero}$, suggesting the slightly higher $E_{j=0}$ values may be the result of slow chemical disproportionation of $H_2O_2$ on Pt NPs (with in situ formed $O_2$ contributing to a reductive current alongside HPRR). Importantly, none of the samples displayed typical behavior of HPRR-active Pt, such as an expected onset potential ($E_{onset}$) around 0.95 V *vs.* RHE[20,46]. Moreover, based on the Levich equation, a theoretical diffusion-limited current density of 18.45 mA cm$^{-2}_{geo}$ for a bulk 10 mM $H_2O_2$ concentration and pH = 1 electrolyte (see calculation in Supplementary Note 2) exceeds the ~10 mA cm$^{-2}_{geo}$ obtained here. This indicates that HPRR activity of Pt NPs in these materials likely does not primarily account for the near-zero $H_2O_2$ detection in RRDE experiments during ORR and the enhanced durability of Pt/Fe−N−C$_{Aero}$.

Tafel slopes were derived between 0.1 and 1.0 A g$^{-1}$ from the polarization curves in Fig. 4c, revealing slopes at the beginning-of-life (BoL, before any AST) of -59 and -75 mV dec$^{-1}$ for Fe−N−C$_{Aero}$ and Pt/Fe−N−C$_{Aero}$, respectively. This difference likely stems from the lower ORR selectivity of Fe−N−C$_{Aero}$, which may imply different rate-determining steps and result in a mixed slope. Additionally, the slopes changed at the end-of-life (EoL, which in this work always corresponds to "after AST-7") for Fe−N−C$_{Aero}$ by -11 mV dec$^{-1}$ (from -59 to -70 mV dec$^{-1}$). This result is in line with observation by Choi et al. for an

Fe−N−C catalyst treated with $H_2O_2$, which introduced oxygen functional groups on the surface, reducing the TOF and selectivity towards 4-electron ORR while increasing the Tafel slope[13]. In contrast, the change in Tafel slopes was only from -75 at BoL to -80 mV dec$^{-1}$ at EoL for Pt/Fe−N−C$_{Aero}$, suggesting mitigated oxidation of the Fe−N−C matrix, likely due to $H_2O_2$ suppression. This is supported also by the lack of change in the CVs under Ar of Pt/Fe−N−C$_{Aero}$ at the BoL and EoL, while Fe−N−C$_{Aero}$ shows signs of surface oxidation with a higher pseudocapacitive current at the EoL (Supplementary Fig. 14).

Quantitative physical and chemical information on Fe−N−C$_{Aero}$ and Pt/Fe−N−C$_{Aero}$ after different ASTs was obtained using TEM coupled with energy dispersive x-ray spectroscopy (X-EDS). TEM images of Fe−N−C$_{Aero}$ showed the absence of Fe agglomeration after exposing the electrode for 17 h to the acid electrolyte (17 h is the average time of an LC-AST), referred to 'Acid exposure' stage, therefore highlighting that any observed agglomeration in the discussed ASTs originates from said ASTs (Supplementary Fig. 15a). A decrease of the Fe content from 0.25 to 0.09 at. % was observed during the 'Acid exposure' for Fe−N−C$_{Aero}$. However, as depicted by X-EDS values in Fig. 4d, regions displaying a low-density Fe content (associated with FeN$_4$ sites only, in Supplementary Fig. 15b) co-exist with other areas with a high-density Fe content (Supplementary Fig. 15c) at 'EoL' stage. The low-density Fe zones maintained an Fe at. % comparable to those at the 'Acid exposure' stage (0.04 *vs.* 0.09 at. % Fe), while high-density Fe zones exhibited a much higher Fe content (1.1 at. %) (Supplementary Table 8). This suggests the transformation during AST-7 on Fe−N−C$_{Aero}$ of a substantial fraction of FeN$_4$ sites into Fe clusters (likely FeO$_x$ particles) leading to high density zones[47], consistent with its reduced ORR activity after AST-7. In contrast, no high-density Fe zones were observed on Pt/Fe−N−C$_{Aero}$ at EoL (Supplementary Fig. 15d and Supplementary Table 9), indicating reduced demetallation of Fe from FeN$_4$ sites, in line with its doubled ORR activity at EoL (Fig. 3). Overall, these findings suggest that FeN$_4$ sites are on average less degraded when Pt NPs are incorporated onto the matrix. The addition of Pt NPs almost suppressed $H_2O_2$ generation during ORR, and no high-density Fe zones were identified on Pt/Fe−N−C$_{Aero}$ after AST-7.

## Evaluation of catalysts in PEMFC and online ICP-MS coupled to GDE setup

Given the enhanced durability of Pt/Fe−N−C$_{Aero}$ in RDE setup, it was evaluated in PEMFC using membrane electrode assemblies (MEAs) and

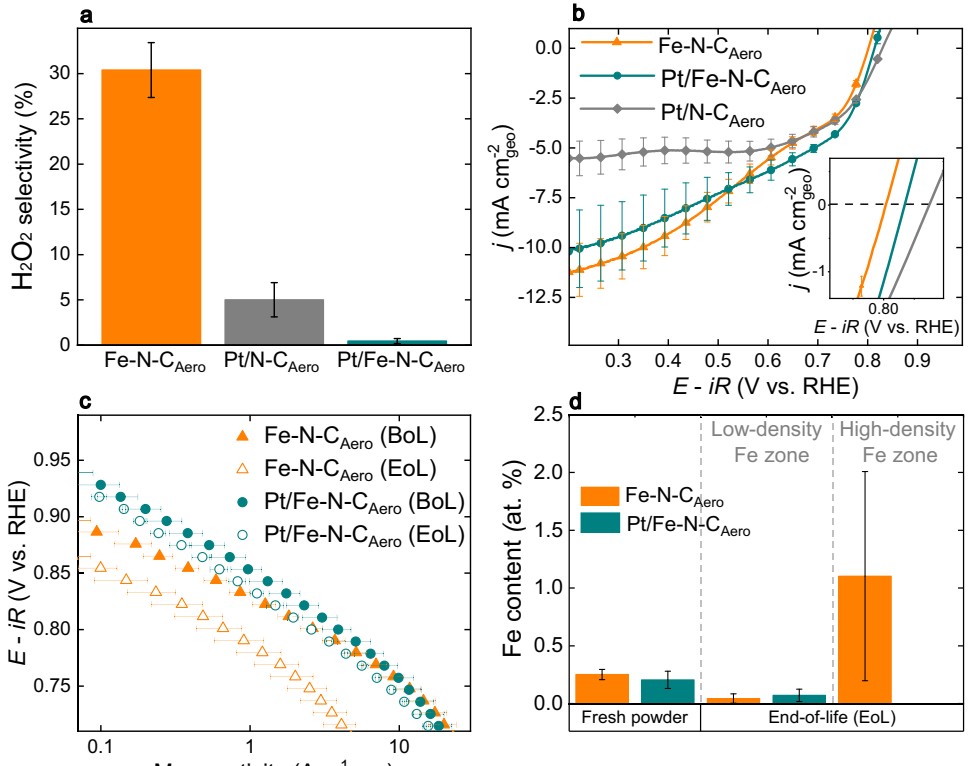

**Fig. 4 | Effect of platinum addition on the ORR selectivity, HPRR activity, ORR stability, and associated changes in Fe speciation.** All electrochemical measurements were performed at $v = 2\ mV\ s^{-1}$, $\omega = 1600\ rpm$, T = 25 °C. **a** $H_2O_2$ percentage during ORR at 0.65 V for 100 $\mu g_{powder}\ cm^{-2}_{geo}$ loading. **b** HPRR polarization curves in Ar-saturated 0.01 M $H_2O_2$ + 0.1 M $H_2SO_4$ electrolyte using 200 $\mu g_{powder}\ cm^{-2}_{geo}$ loading. **c** ORR polarization curves before/after AST-7,

corrected for background current and $O_2$-diffusion limitation. **d** Fe content measured by energy-dispersive X-ray spectroscopy for fresh powder and end-of-life of Fe−N−C$_{Aero}$ and Pt/Fe−N−C$_{Aero}$. The error bars are the standard deviation obtained from at least two different measurements. The potential was corrected for iR-drop and the measured resistance was ca. 18 Ω. Source data for (**a**–**d**) are provided as a Source Data file.

compared to Fe−N−C$_{Aero}$. Figure 5a presents the polarization curves measured before and after 20 h at 0.5 V, referred to as beginning-of-test (BoT) and end-of-test (EoT) stages, respectively. The BoT polarization curves were similar for both catalysts, with Pt/Fe−N−C$_{Aero}$ exhibiting slightly better performance, consistent with RDE results. After 20 h at 0.5 V, the performance of Fe−N−C$_{Aero}$ dropped dramatically across all current densities, while the MEA based-on Pt/Fe−N−C$_{Aero}$ showed only a slight decrease. At 0.8 V, the relative decrease in current density was 89 % for Fe−N−C$_{Aero}$ but only 27 % for Pt/Fe−N−C$_{Aero}$ (inset of Fig. 5a). The chronoamperometries recorded during the 20 h at 0.5 V (Fig. 5b) revealed rapid degradation for MEA with Fe−N−C$_{Aero}$, whereas those based on Pt/Fe−N−C$_{Aero}$ showed a slight improvement over the first 3 h before a subtle decline. Importantly, the Fe-free Pt/N−C$_{Aero}$ catalyst exhibited a very low performance at both BoT and EoT (Supplementary Fig. 16), demonstrating that the Pt NPs cannot independently account for the initial and final ORR activity of Pt/Fe−N−C$_{Aero}$. Together with the improved durability of Pt/Fe−N−C$_{Aero}$, this unambiguously demonstrates that the low amount of Pt NPs in the Pt/Fe−N−C$_{Aero}$ hybrid catalyst significantly mitigates the decrease in site density and/or TOF of FeN$_4$ sites, rather than directly contributing to ORR activity.

We then studied the sensitivity of the stabilization effect on the amount of Pt added, with Pt contents of 0.25, 0.50 wt. % on Fe−N−C$_{Aero}$. The results of 50 h PEMFC tests at 0.5 V show that full stabilization is reached with 1 wt. % Pt while a performance loss is observed for lower Pt contents, the loss increasing with decreasing Pt content (Supplementary Fig. 17). Thus, for Pt/Fe−N−C$_{Aero}$, the lowest Pt amount needed for efficient stabilization is *ca.* 1 wt. %. This threshold Pt amount may differ across various Fe−N−C materials.

For demonstrating the applicability of our approach to other Fe−N−C materials, we used a commercial Fe−N−C catalyst from Pajarito Powder (PMF D14401, labeled Fe−N−C$_{Paj}$) comprising only FeN$_4$ sites, as confirmed by Mössbauer spectroscopy (Supplementary Fig. 2b). Fe−N−C$_{Paj}$ is highly mesoporous while possessing also a significant amount of micropores (Supplementary Fig. 1 and Supplementary Table 1). This catalyst was functionalized with *ca.* 1 wt. % Pt NPs. Representative TEM images of Fe−N−C$_{Paj}$ and Pt/Fe−N−C$_{Paj}$ show the absence of Fe NPs and the presence of Pt NPs, respectively (Supplementary Fig. 18a, b). The average Pt particle size in Pt/Fe−N−C$_{Paj}$ was 1.96 nm, slightly higher but comparable to the one measured in Pt/Fe−N−C$_{Aero}$ (1.65 nm). The higher Pt particle size of Pt/Fe−N−C$_{Paj}$ may be assigned to its lower BET area (755 $m^2\ g^{-1}$) compared to Pt/Fe−N−C$_{Aero}$ (1191 $m^2\ g^{-1}$).

The BoT polarization curves were similar for Fe−N−C$_{Paj}$ and Pt/Fe−N−C$_{Paj}$ (Supplementary Fig. 18c). During the 20 h durability test at 0.5 V, Pt/Fe−N−C$_{Paj}$ exhibited a break-in process in the first 10 h, increasing its current density until reaching a steady value (Supplementary Fig. 18d). In contrast, the CA curve at 0.5 V for Fe−N−C$_{Paj}$ showed a continuous decay in current density from the start.

To gain quantitative insights into Fe dissolution from Fe−N−C catalysts and their platinized versions, we combined a scanning GDE half-cell with online ICP-MS (S-GDE-ICP-MS)[48]. A scheme of the custom-built cell is shown in Supplementary Scheme 1. Galvanostatic ASTs were applied in oxygenated 0.1 M HClO$_4$ electrolyte to Fe−N−C$_{Aero}$ and Pt/Fe−N−C$_{Aero}$ catalysts (see Methods section and Supplementary Note 3 for details). As shown in Fig. 5c, first, a low current density of 0.1 mA $cm^{-2}_{geo}$ was applied to stabilize the Fe dissolution signal near open-circuit potential. After, an AST switching between the current

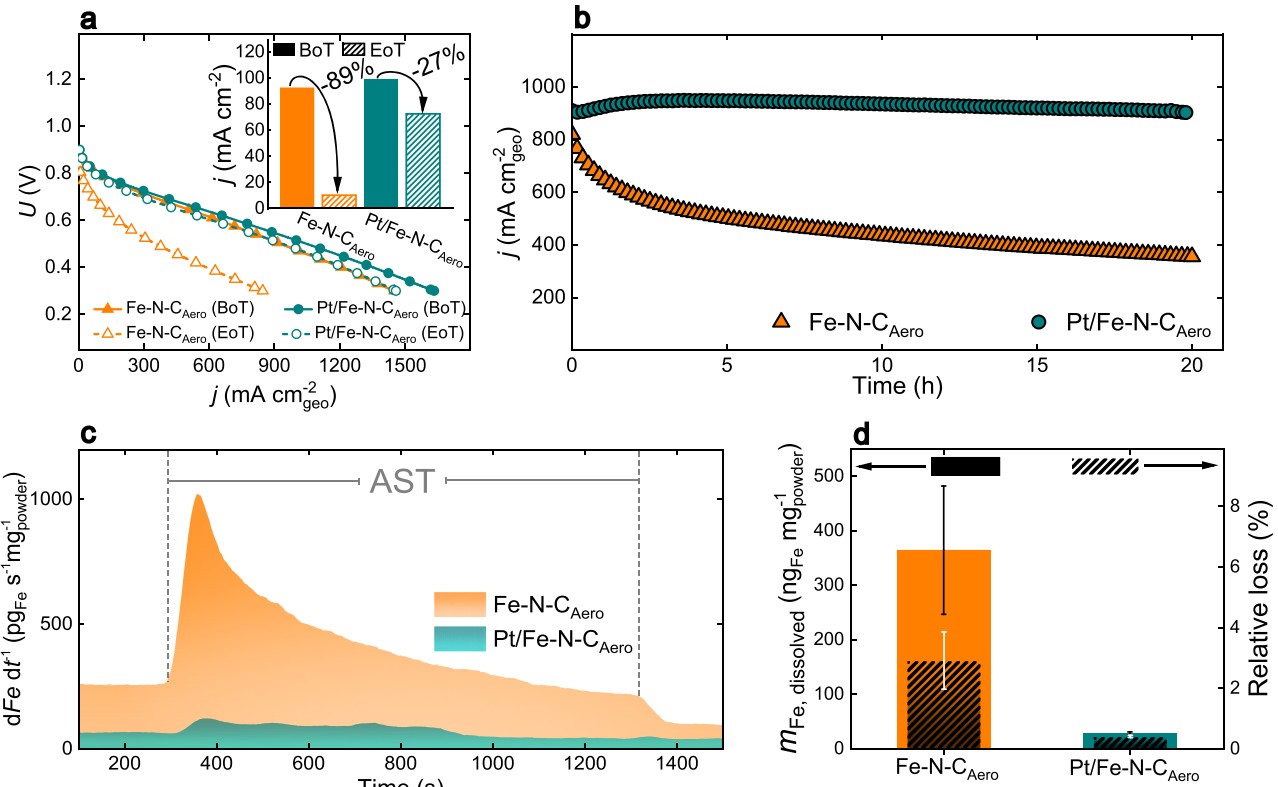

**Fig. 5 | Characterization of the catalysts in PEMFC device or in GDE setup.** PEMFC experiments under an operating temperature of 80 °C, with fully humidified $H_2$ and $O_2$ with flow rate of 150 mL min$^{-1}$ and a backpressure of 1 bar on each side. The loadings were 4.1 mg$_{powder}$ cm$^{-2}_{geo}$ at cathode and 0.5 mg$_{Pt}$ cm$^{-2}_{geo}$ at anode. **a** polarization curves at $v = 1$ mV s$^{-1}$ at the beginning and end of the test (at $U = 0.5$ V for 20 h), BoT and EoT, respectively for Fe–N–C$_{Aero}$, and Pt/Fe–N–C$_{Aero}$. The inset shows the current densities at $U = 0.8$ V. **b** Chronoamperometry at $U = 0.5$ V. All reported PEMFC voltages and polarization curves are not corrected for iR-drop. **c** Online ICP-MS results conducted in a scanning GDE setup under room temperature in an $O_2$-saturated 0.1 M HClO$_4$ electrolyte at a flow rate of 50 mL min$^{-1}$.

The Fe dissolution rates of Fe–N–C$_{Aero}$ and Pt/Fe–N–C$_{Aero}$ were normalized to catalyst loading of 0.93 mg$_{powder}$ cm$^{-2}_{geo}$ and 1.03 mg$_{powder}$ cm$^{-2}$, respectively, during an accelerated stress test consisting of 200 square cycles of 3.2 s at -49.7 mA·cm$^{-2}_{geo}$ and 2.1 s at -0.1 mA·cm$^{-2}_{geo}$. The iR-drop was 100 % post corrected for each current density. **d** Total amount of dissolved Fe species normalized by catalyst loading (absolute amount) as well as relative to the initial Fe content. For (**a**) and (**b**) these measurements were only performed once. The error bars in (**d**) are the standard deviation obtained from two different measurements for each sample. Source data for (**a**–**d**) are provided as a Source Data file.

density steps at -0.1 and -49.7 mA cm$^{-2}_{geo}$ show that the Fe dissolution signal significantly increased for Fe–N–C$_{Aero}$, confirming direct Fe demetallation, as well as the possible adverse effects of $H_2O_2$ and ROS produced during ORR. In contrast, the Fe dissolution rate was almost negligible at any given time for Pt/Fe–N–C$_{Aero}$, highlighting the beneficial role of Pt NPs in protecting FeN$_4$ sites. Consequently, a 30-fold reduction in absolute Fe dissolution was observed for Pt/Fe–N–C$_{Aero}$ compared to Fe–N–C$_{Aero}$ (filled bars in Fig. 5d). When normalized by the initial amount of Fe, losses amounted to 2.9% for Fe–N–C$_{Aero}$, contrasting with a mere 0.4 % loss for Pt/Fe–N–C$_{Aero}$ (striped bars in Fig. 5d). Activity tests performed pre- (Supplementary Fig. 19a–d) and post-AST (Supplementary Fig. 19e–h) in the S-GDE-ICP-MS system further confirmed the pronounced instability of Fe–N–C$_{Aero}$ compared to the much lower Fe dissolution rates for Pt/Fe–N–C$_{Aero}$. These results quantitatively confirm that the enhanced durability of the platinized samples is linked to the preservation of FeN$_4$ species, also related to the absence of clusters at EoL in Pt/Fe–N–C$_{Aero}$ catalysts (Fig. 4d).

### Analysis by $^{57}$Fe Mössbauer spectroscopy of FeN$_4$ sites' stabilization
To explore in further details the fate of various FeN$_4$ sites, we resorted to $^{57}$Fe Mössbauer spectroscopy, a proven technique to identify the structure and amount of FeN$_4$ sites in Fe–N–C materials, and which can also be extended to in situ or *post mortem* studies when working with $^{57}$Fe enriched samples[17,18,35,49]. For this purpose, we used a previously

well characterized Fe–N–C catalyst derived from ZIF-8, phenanthroline and ferrous acetate by an Ar pyrolysis, labeled Fe$_{0.5}$ (see synthesis in Methods section). Its $N_2$ sorption isotherm has a shape similar to that of Fe–N–C$_{Aero}$, but with lower micropore volume and slightly higher mesopore volume (Supplementary Fig. 1 and Supplementary Table 1). $^{57}$Fe(II) acetate was used to prepare the sample ensuring a sufficient signal-to-noise ratio for Mössbauer spectroscopy on a 5 cm$^2$ cathode. The Mössbauer spectrum of the Fe$_{0.5}$ powder (Supplementary Fig. 2c) displays two quadrupole doublets, D1 and D2, identified as high-spin Fe(III)N$_4$ sites and low- or medium-spin Fe(II)N$_4$ sites, respectively[17,18]. The spectrum of the as-prepared Fe$_{0.5}$ cathode (Fig. 6a) and the corresponding fitting parameters (Supplementary Table 10) reveal the same D1 and D2 quadrupole doublets[17], along with a low amount of a sextet assigned to ferric oxide. The Mössbauer spectrum of the as-prepared Pt/Fe$_{0.5}$ cathode (Fig. 6b), following the deposition of *ca.* 1 wt. % Pt by the same method used for other Fe–N–Cs in this work and subsequent cathode preparation, reveals the same D1 and D2 signatures as the Fe$_{0.5}$ powder. Notably, there is no discernable sextet signal. These results indicate that our soft deposition method for Pt NPs did not alter the FeN$_4$ speciation initially present in Fe$_{0.5}$ and suggest that the Pt NPs' presence mitigated the transformation of some FeN$_4$ sites into ferric oxide during cathode preparation.

The PEMFC polarization curves for Fe$_{0.5}$ and Pt/Fe$_{0.5}$ cathodes are shown in Fig. 6c, illustrating both the BoT and EoT stages (after 50 h at 0.5 V in $H_2$ / $O_2$ PEMFC at 80 °C). The Fe$_{0.5}$ cathode experienced a

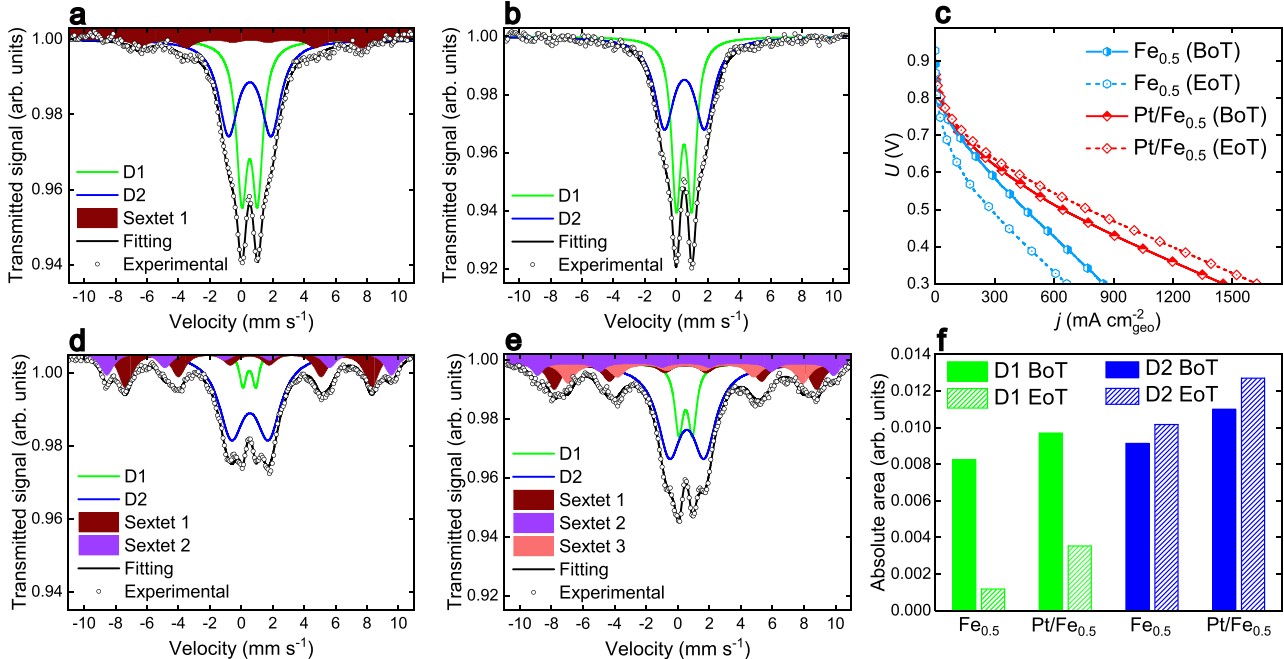

**Fig. 6 | $^{57}$Fe Mössbauer spectroscopy identification of FeN$_4$ sites partial stabilization by Pt addition. a** Spectrum of the as-prepared Fe$_{0.5}$ cathode, (**b**) spectrum of the as-prepared Pt/Fe$_{0.5}$ cathode, (**c**) polarization curves at $v = 1$ mV s$^{-1}$ at BoT and EoT for Fe$_{0.5}$ and Pt/Fe$_{0.5}$, (**d**) spectrum of the Fe$_{0.5}$ cathode after 50 h at $U = 0.5$ V in PEMFC, (**e**) spectrum of the Pt/Fe$_{0.5}$ cathode after 50 h at $U = 0.5$ V in PEMFC, (**f**) absolute area below the fitted D1 (green) or D2 (blue) component at BoT (filled column) and EoT (dashed column). PEMFC measurements were performed at 80 °C, with fully humidified H$_2$ and O$_2$ with flow rate of 150 (for Pt/Fe$_{0.5}$) or 60 mL min$^{-1}$ (for Fe$_{0.5}$) and a backpressure of 1 bar on each side. The loadings were 4.1 mg$_{powder}$ cm$^{-2}_{geo}$ at cathode and 0.5 mg$_{Pt}$ cm$^{-2}_{geo}$ at anode. All reported PEMFC voltages and polarization curves are uncorrected for iR-drop. $^{57}$Fe Mössbauer spectra were acquired at -268 °C. The data for Fe$_{0.5}$ in Fig. 6a, c, and d were published in ref. 17. Source data for (**a–f**) are provided as a Source Data file. These measurements were only performed once.

significant performance drop over the 50 h test (-38% at 0.5 V, -58 % at 0.8 V), while the Pt/Fe$_{0.5}$ cathode showed slight improvement ( +20 % at 0.5 V, +30 % at 0.8 V). The $^{57}$Fe Mössbauer spectra for the Fe$_{0.5}$ and Pt/Fe$_{0.5}$ cathodes post-durability test shed light on the fate of the FeN$_4$ species. The two striking observations are (i) retained D2 signal intensity in both Fe$_{0.5}$ (compare Fig. 6a and d) and Pt/Fe$_{0.5}$ (compare Fig. 6b and e) cathodes before and after the tests, and (ii) a higher D1 signal intensity at EoT in the Pt/Fe$_{0.5}$ cathode compared to the Fe$_{0.5}$ cathode (compare the signal intensity of D1 in Fig. 6d, e). Since all the spectra in Fig. 6 were acquired on cathodes made with the same catalyst loading of 4 mg$_{powder}$ cm$^{-2}_{geo}$, one can moreover lean on the relative absorption of the γ-ray flux (the y-axis scale) for a quantitative comparison of the absolute amount of each Fe species before and after test. The absolute area below the D1 and D2 quadrupole doublets was mathematically obtained from the fittings, and is plotted in Fig. 6f to assess the absolute changes in FeN$_4$ species. As previously reported, one can see that the high-spin Fe(III)N$_4$ species associated with D1 is the less durable site[17]. However, the absolute amount of D1 sites after 50 h test is three times higher in Pt/Fe$_{0.5}$ than in Fe$_{0.5}$. This demonstrates that the presence of Pt NPs partially stabilized the intrinsically unstable D1 site, explaining at least in part the improved durability of the Pt/Fe$_{0.5}$ cathode during PEMFC operation. In contrast, the D2 signal intensity remained constant or slightly increased from BoT to EoT, (Fig. 6f). For further details on the lower performance at high current density of Fe$_{0.5}$ *vs.* Pt/Fe$_{0.5}$ BoT curves, see Supplementary Note 4 and Supplementary Fig. 20.

Moreover, comparing the BoT and EoT current density at 0.8 V, the slight improvement in ORR activity seen for Pt/Fe$_{0.5}$, Pt/Fe–N–C$_{Paj}$ and Pt/N–C$_{Aero}$ during the test can be assigned to fully stabilized FeN$_4$ sites combined with an activation of Pt NPs, as observed on Pt/N–C$_{Aero}$ (Supplementary Figs. 21–22). While the relative contribution of the activated Pt to the overall EoT activity may be non-negligible for the less active Pt/Fe$_{0.5}$ and Pt/Fe–N–C$_{Paj}$

cathodes, it can contribute only to a minor extent to the EoT activity of Pt/Fe–N–C$_{Aero}$, the EoT activity of the latter being *ca.* 4 times higher than the EoT activity of Pt/N–C$_{Aero}$ (Supplementary Fig. 22). Interestingly, there seems to be a correlation between the change in activity at 0.8 V of Pt/Fe–N–C hybrids during the PEMFC durability test and the ratio of D2/(D1 + D2) signals as derived from Mössbauer spectroscopy. The higher the D2/(D1 + D2) signal in the parent Fe–N–C material, the higher is the activity increase during PEMFC durability test (Supplementary Fig. 22). The latter may be assigned to Pt NP activation, which however requires a high amount of D2 (low amount of D1) initially, to not be offset by the activity decrease related to the loss of some of the D1 sites.

To answer whether the improved durability of FeN$_4$ sites in Pt/Fe–N–C hybrids requires a direct interaction between a Pt NP and an FeN$_4$ site, we first performed calculations to compare the number of Pt NPs (particles·g$^{-1}$) to the number of FeN$_4$ sites (sites·g$^{-1}$) in Pt/Fe–N–C hybrids. This was performed for various FeN$_4$ site densities, SD (10$^{19}$, 3·10$^{19}$ and 10$^{20}$ sites·g$^{-1}$), and Pt particle size (1.0, 1.7 and 2.5 nm). The methodology is detailed in Supplementary Note 6. The SD of 3·10$^{19}$ sites·g$^{-1}$ corresponds to the value experimentally measured for Fe$_{0.5}$[50]. For the base case (3·10$^{19}$ FeN$_4$ sites·g$^{-1}$, Pt particle size of 1.7 nm), the ratio of FeN$_4$ sites to Pt particles is *ca.* 160 at the Pt loading of 1 wt. % (Supplementary Fig. 23). This infers that the stabilization effect of FeN$_4$ sites by Pt is long-range and does not require a Pt particle to be in atomic distance to an FeN$_4$ site. This holds promise for developing strategies to rationally stabilize FeN$_4$ sites, by post-synthesis modification of Fe–N–C catalysts. While 1 wt. % Pt on Fe–N–C may at first sight appear to be a low content, with PEMFC cathodes of 4 mg$_{powder}$·cm$^{-2}_{geo}$, it still leads to a significant 40 µg$_{Pt}$·cm$^{-2}_{geo}$ loading at the cathode. The comparison of the PEMFC performance obtained with a Pt/Fe–N–C$_{Aero}$ cathode (4 mg$_{powder}$·cm$^{-2}_{geo}$ of Pt/Fe–N–C$_{Aero}$, corresponding to 40 µg$_{Pt}$·cm$^{-2}_{geo}$) or with a cathode prepared from a commercial 40 wt.% Pt/C (at 40 µg$_{Pt}$·cm$^{-2}_{geo}$) shows that similar

activities at 0.8 V are obtained (after the break-in period needed to activate commercial Pt/C, as is well-known), but at high current density, the Pt/Fe−N−C$_{Aero}$ cathode surpasses the thin Pt/C layer (Supplementary Fig. 21 and Supplementary Note 5). Nevertheless, the ultimate goal is still the stabilization of Fe−N−C catalysts by non-PGM sites, which can synergistically act with FeN$_4$ sites in the same way as is reported in the present work.

### Possible long-range interaction of Pt from computational chemistry

To better understand the long-range interaction between Pt NPs and FeN$_4$ sites, we resorted to density functional theory (DFT) calculations on periodic models of the FeN$_4$C$_{10}$ and FeN$_4$C$_{12}$ sites (Supplementary Fig. 24). We carried out geometry optimization of the model structures, with or without Pt on the graphene plane, and with or without OOH adsorbed on the Fe center (the atomic coordinates of the optimized models are provided in Supplementary Data 1–28). The effect of the total spin ($S_{tot}$) imposed on the periodic structures was also investigated. The unit cells comprise a single FeN$_4$ site and a single Pt atom, which is adsorbed on the graphene sheet at the longest possible distance from the Fe center (Supplementary Fig. 24). While different from the physicochemical description of the real catalysts (a Pt NP comprising > 100 atoms), such models are appropriate to investigate the long-range effect since the Pt-Fe distances (7–10 Å) are sufficiently long to avoid direct cooperative effects between Pt and Fe, and the Pt to Fe atomic ratio of 1 in the models is close to the ratio existing at the surface of the real catalysts. For example, with bulk contents of 1.5 wt. % Fe and 1 wt. % Pt and assuming 20% of Fe is on the surface, the Pt/Fe surface atomic ratio is 0.9. For both the FeN$_4$C$_{10}$ and FeN$_4$C$_{12}$ models, the Pt atom adsorbs in a bridge position (Supplementary Fig. 24) and has a negligible effect on the Fe-N distances, except for the combination of the highest total spin ($S_{tot}$ = 3) and the FeN$_4$C$_{10}$ model. In the latter case, the adsorption of Pt decreases the Fe-N distance by 0.057 Å (See Supplementary Tables 11-12), breaking the trend of increasing Fe-N distance with increasing $S_{tot}$ that is observed without Pt (Supplementary Table 12). The latter trend is in accordance with increased Fe-N distance with increasing spin that was experimentally reported for Fe(II)N$_4$ porphyrins[51], with the maximum Fe-N increase of *ca.* 0.1 Å also quantitatively in line with our results (Supplementary Table 12). This agreement supports the validity of our calculations. Similar effects (or lack of) of Pt are observed for the models including an OOH intermediate adsorbed atop the Fe center (Supplementary Fig. 24). For the FeN$_4$C$_{12}$ model, the Pt adsorption has a negligible effect on the Fe-N and Fe-O distances (variation of 0.001-0.002 Å, see Supplementary Tables 13-14), for any of the imposed spin states. In contrast, the Pt adsorption leads to significant shortening of the Fe-N bond for the FeN$_4$C$_{10}$ model for $S_{tot}$ = 3/2 (-0.023 Å) and $S_{tot}$ = 5/2 (-0.126 Å). The Fe-N bond shortening upon Pt adsorption is partly due to the mitigated out-of-plane displacement of Fe above the N$_4$ plane when it adsorbs OOH (see Supplementary Fig. 24 and Supplementary Table 15). For $S_{tot}$ = 5/2, the out-of-plane Fe displacement is even reduced from 0.049 Å without Pt to only 0.020 Å with Pt (Supplementary Table 15). This can explain the improved stability of such an FeN$_4$ site during ORR. Regardless of the rate-determining step of the ORR on FeN$_4$ sites (in enzymes or bio-inspired catalytic sites), it is recognized that the Fe center undergoes changes in its oxidation and spin states as a result of the necessary oxygenation and deoxygenation events[49,52]. Therefore, it can be concluded from DFT and for the FeN$_4$C$_{10}$ model that the presence of an adsorbed Pt atom situated far from the FeN$_4$ site stabilizes it by minimizing its structural changes experienced during the catalytic cycle. In contrast, for the considered FeN$_4$C$_{12}$ model, the Pt adsorption does not lead to shortening of the Fe-N bond. The different behavior is due to the presence of zig-zag carbon edge defects near the FeN$_4$ site for that model, as discussed later. While the FeN$_4$ structure is affected by the spin polarization of the distant Pt atom, the interaction energy

between OOH and Fe is not strongly modified (See Supplementary Fig. 25 and associated description).

To better understand the role of the adsorbed platinum atom in maintaining the geometric structure of FeN$_4$ upon OOH* adsorption for medium- and high-spin states in the FeN$_4$C$_{10}$ model and how this differs from the FeN$_4$C$_{12}$ structure, we computed the spin-density distribution. Supplementary Fig. 26 compares the spin density at Fe, $\rho_s$, with the total spin, while Supplementary Fig. 27 shows the spin-density distribution across all atoms in the (Pt)/FeN$_4$C$_x$ models for $S_{tot}$ = 2. The results indicate that when Pt is adsorbed on FeN$_4$C$_{10}$, it becomes spin-polarized, mitigating the increase of the spin density at Fe with increasing $S_{tot}$. This explains the mitigated deformation of the FeN$_4$ site upon OOH* adsorption in the presence of Pt. In the chosen FeN$_4$C$_{12}$ model, the spin polarization of nearby zig-zag edge carbon atoms limits the spin density at the Fe site. This in turn explains the lack of effect of Pt for the selected FeN$_4$C$_{12}$ model. Recently, the spin polarization of carbon atoms (and limited spin at the Fe center) was calculated by us for FeN$_4$C$_{10}$ moieties when they are integrated in carbon nanoribbons and in close proximity with the zig-zag carbon edge[53]. Thus, one can expect that a FeN$_4$C$_{12}$ model free of nearby defects in the graphene plane (such as the 2 f model in Fig. 1 of ref. 18) would behave similarly as the defect-free FeN$_4$C$_{10}$ model and that adsorption of Pt would similarly stabilize it.

Overall, the DFT insights provide a theoretical frame revealing the possible type of long-range electronic effects at play between Pt NPs and various FeN$_4$ sites in the real materials. More theoretical work is however desirable in the future, including larger systems and considering also the electrochemical potential and ORR catalytic cycle. The partial stabilization of the D1 fingerprint of FeN$_4$C$_{12}$ sites (Mössbauer spectroscopy) by Pt that is experimentally observed supports the existence of a plurality of D1 sites and, connected with the present DFT results, it is proposed that only the FeN$_4$C$_{10}$ and FeN$_4$C$_{12}$ sites far from zig-zag carbon edges can be stabilized by distant platinum. These results fall within but also expand the known magneto-structural correlations of FeN$_4$ square-planar complexes and ties with the recognized strong dependence of their metal-nitrogen bond strength on spin state[54]. On a broader perspective, it also connects with studies on the spin effect on oxygen electrocatalysis for various materials[55–58].

In summary, this study offers experimental and theoretical insights on the synergy taking place between Pt NPs and FeN$_4$ sites and how the durability of various Fe−N−C catalysts having different textural properties is improved. The work also provides a facile method for Pt NP deposition while preserving the physicochemical properties and Fe speciation of Fe−N−C matrices. Unexpectedly, the Pt NPs proved to be poorly active in the electroreduction of H$_2$O$_2$, but almost suppressed its production during ORR when combined synergistically with FeN$_4$ sites. This substantial decrease in H$_2$O$_2$ likely contributes to stabilizing FeN$_4$ sites during ORR, as evidenced by reduced Fe dissolution rates and partial stabilization of the less durable high-spin Fe(III)N$_4$ site (D1 Mössbauer fingerprint). Computational chemistry reveals how distant platinum objects can stabilize the structure of FeN$_4$ moieties by spin polarization effects. Beyond advancing fundamental understanding, these findings hold promise for developing a class of fuel cell catalysts with significantly reduced noble metal content.

## Methods

### Synthesis of Fe−N−C and Pt/Fe−N−C catalysts

An Fe−N−C aerogel catalyst was synthesized through a one-pot method[34,59]. In summary, a hydrogel composed of optimized quantities of resorcinol (99%, Alfa Aesar), formaldehyde (37 wt. % solution, Acros Organics), melamine (99%, Acros Organics), and anhydrous FeCl$_3$ (98%, Acros Organics) was first synthesized by a sol-gel process. Typical quantities used to synthesize Fe−N−C$_{Aero}$ were 3.6 g of resorcinol, 4.0 g of melamine (leading to a molar ratio of resorcinol/melamine of 2/2), 12 mL of formaldehyde and 0.11 g of FeCl$_3$. Resorcinol,

melamine and formaldehyde were mixed with the above specified amounts in deionized water, together with anhydrous sodium carbonate (99.9999%, Fluka) to promote polymerization. The mass of sodium carbonate was chosen so as to result in a molar ratio of sodium carbonate to resorcinol of 0.0165 (or 0.086 g sodium carbonate for 3.6 g resorcinol). The solids' content (mass of resorcinol plus melamine plus formaldehyde in the liquid phase) was adjusted to 20 wt. % by adapting the aliquot of water. The reactants and solvent were first stirred at 70 °C for 30 min, after which FeCl₃ was added. Subsequently, the pH was adjusted to 8 by adding 2 M NaOH solution (98% Alfa Aesar) under continuous stirring until gel formation. The hydrogel was then cured in a water bath at 70 °C for 120 h. The obtained hydrogels were subjected to a water-acetone exchange 3 times per day, for 3 days. This was followed by supercritical $CO_2$ drying, and subsequent pyrolysis of the organic aerogel under $N_2$ flow at 800 °C for 1 h. The aerogel piece was cut into small pieces and installed in a tube furnace. The latter was purged with $N_2$, and then heated with a ramp rate of 4 °C min⁻¹ and maintained for 1 h at 800 °C. Afterward, the furnace was turned off to cool down to room temperature while the $N_2$ flux was still on. The resulting carbonized aerogel underwent ball-milling before the acid wash step. A zirconium-oxide crucible of 50 mL with zirconium-oxide balls was used and installed in a Retsch S100 ball miller. The mass ratio of balls to sample was 30. The large pieces were pre-ground with 10 mm diameter balls at 400 rpm for 15 min, and further ground with 5 mm diameter balls at 400 rpm for 60 min. After this, the material was subjected to an acid-wash step at 80 °C using 0.5 M $H_2SO_4$ for 7 h under reflux conditions. Finally, the material was heat-treated at 950 °C for 45 min in 10% $NH_3$, 90% $N_2$. The optimized resorcinol/melamine molar ratio of 2/2 showed significant micro- and mesoporous volumes and enhanced ORR activity[34]. Herein, the catalyst is labeled Fe−N−C$_{Aero}$. We also studied a commercial Fe−N−C catalyst from Pajarito Powder prepared by silica templating (PMF D14401, labeled Fe−N−C$_{Paj}$), and an Fe−N−C catalyst from CNRS derived from ZIF-8 (labeled Fe$_{0.5}$)[13,16]. To prepare Fe$_{0.5}$, commercial ZIF-8 (Basolite® Z1200, Sigma Aldrich), 1,10-phenanthroline (≥99%, Sigma Aldrich) and Fe(C₂H₃O₂)₂ (≥99.99%, Sigma Aldrich) were mixed in weight ratios of 4/1 for ZIF-8/phenanthroline and 0.5 wt. % of iron using planetary ball milling (Fritsch Pulverisette). Typically, 800 mg ZIF-8 (Basolite Z1200), 200 mg phenanthroline and 16 mg of Fe(II) acetate (99.99 %, sigma Aldrich, ref. 517933) were mixed together. These amounts are poured in a zirconium oxide crucible along with 100 zirconium oxide balls (5 mm diameter), then milled at 400 rpm for 2 h (Fritsch Pulverisette 7 Premium, Fritsch, Idar-Oberstein, Germany). Subsequently, a flash pyrolysis of 1 h at 1050 °C under Ar flow was performed after which the split-hinge oven was opened, the tube was removed and let to cool-down for 20 min under Ar flow. The catalyst was not subjected to any acid wash or other post-pyrolysis treatment.

Pt NPs were prepared by a modified polyol method and deposited on Fe−N−C$_{Aero}$, Fe$_{0.5}$ or Fe−N−C$_{Paj}$, labeled as Pt/Fe−N−C$_{Aero}$, Pt/Fe$_{0.5}$ and Pt/Fe−N−C$_{Paj}$, respectively. Pt NPs were first synthesized by dissolving $H_2PtCl_6 \cdot 6H_2O$ (266.4 mg, Sigma-Aldrich) in 25 mL of ethylene glycol under vigorous stirring for 30 min. An aqueous solution of 0.1 M NaOH (Sigma-Aldrich, >99%) was slowly added to the solution until the pH reached 12. The pH was continuously measured with a pH meter. Then, the solution was placed in a microwave oven (Proline model GM2025) and set to 700 W for 75 s, forming a colloidal suspension of Pt NPs. The solution was cooled to room temperature and left for 12 h with continuous stirring (solution A). Solution B was prepared by dispersing the Fe−N−C catalyst (150 mg) in a solution of water (16 mL): ethanol (4 mL) (molar ratio of 4:1). Then, an aliquot of Solution A (822 μL) was added in Solution B to reach a nominal 1 wt. % Pt loading onto any given Fe−N−C. A solution of 0.1 M $H_2SO_4$ (Sigma-Aldrich) was added to reach pH 3. After stirring for 24 h, the solution was filtered and washed with water and ethanol. The resulting hybrid catalyst was dried in air for 12 h at 80 °C.

## Electrochemical measurements

RDE measurements were performed in a three-electrode cell composed of a reversible hydrogen electrode (RHE, Hydroflex, Gaskatel GmbH), a glassy carbon (GC) plate as counter-electrode and a thin-film catalyst layer drop-casted onto GC as a working electrode. RRDE measurements were performed using a GC disk and a Pt-ring (Pine Research) separated by polytetrafluoroethylene (PTFE). The geometric area of the working electrode was 0.196 cm² for both RDE and RRDE setups. A Pt-wire immersed in the electrolyte was also connected to the above-mentioned RHE reference electrode in parallel through a capacitor. This is used to filter the high frequency electrical noise of the electrical network and to avoid disturbing the low frequency electrochemical measurements (it acts as a high-pass filter). More details on the dual-reference system used in this work can be found in ref. 60. The electrode potentials are reported vs. this RHE. These measurements were conducted using a bi-potentiostat (Autolab PGSTAT302N) and recorded using NOVA 2.1 software. The data was exported and plotted using OriginLab software. All glassware used was cleaned overnight in Caro's acid (50% v/v of $H_2SO_4/H_2O_2$), rinsed, and boiled in pure water (Millipore, 18.2 MΩ cm, 1 – 3 ppm total organic compounds) before use. The catalyst suspension was prepared at least 12 h prior to use by mixing 10 mg of the catalyst powder with 50 μL of a Nafion solution (5 wt. % in a mixture of alcohols and water), 854 μL of 2-propanol, and 372 μL of pure water (Millipore, 18.2 MΩ cm, total organic compounds <3ppb). The mixture was sonicated for 20 min after its preparation and another 20 min before use. An aliquot of the suspension (10 μL for RDE and 2.5 μL for RRDE) was drop-casted onto a GC disk embedded in a polychlorotrifluoroethylene (PCTFE) cylinder, and used as the working electrode. The catalyst loading was 400 and 100 μg$_{powder}$ cm⁻²$_{geo}$ for RDE and RRDE measurements, respectively. Since the same ink formulation is used for Fe−N−C and Pt/Fe−N−C materials, the same total mass of catalyst powder is deposited but in the case of Pt/Fe−N−C, the total mass comprises the Pt mass. If the Pt content in the total mass of a given Pt/Fe−N−C is 1.0 wt. %, then the calculated Pt geometric loadings are 4.0 and 1.0 μg$_{Pt}$ cm⁻²$_{geo}$ for RDE and RRDE measurements, respectively. Upon introduction in the glass cell at open-circuit potential, 50 linear scan CVs between 0.0 and 1.0 V, at 100 mV s⁻¹, followed by 3 linear scan CVs at 10 mV s⁻¹ and 1 staircase CV at 2 mV s⁻¹ in the same potential range were performed in 0.1 M $H_2SO_4$ (Merck, Suprapur 96 wt. %) electrolyte previously de-aerated with Ar (>99.999 %, Messer). The ORR polarization curves were obtained by performing 1 staircase CV between 0.0 and 1.0 V at $v = 2$ mV s⁻¹, $\omega = 1600$ rpm, $T = 25$ °C in $O_2$-saturated 0.1 M $H_2SO_4$ electrolyte. They were corrected for pseudocapacitive currents by subtracting the CV performed at 2 mV s⁻¹ in the Ar-purged electrolyte, and only the negative-going scans were plotted. The LC-AST consisted of 10,000 square wave potential cycles between 0.60 V (3 s) and the UPL (3 s) in various temperature, gas atmosphere, and UPL conditions, as listed in Supplementary Table 5. The CVs and ORR polarization curves were performed with dynamic Ohmic-drop (iR-drop) correction. The resistance was ca. 18 Ω and compensated at 80 % during the measurements. The same procedure was employed to assess the activity of the catalysts for HPRR. However, in this case, a 200 μg$_{powder}$ cm⁻²$_{geo}$ catalyst loading was used (5 μL aliquot of the ink). The curves were performed in an Ar-saturated 0.01 M $H_2O_2$ + 0.1 M $H_2SO_4$ electrolyte. All electrolytes were prepared on the same day as the measurements, and were prepared from concentrated sulfuric acid (and hydrogen peroxide) that was stored in a fridge. It implies in particular that a fresh electrolyte was prepared and used for each AST. The electrolyte volume contained in the electrochemical cell was 50 mL for the 'characterization cell' used for ORR activity measurements and 75 mL for the 'degradation cell' used for AST. For the ORR polarization curves a 'characterization cell' was kept at 25 °C. Then, the thin-film electrode was transferred to a 'degradation cell,' where the LC-ASTs were performed in 0.1 M $H_2SO_4$ electrolyte purged with Ar or $O_2$ gas and

maintained at a controlled temperature of 25 or 80 °C. Following AST, the thin-film electrode was once again transferred to the 'characterization cell,' and the procedure involving break-in, characterization cycles, and ORR polarization curves was repeated again at 25 °C. In this study, beginning-of-life ('BoL') denotes the catalyst stage following the break-in, characterization CVs and the ORR polarization curves, *i.e.* prior any AST. Furthermore, end-of-life ('EoL') pertains to the final stage, occurring after the AST. When not explicitly stated, 'EoL' specifically denotes the period 'After AST-7'.

Kinetic current values for ORR ($i_k$) were calculated from the Koutecky-Levich equation (Eq. 1) and subsequently normalized by the catalyst loading to obtain the mass activities (Eq. 2).

$$i_k = -\frac{(i_L . i)}{(i_L - i)} \qquad (1)$$

$$MA = \frac{i_k}{m} \qquad (2)$$

where $i_L$ is the diffusion-limited current at 0.2 V and $\omega = 1600$ rpm, $i$ is the faradaic current obtained after corrections for iR-drop and pseudocapacitive current, and $m$ is the catalyst loading deposited on the glassy carbon electrode.

RRDE experiments were performed using the same procedure as for RDE measurements. The Pt-ring was held at 1.2 V *vs.* RHE during the polarization curve. The average percentage of $H_2O_2$ species generated at the disk (% $H_2O_2$) and the average number of electrons in the ORR ($n_{e^-}$) was obtained from Eq. 3 and Eq. 4, respectively:

$$\% H_2O_2 = \frac{2\frac{i_r}{N}}{\left(\frac{i_r}{N}\right) + i_d} \times 100 \qquad (3)$$

$$n_{e^-} = \frac{4i_d}{i_d + \left(\frac{i_r}{N}\right)} \qquad (4)$$

where $i_d$ and $i_r$ are the absolute values of the disk and the ring current, respectively, and $N$ is the collection efficiency (0.24 for this system).

The MEAs for PEMFC measurements with Fe−N−$C_{Aero}$ or Pt/Fe−N−$C_{Aero}$ were prepared using a procedure similar to that described in our previous publication[34]. The ink composition for the cathodic catalyst layers was prepared by mixing 20 mg of catalyst, 915 μL of 5 wt. % Nafion solution (1100 EW Nafion, Sigma-Aldrich), 300 μL of 1-propanol (>99.9% purity, Sigma-Aldrich), and 272 μL of deionized water, resulting in a mass ratio of dry Nafion to catalyst of 2. The ink was sonicated for 1 h and drop casted with aliquot of 210 μL each time, followed by drying, on the microporous layer side of a gas diffusion layer (Sigracet 28-BC from Baltic Fuel Cells) with an area of 4.84 cm². This process was repeated until the complete ink had been drop casted, resulting in a catalyst loading of 4.1 $mg_{powder}$ cm⁻²$_{geo}$. Then, the cathode was dried at 80 °C, subsequently hot-pressed at 135 °C onto a Nafion membrane (Nafion NR-211, used as-received, thickness of 25.4 μm). The anode used for all PEMFC tests was 0.5 $mg_{Pt}$ cm⁻²$_{geo}$ on Sigracet 28BC (balticFuelCells GmbH). For the two other Fe−N−C catalysts and their platinized versions (Fe−N−$C_{Paj}$, Pt/Fe−N−$C_{Paj}$, $Fe_{0.5}$ and Pt/$Fe_{0.5}$), the cathode ink was composed of 652 μL of 5 wt. % Nafion solution (1100 EW Nafion, Sigma-Aldrich), 326 μL of 1-propanol (>99.9 % purity, Sigma-Aldrich), and 272 μL of deionized water, all mixed with 20 mg of catalyst, resulting in a Nafion/Catalyst mass ratio of 1.4. The ink deposition process was as described above. For comparison, a Pt/C cathode with a low loading of 40 $μg_{Pt}$·cm⁻² was prepared using a commercial Pt/C catalyst (40 wt. % Pt on Vulcan XC72). The ink composition was 2.0 mg of Pt/C catalyst, 16.5 μL of 5 wt. % Nafion solution (1100 EW Nafion, Sigma-Aldrich), 410 μL of 1-propanol (>99.9 % purity,

Sigma-Aldrich), and 272 μL of deionized water, resulting in a mass ratio of dry Nafion to carbon of 0.6. The ink was sonicated for 1 h and an aliquot of 174.6 μL was drop casted, followed by drying, on Sigracet 28-BC. The resulting MEAs were then tested using a 5 cm² active area single-cell equipped with a serpentine flow field (Fuel Cell Technologies). An 850E Scribner Associates Fuel Cell test station was used while the cell was controlled by a potentiostat from BioLogic (SP-150) and a 20 A booster from BioLogic. The EC-Lab software was used for data acquisition. The data was exported and plotted using OriginLab software. The operating conditions were temperature of 80 °C, fully humidified $H_2$ and $O_2$ on anode and cathode side, respectively, flow rate of 150 mL min⁻¹ and a backpressure of 1 bar on each side. Here, beginning-of-test and end-of-test will be referred to as BoT and EoT respectively, with 'test' referring to either a 20 h or 50 h chronoamperometry at a cell voltage of 0.5 V. The polarization curves were recorded by scanning the cell voltage at 1 mV s⁻¹, from 0.9 to 0.3 V and then back to 0.9 V. The plotted polarization curves are the average of the negative-going and positive-going scans. The reported cell voltage in all figures is uncorrected for *iR*-drop.

## Physicochemical characterizations

TEM coupled to X-EDS was carried out with a JEOL JEM-2010 series TEM instrument operated at 200 kV with a resolution of 0.19 nm. X-EDS analyses were recorded in at least five different regions of the TEM grid. STEM-X-EDS elemental maps were acquired using a JEOL 2100 F microscope operated at 200 kV equipped with a retractable large angle Centurio Silicon Drift detector. The quantitative analyses were performed on the Fe *K*, C *K* and Pt *M* lines using the *K*-factors provided by the JEOL software. XRD results were obtained using an X'Pert PRO MPD PANalytical diffractometer, conducted at 45 kV and 40 mA in Bragg-Brentano mode, utilizing Cu (K$\alpha$ mean) radiation with a wavelength ($\lambda$) of 1.5419 Å. The scanning range covered 2θ angles from 10 to 100°, with increments of 0.033° and a step-time of 9.4 s. The $d_{002}$ and $L$c values were calculated using the Bragg and Debye-Scherrer equations[61]. Raman spectroscopy measurements were conducted using a Renishaw InVia instrument configured for backscattering, with an excitation wavelength of 532 nm. An x50 objective (NA = 0.75) was utilized to focus and capture both the incident and scattered radiation. The spectral resolution was ~1 cm⁻¹. The instrument was operated in "line mode" and low incident power density of 50 μW μm⁻² was used to prevent excessive sample heating or degradation under the focused laser beam. This setup allowed for the acquisition of averaged information over an area of *ca.* 50 μm² in a single exposure. XPS analysis were conducted using a Thermo Scientific K$\alpha$ spectrometer, equipped with a monochromatic Al X-ray source (spot size 400 μm). Binding energies were corrected with CasaXPS software, considering as reference the graphitic carbon ($sp^2$) component of C 1$s$ peak at 284.8 eV. A commercial 10 wt. % Pt/C benchmark catalyst (TEC10E10A from Tanaka Kikinzoku Kogyo, TKK) was used as reference. Fe *K*$\alpha$ HERFD-XANES and Fe *K*$\beta$ XES spectra were recorded at the ID26 beamline of the European Synchrotron Radiation Facility (ESRF). To generate the incoming photons, three undulators (u35) were employed, and this radiation was subsequently monochromatized using a pair of cryogenically cooled Si(111) crystals. The incident beam's energy calibration was accomplished by using a reference metallic iron foil, setting the first inflection point of the Fe *K*-edge at 7112 eV. For *K*$\beta$ XES spectra, data was collected with a step size of 0.2 eV at an incident energy of 7500 eV. In the experimental setup, Germanium (Ge) crystal analyzers with a radius of 1 m were used alongside the sample and photon detector, arranged in a Rowland geometry. Specifically, Ge(620) and Ge(440) analyzers were employed to detect the *K*$\beta$ (7058 eV, Bragg angle 79°) and *K*$\alpha$ (6404 eV, Bragg angle 75°) emission lines, respectively. For the measurements, the catalyst powders were mixed in a proportion of 1:3 in mass with $BN_3$ (Aldrich®) and 1 ton pressed to obtain pellets.

For EXAFS, Fe $K$-edge X-ray absorption spectra were collected at room temperature at SAMBA beamline (Synchrotron SOLEIL, France). The beamline is equipped with a Si 220 monochromator and two Pd-coated mirrors, also used to remove X-ray harmonics. The catalysts were pelletized as disks of 10 mm diameter using boron nitride as a binder. Spectra were recorded in transmission (Fe–N–$C_{Aero}$) and fluorescence (Pt/Fe–N–$C_{Aero}$) mode using ionization chambers and a 35-elements Ge detector, respectively. The EXAFS data analysis was performed with the GNXAS code, which interprets the $\chi(k)$ signal by decomposing it into a sum of n-body distribution functions, $\gamma(n)$, derived through multiple-scattering (MS) theory. A detailed explanation of the GNXAS theoretical approach can be found in refs. 62,63. The Fe–N–$C_{Aero}$ and Pt/Fe–N–$C_{Aero}$ coordination shells around Fe were modeled with Γ-like distribution functions, which depend on four parameters: the coordination number CN, the distance $R$, the Debye-Waller factor $\sigma^2$ and the skewness $\beta$. Optimization was carried out also for $E_0$ (core ionization threshold energy) and $S_0^2$.

[57]Fe Mössbauer spectra were acquired in transmission mode with a spectrometer from Wissel, Germany. The γ-rays were produced by a [57]Co:Rh source. The velocity driving system was used in constant acceleration mode and the velocity waveform was triangular. Calibration of the velocity scale leaned on the sextet of a high-purity α-Fe foil measured at room temperature. The spectra were fitted by least-square method using combinations of a variable number of quadrupole doublets and sextets with a Lorentzian distribution of their spectral lines. The values of isomer shift of each fitted spectral component are reported relative to that of α-Fe measured at room temperature. No constraints were applied to the parameters used to fit the spectra. A circular holder with 2 cm$^2$ area was used to install the materials in powder form. To analyze $Fe_{0.5}$ and Pt/$Fe_{0.5}$ catalysts, a $Fe_{0.5}$ batch was prepared with [57]Fe(II) acetate while all other steps were identical ($Fe_{0.5}$ and Pt/$Fe_{0.5}$ synthesis, etc). A PerkinElmer NexION 2000c ICP-MS instrument was used to determine the total Fe and Pt content in the catalysts. Calibration curves were established using daily prepared [56]Fe and [195]Pt solutions derived from a commercial standard mono-element ICP-MS solution (Carl Roth GmbH & co. KG, 1000 mg L$^{-1}$), which were designed to create five-point calibration curves at concentrations of 0, 2, 5, 10, and 20 µg L$^{-1}$. Rhodium ([103]Rh) served as the internal standard. To eliminate polyatomic interferences associated with $^{40}Ar^{16}O^+$ ions, analyses were performed using the helium collision mode.

## Online S-GDE-ICP-MS

S-GDE-ICP-MS measurements were conducted on the Fe–N–$C_{Aero}$ and on Pt/Fe–N–$C_{Aero}$ catalysts. The electrodes were first prepared for S-GDE-ICP-MS measurements. The layers were prepared by doctor-blade coating an ink on a gas diffusion medium with a microporous layer (H23C8, Freudenberg, 3 × 3 cm$^2$, 216 ± 8 µm). The ink consists of a catalyst powder, commercial Nafion solution (Fuel Cell Store, D2021, 20 wt.% Nafion), and 2-Propanol (Supelco, EMSURE, ACS ISO), being 53.8 mg, 382 mg and 650 mg for Fe–N–$C_{Aero}$ and 55.9 mg, 367 mg and 675 mg for Pt/Fe–N–$C_{Aero}$, respectively. With this ink formulation, the weight ratio of dry Nafion to catalyst is 1.42 and the sum of the mass of the catalyst and dry Nafion represents 12 wt. % in the ink. For the ink preparation, the commercial Nafion solution was first fully mixed with 2-propanol, and then the catalyst power was added into this solution. Then, the ink was stirred for 1 h, sonicated for 1 h at a temperature between 20 and 30 °C (100 W VWR Ultrasonic Cleaner USC 500 THD), and then stirred until the coating step. During the doctor-blade coating, the plate of the automated film applicator (Zehntner, ZAA 2300) was at 30 °C and the thickness of the applied ink was 300 µm. The catalyst layers were dried at 60 °C under atmospheric pressure for 1 h and then at 60 °C under reduced pressure for 1 h. The resulting catalyst loadings are 0.93 mg$_{Fe-N-CAero}$·cm$^{-2}_{geo}$ and 1.03 mg$_{Pt/Fe-N-CAero}$·cm$^{-2}_{geo}$,

as measured by weighing the punched out circular area of the GDL before and after ink deposition and drying ($\Delta m$), and considering the mass ratio of catalyst to the total mass of solids deposited (dry Nafion ionomer and catalyst). According to the ink formulation, this ratio is equal to 1/(1 + 1.42). Before measurements, the catalyst layers were wetted with ultrapure water (Milli-Q IQ 7000, Merck) and kept floating for 1 h. As only a small part of the produced electrode (circle with diameter of 4 mm, corresponding to a geometric area of 0.1256 cm$^2$) is needed for the S-GDE, multiple measurements could be conducted using a fresh part of the prepared catalyst layer. For details of standard solutions for calibrations, see Supplementary Note 3. Using a GDE half-cell coupled to an ICP-MS (Perkin Elmer, NexIONTM 350X), the online Fe and Pt dissolution was detected by the ICP-MS. The S-GDE-ICP-MS technique was recently introduced by Reichmann et al., consisting of a small flow cell enabling high current measurements[48]. The ICP-MS was operated in dynamic reaction cell mode using CH$_4$ (N45, Air Liquide) to mitigate interferences associated with $^{40}Ar^{16}O^+$ ions. To track the long-term status of the ICP-MS, the internal standard solution was 1 wt. % HNO$_3$ (Rotipuran®Supra, ROTH) containing 2.5 µg·L$^{-1}$ of [74]Ge (Merck Centripur) and 2.5 µg·L$^{-1}$ of [187]Re (Merck TraceCERT), for calibrating the signals of [56]Fe and [195]Pt, respectively. Before every measurement day, the four-point calibration curves for Fe and Pt were obtained with the blank electrolyte and three standard solutions (0, 1, 5, 25 µg$_{Fe}$·L$^{-1}$ and 0, 1, 5, 25 µg$_{Pt}$·L$^{-1}$). The electrochemical protocol was applied on the GDEs in 0.1 M HClO$_4$ at room temperature with O$_2$ at a flow rate of 50 mL·min$^{-1}$ being purged from the gas diffusion layer-side of the GDEs. The electrolyte was freshly prepared every two to 3 days by diluting perchloric acid (supra quality, ROTIPURAN® (C), 70%, Roth, Germany) and was tested daily for iron contamination using ICP-MS before measurements. After each measurement, the system was flushed with 1% nitric acid (supra quality, ROTIPURAN® (C), 69%, Roth, Germany) to ensure the cell remained clean. Both the electrolyte and the internal standard were stored in plastic bottles, which were also prepared every two to 3 days. Each bottle was thoroughly cleaned before use and dedicated to a single purpose—one for the internal standard and one for the electrolyte. Here, 0.1 M HClO$_4$ was chosen as the electrolyte, instead of 0.1 M H$_2$SO$_4$ used mainly for R(R)DE measurements, in order to keep the $^{74}Ge^+$ signal from interfering with the noises contributed by $^{40}Ar^{34}S^+$ or $^{38}Ar^{36}S^+$ (with same $m/z$ value of 74 as $^{74}Ge^+$) that form if the H$_2$SO$_4$ electrolyte had been chosen. The protocol consists in galvanostatic techniques with three sub-protocols: the (i) pre-AST activity test, the (ii) AST, and the (iii) post-AST activity test. The sub-protocols (i) and (iii) were performed similarly, and they comprise six galvanostatic steps at -0.24, -4.96, -9.90, -24.83, -49.68 and finally -4.96 mA·cm$^{-2}_{geo}$. The first step has negligible dissolution and was held for 120 s to reach a steady potential, while 25 s was used for the other five steps. Between these steps, galvanostatic steps at -0.1 ± 0.02 mA·cm$^{-2}_{geo}$ were applied to separate the dissolution peaks induced at higher current densities. The AST consisted of 200 square cycles of 3.2 s at -49.7 mA·cm$^{-2}_{geo}$ and 2.1 s at -0.1 mA·cm$^{-2}_{geo}$. The measurements were carried out twice (each time on a fresh sample) to check reproducibility. The uncompensated resistance ($R_u$) was measured with electrochemical impedance spectroscopy at each current density using a BioLogic potentiostat (SP 150). The data was 100 % post iR-corrected. A customized Hydrogen Reference Electrode (Mini-Hydroflex, Item number 81020, gaskatel, shaft lengths: 15 mm) and carbon paper (Freudenberg E20) were used as reference and counter electrode, respectively. The S-GDE was situated above the flow field and equipped with a force sensor (ME Messinstrumente GmbH, KD45 50 N/VA/HT) to regulate a contact pressure of 45 N against the flow field. The samples were placed into the flow field, which itself is connected to an XYZ translational stage (Physik Instrumente, Germany M-403), enabling precise positioning of the flow field beneath the opening of the flow cell. The geometric area of the catalyst layer Ss-$_{GDE}$ was 0.0314 cm$^2$. The data analysis carried out to extract the metal

## DFT models of adsorbed Pt atom on graphene comprising $FeN_4C_x$ moieties

DFT-based calculations were carried out in periodic approaches using VASP.6.1 computer program[64–66]. Two models of graphene layers that integrate $FeN_4$ moieties were considered. The first model in Supplementary Fig. 24a, labeled as $FeN_4C_{10}$, includes pyridinic N atoms engaged in six-membered rings, whereas the other model, $FeN_4C_{12}$ in Supplementary Fig. 24b includes pyrrolic N-atoms engaged in five-membered rings. These two models were previously considered by us[18] and therefore, we selected them as representative structures in the present DFT investigation. A single Pt atom is adsorbed on the graphene and we explore the possible electronic, magnetic or geometrical changes on graphene and/or $FeN_4$ moieties resulting from this. The Pt atom is adsorbed at the border of every unit cell to allow for the longest possible Fe-Pt distance with the considered unit cells, i.e. about 7 Å in $FeN_4C_{10}$ models (Supplementary Fig. 24a) and about 10 Å in $FeN_4C_{12}$ models (Supplementary Fig. 24b). The effect of Pt adsorption on the Fe-OOH* interaction was also studied considering the end-on adsorption of OOH* on Fe. The adsorption strength of this intermediate is often regarded as a key in the ORR mechanism and selectivity ($H_2O_2$ formation).

The geometry of all structures in Supplementary Fig. 24a was fully optimized by imposing periodic boundary conditions on a 9.94 Å x 12.64 Å graphene unit cell that integrates in-plane the $FeN_4C_{10}$ moiety and on a 17.18 Å x 20.87 Å graphene unit cell that integrates in-plane but near zig-zag defects the $FeN_4C_{12}$ moiety. Vacuum region of 15 Å was introduced in the $z$-direction in order to eliminate interactions between the graphene sheet and its periodic images. Perdew−Burke−Ernzerhof's (PBE) exchange-correlation functional was used together with the recommended projector augmented-wave pseudopotentials (PAW)[66]. For the calculation of structural and electronic properties, standard PAW potentials supplied with VASP were used, with 4 valence electrons for C ($2s^2\,2p^2$), 5 valence electrons for N ($2s\,2p^3$), 8 valence electrons for Fe ($4s^2\,3d^6$), and 6 valence electrons for O ($2s\,2p^4$). In all cases, the Fermi-Dirac smearing method with sigma set to 0.03 was used.

Structural optimizations were carried out by imposing different total spins, $S_{tot.}$, on each structure shown in Supplementary Fig. 24, i.e. $S_{tot.} = 0, 1, 2$ or 3 on the ferrous (Pt)/$FeN_4C_x$ models and $S_{tot.} = 1/2, 3/2$, or $5/2$ on the ferric OOH/(Pt)/$FeN_4C_x$ models. In addition, all the calculations included support grid for the evaluation of the augmentation charges. The Bader charge density analysis with the implementation of Henkelman and co-workers in VASP code was employed to obtain the spin charge density[67], defined as $\rho_s = \rho_s\,(\alpha) - \rho_s\,(\beta)$, where α denotes spin-up and β denotes spin-down densities. The interaction energies of OOH* and the $FeN_4$-graphene models were computed as follows:

$$\Delta E = E_{tot}\,[OOH/FeN_4C_x]_{S_{tot}} - E_{tot}[FeN_4C_x]_{S_{tot}} \qquad (5)$$

where $S_{tot}$ is the total spin imposed on a structure. The same equation was used for the OOH / Pt / $FeN_4C_x$ structures.

## Data availability

Source data for all main figures and Supplementary Figures. and that are based on numerical values is provided with this paper. Coordinates of the optimized DFT models is provided as Supplementary data. The raw data of PEMFC measurements and RDE measurements (AST-7) for Pt/Fe-N-C$_{Aero}$ and Fe-N-C$_{Aero}$ generated in this study has been deposited in the figshare database [https://doi.org/10.6084/m9.figshare.26347222]. The X-ray absorption spectroscopy raw data associated with this work is permanently stored at SOLEIL and ESRF and available upon request. Source data are provided with this paper.

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

## Acknowledgements

N. B., K. K., L. D., M. M., H. Ge., S. B-F., F. M. and F. J. acknowledge support by the French National Research Agency in the frame of the ANIMA project (grant number ANR-19-CE05-0039) and the DEEP project (grant number ANR-21-CE05-0021). We acknowledge the European Synchrotron Radiation Facility (ESRF) for provision of synchrotron radiation facilities under proposal number MA-5765 and Synchrotron SOLEIL (Gif-sur Yvette, France) for provision of synchrotron radiation facilities at the SAMBA beamline. We also acknowledge Moulay Sougrati for the acquisition of Mössbauer spectroscopy data. N. A. I. acknowledges Conselho Nacional de Desenvolvimento Científico (CNPq, #140813/2021-7) and Coordenação de Aperfeiçoamento de Pessoal de Nível Superior/Comité Français d'Evaluation de la Coopération Universitaire avec le Brésil (CAPES/COFECUB, #88887.694322-2022-00). E. A. T. acknowledges São Paulo State Research Foundation (FAPESP, n° 2019/22183-6). I. R. and Y.-P. K acknowledge Christian Göllner for technical support during ICP-MS measurements and Jonas Möller for software development for data analysis. N. B. and F. J. acknowledge Marc Dupont and Frédéric Lecoeur for the support with PEMFC measurements. T. M. and H. G. acknowledge access to the HPC resources of CCRT/CINES/IDRIS, which was granted under the allocation AXXX0807369.

## Author contributions

N. A. I. and K. T. S.: Electrochemical characterizations in rotating disk electrodes and rotating ring disk electrodes and data analysis. N. B. and T. A.: Pt deposition on the various Fe-N-C catalysts, as well as MEA preparation and PEMFC tests and data analysis. Y-P. K, K. K., I. R and S. C.: ICP-MS online study with the GDE setup and data analysis. L. D.: Electron microscopy characterizations. M. M.: Raman spectroscopy measurements. H. Ge. and S. B-F.: Fe–N–C$_{Aero}$ catalyst development. V. S.; V. K. P and P.G.: XES measurements and data analysis. A. Z.: EXAFS measurement and analysis. T. M. and H. G.: DFT study. N. A. I., F. M. and F. J. conceptualized the study and wrote the first draft of the manuscript. E. A. T., F. M. and F. J.: Funding acquisition, work supervision and data validation. All authors actively contributed to the discussion and read and edited the manuscript.

## Competing interests

The authors declare no competing interests.
