## [Peer Review file · Nature Communications]

Evidence for the stabilization of FeN₄ sites by Pt particles during acidic oxygen reduction

Corresponding Author: Dr Frédéric Jaouen

Version 0:

Reviewer comments:

Reviewer #1

(Remarks to the Author)

Refer to submitted PDF file

Reviewer #2

(Remarks to the Author)

The submitted manuscript reports the stabilization mechanism of FeN₄ sites by Pt particles for oxygen reduction reaction in the acid medium. The characterization and electrochemical tests of catalysts are defined well, and essential presentation is also provided for them. Particularly, it is impressive that operando techniques are used combined with post mortem Mössbauer spectroscopy analysis. However, this manuscript might not be suitable for publication in Nature Communications due to some weaknesses described below.

1. The atomic-scale Pt-FeN₄ interaction has been well reported in the literature, and here the authors demonstrate the stabilization effect of the long-distance electronic effect. So, what's the contribution from each? There is a lack of experimental evidence showing the contact of Pt nanoparticles to FeN₄ sites or the Fe-Pt coordination. The authors create a complicated structure, but not uniform with high and low Fe zones, preventing the fundamental understanding of Pt-FeN₄ sites.

2. During the MEA test, Pt has a loading of 1% of the total 4.1 mg/cm², this leads to about 0.041 mgPt/cm² at the cathode side. The power density shown in the manuscript could be easily obtained for a MEA with only Pt/C nanoparticle catalysts at this loading. There's no experimental evidence with the manuscript to clarify whether the power performance obtained comes from the Pt nanoparticles or the Pt-FeN₄ sites. The manuscript compares Pt/Fe-N-C and Pt/N-C, showing that Pt/N-C has some much larger Pt nanoparticle and much lower power performance. So, does the higher activity of Pt/Fe-N-C comes from the interaction between FeN₄ sites and Pt nanoparticles, or just due to the improved dispersion and particle size distribution of Pt nanoparticles because of the FeN₄ sites?

3. During the AST under Ar, the including of Pt results in a worse stability for test 1, 2 and 3. The authors might add some further discussions to clarify this result.

Reviewer #3

(Remarks to the Author)

In this manuscript, the authors present a novel approach to enhancing Fe-N-C catalyst durability by incorporating Pt nanoparticles (NPs) via a 'soft polyol method', minimizing FeN₄ active structure modifications. Unlike conventional hybrid catalysts where Pt NPs contribute to ORR activity and/or act as H₂O₂ scavengers, the authors propose that Pt NPs in Fe-N-C catalysts suppress H₂O₂ generation and reduce Fe dissolution rates, improving catalyst stability. Furthermore, the study elucidates the stabilization of specific high-activity but unstable active sites by Pt NPs. The study also reveals the stabilization of high-activity but unstable active sites by Pt NPs, offering a new perspective on Fe-N-C catalyst durability enhancement. While some aspects require further refinement, this research has the potential to make a valuable contribution suitable for publication in Nature Communications. My major criticisms to this manuscript are given in below:

1. In this study, Pt 4f XPS spectral analysis revealed electronic interactions between Pt NPs and FeN₄ sites. However, the absence of metal-support interaction effects in N-CAero raises questions. Typically, the introduction of heteroatoms such as nitrogen into carbon supports is known to alter the support's electron density, subsequently influencing the electronic state of the deposited platinum. This phenomenon leads to the expectation that Pt/N-CAero should exhibit different characteristics from Pt/C. What, then, accounts for the similarity in electronic states between Pt/N-CAero and Pt/C, while only Pt/Fe-N-CAero demonstrates a change in the electronic state of Pt?
2. The XPS survey spectra of Fe-N-CAero and its platinum-deposited counterpart in Supplementary Fig. 4a do not provide information on iron content. It would be beneficial to include high-resolution XPS analysis results for the Fe 2p region.
3. The authors demonstrated electronic interactions between FeN₄ and Pt NPs through the observed up-shift in peak binding energy of the XPS 4f spectra, and changes in the electronic state of Fe as evidenced by Fe K-edge XANES analysis. However, they argue against the formation of PtFe alloys or Fe particle agglomeration based on the nearly identical XES spectra of Fe-N-CAero and Pt/Fe-N-CAero. Given that XES analysis is also sensitive to the oxidation state of elements, it is puzzling that no differences in Fe species were detected, unlike in the XANES results.
4. The observation of increased ORR activity after AST in platinum catalysts such as Pt/N-CAero, Pt/Fe-N-CPaj, and Pt/Fe_{0.5} warrants further investigation. A more comprehensive analysis of the changes in Pt NPs' contribution to ORR activity due to activation during the AST process is necessary. It would be valuable to examine the differences in chemical characteristics of Pt post-AST, as well as to assess performance trends following the blocking of Pt NPs in post-AST catalysts.
5. What is the rationale behind employing two distinct types of FeN₄ catalysts (Fe-N-CAero and Fe_{0.5}) in elucidating the interactions between FeN₄ sites and Pt NPs?
6. In the electrode preparation for Mössbauer spectra measurements presented in Fig. 6, the experimental design ensured equal loading of Fe-N-C, enabling quantitative comparison of the absolute amount of each Fe species. In this context, it is crucial to ascertain whether the electrode fabrication for Pt-deposited samples was adjusted to compensate for the platinum content. Specifically, was the amount of pure Fe-N-C catalyst (i.e., the Fe content) corrected by excluding the weight of platinum when preparing these electrodes?
7. The study appears to lack comprehensive verification of the local structure in Fe-N-C catalysts. It would be beneficial to address whether EXAFS curve fitting analysis was conducted to confirm the presence of FeN₄ active structures in the Fe-N-C catalysts. Such analysis could provide crucial insights into the atomic-level arrangement and coordination environment of iron atoms, thereby substantiating the assumed FeN₄ configuration.

Version 1:

Reviewer comments:

Reviewer #1

(Remarks to the Author)
See my attached PDF file.

Reviewer #2

(Remarks to the Author)
The authors have addressed most comments from the reviewers and I suggest it is accepted for publication.

Reviewer #3

(Remarks to the Author)
After the first review process, it seems like that authors tried to respond to reviewer's comment sincerely, The quality of the manuscript has been improved with a lot more evidence supporting their idea and strategy. Although author's main idea (the role of Pt NP is mainly focused on stabilizing Fe SA sites, rather than an primary active component in ORR) is somewhat arguable, but it would be very innovative and impactful to the general reader on this research area. Therefore, reviewer think it is publishable in Nature Communications.

Note: our answers are highlighted with **blue font**, and the new text added in the revised manuscript highlighted with **red font**

Response to reviewers' comments:

Reviewer #1

Durability of Fe-N-C cathode catalysts in PEFC is a key topic to be addressed for further commercialization of this technology while replacing precious Pt. As mentioned by the authors of the submitted manuscript the two main factors for the rather rapid degradation of Fe-N-C are demetallation of FeN₄ sites and a reactive oxygen species (ROS) degradation path. The degradation mitigation approach presented by the authors in this manuscript is a promising approach. Compared to previously reported Pt/Fe-N-C catalysts the authors combine the rather mild condition Pt nanoparticle (Pt NP) deposition via the soft polyol method with a low Pt amount of approx. 1 wt. % resulting in a *supposedly* uniform dispersion of well-defined Pt NP.

While this method shows promising results for three different Fe-N-C species, my main concern is the evidence for the enhanced durability of the Pt/Fe-N-C catalyst. The authors present a theoretical approach by calculation of the Pt NP to FeN₄ sites ratios for different Pt NP sizes and FeN₄ site densities and suggest that the enhanced durability is due to a long-range stabilization effect and not atomic distance between Pt NP and FeN₄ sites.

To be published in Nature Communications the authors should either provide some basic theoretical calculations, which support that one Pt NP can stabilize many FeN₄ sites by long-range interaction or suggest a model, what kind of interaction is at work here and how this interaction can be established over a length of tens of atoms and simultaneously with different FeN₄ sites.

Our answer: We thank the reviewer for her/his constructive generally positive comment on the work, for the in-depth evaluation and the detailed questions. To respond to this first and also most challenging question, which indeed puzzled ourselves as well, we performed a theoretical study to try to find a detailed explanation to our experimental observations. To allow screening a number of possible effects in the search of the most plausible explanation of the long-range effect, we studied models in which one Pt single atom is deposited atop a Fe-N-C layer model. A single Pt nanoparticle contains 100 or more of Pt atoms and would have required calculation capability that is beyond the scope of the present study. Moreover, the advantage of adding only one Pt single atom in the chosen Fe-N-C model is that the resulting Pt/Fe atomic ratio is comparable to those in the experimental materials.

The calculations were performed in collaboration with Tzonka Mineva and Hazar Guesmi, from ICGM - CNRS (who were added as new co-authors), and based on different Fe-N-C models that had been previously established by Tzonka Mineva in the frame of past combined experimental and theoretical research with Frederic Jaouen and others.

In summary, the calculations show that adding Pt single atoms above the graphene plane can stabilize FeN₄ sites (both FeN₄C₁₀ and FeN₄C₁₂, depending on the absence or presence of zig-zag carbon defects near the FeN₄ sites) by mitigating structural changes experienced by FeN₄ sites (especially the out-of-plane displacement) upon increased spin state and/or upon adsorption of OOH species atop the Fe center. The Pt atom becomes spin polarized when increasing the total spin of the structures, which

reduces the spin density at the Fe center and mitigates out-of-plane displacement and the Fe-N bond elongation.

Our action:

The following paragraph was added near the end of the manuscript:

Insights on the long-range electronic effect of Pt from computational chemistry

To better understand the long-range interaction between Pt NPs and FeN₄ sites, we resorted to density functional theory (DFT) calculations on periodic models of the FeN₄C₁₀ and FeN₄C₁₂ sites (**Supplementary Fig. 24**). We carried out geometry optimization of the model structures, with or without Pt on the graphene plane, and with or without OOH adsorbed on the Fe center. The effect of the total spin (S_{tot}) imposed on the periodic structures was also investigated. The unit cells comprise a single FeN₄ site and a single Pt atom, which is adsorbed on the graphene sheet at the longest possible distance from the Fe center (**Supplementary Fig. 24**). While different from the physicochemical description of the real catalysts (a Pt NP comprising > 100 atoms), such models are appropriate to investigate the long-range effect since the Pt-Fe distances (7-10 Å) are sufficiently long to avoid direct cooperative effects between Pt and Fe, and the Pt to Fe atomic ratio of 1 in the models is close to the ratio existing at the surface of the real catalysts. For example, with bulk contents of 1.5 wt. % Fe and 1 wt. % Pt and assuming 20 % of Fe is on the surface, the Pt/Fe surface atomic ratio is 0.9. For both the FeN₄C₁₀ and FeN₄C₁₂ models, the Pt atom adsorbs in a bridge position (**Supplementary Fig. 24**) and has a negligible effect on the Fe-N distances, except for the combination of the highest total spin ($S_{\text{tot}} = 3$) and the FeN₄C₁₀ model. In the latter case, the adsorption of Pt decreases the Fe-N distance by 0.057 Å (See **Supplementary Tables 11-12**), breaking the trend of increasing Fe-N distance with increasing S_{tot} that is observed without Pt (**Supplementary Table 12**). The latter trend is in accordance with increased Fe-N distance with increasing spin that was experimentally reported for Fe(II)N₄ porphyrins,⁵¹ with the maximum Fe-N increase of *ca.* 0.1 Å also quantitatively in line with our results (**Supplementary Table 12**). This agreement supports the validity of our calculations. Similar effects (or lack of) of Pt are observed for the models including an OOH intermediate adsorbed atop the Fe center (**Supplementary Fig. 24**). For the FeN₄C₁₂ model, the Pt adsorption has a negligible effect on the Fe-N and Fe-O distances (variation of 0.001-0.002 Å, see **Supplementary Tables 13-14**), for any of the imposed spin states. In contrast, the Pt adsorption leads to significant shortening of the Fe-N bond for the FeN₄C₁₀ model for $S_{\text{tot}} = 3/2$ (-0.023 Å) and $S_{\text{tot}} = 5/2$ (-0.126 Å). The Fe-N bond shortening upon Pt adsorption is partly due to the mitigated out-of-plane displacement of Fe above the N₄ plane when it adsorbs OOH (see **Supplementary Fig. 24** and **Supplementary Table 15**). For $S_{\text{tot}} = 5/2$, the out-of-plane Fe displacement is even reduced from 0.049 Å without Pt to only 0.020 Å with Pt (**Supplementary Table 15**). This can explain the improved stability of such an FeN₄ site during ORR. Regardless of the rate-determining step of the ORR on FeN₄ sites (in enzymes or bio-inspired catalytic sites), it is recognized that the Fe center undergoes changes in its oxidation and spin states as a result of the necessary oxygenation and deoxygenation events.^{49,52} Therefore, it can be concluded from DFT and for the FeN₄C₁₀ model that the presence of an adsorbed Pt atom situated far from the FeN₄ site stabilizes it by minimizing its structural changes experienced during the catalytic cycle. In contrast, for the considered FeN₄C₁₂ model, the Pt adsorption does not lead to shortening of the Fe-N bond. The different behavior is due to the presence of zig-zag carbon edge defects near the FeN₄ site for that model, as discussed later. While the FeN₄ structure is affected by the spin polarization of the distant Pt atom, the interaction energy between OOH and Fe is not strongly modified (See **Supplementary Fig. 25** and associated description).

To better understand the role of the adsorbed platinum atom in maintaining the geometric structure of FeN₄ upon OOH* adsorption for medium- and high-spin states in the FeN₄C₁₀ model and how this differs from the FeN₄C₁₂ structure, we computed the spin-density distribution. **Supplementary Fig. 26** compares the spin density at Fe, ρ_s , with the total spin, while **Supplementary Fig. 27** shows the spin-density distribution across all atoms in the (Pt)/FeN₄C_x models for $S_{\text{tot}} = 2$. The results indicate that

when Pt is adsorbed on FeN₄C₁₀, it becomes spin-polarized, mitigating the increase of the spin density at Fe with increasing S_{tot}. This explains the mitigated deformation of the FeN₄ site upon OOH* adsorption in the presence of Pt. In the chosen FeN₄C₁₂ model, the spin polarization of nearby zig-zag edge carbon atoms limits the spin density at the Fe site. This in turn explains the lack of effect of Pt for the selected FeN₄C₁₂ model. Recently, the spin polarization of carbon atoms (and limited spin at the Fe center) was calculated by us for FeN₄C₁₀ moieties when they are integrated in carbon nanoribbons and in close proximity with the zig-zag carbon edge.⁵³ Thus, one can expect that a FeN₄C₁₂ model free of nearby defects in the graphene plane (such as the 2f model in Fig. 1 of Ref¹⁸) would behave similarly as the defect-free FeN₄C₁₀ model and that adsorption of Pt would similarly stabilize it.

Overall, the DFT insights provide evidence for the long-range electronic effects at play between Pt NPs and various FeN₄ sites. The partial stabilization of the D1 fingerprint of FeN₄C₁₂ sites (Mössbauer spectroscopy) by Pt that is experimentally observed supports the existence of a plurality of D1 sites and, connected with the present DFT results, it is proposed that only the FeN₄C₁₀ and FeN₄C₁₂ sites far from zig-zag carbon edges can be stabilized by distant platinum. These results fall within but also expand the known magneto-structural correlations of FeN₄ square-planar complexes and ties with the recognized strong dependence of their metal-nitrogen bond strength on spin state.⁵⁴ On a broader perspective, it also connects with studies on the spin effect on oxygen electrocatalysis for various materials.⁵⁵⁻⁵⁸

The following figures and Tables were added in Supplementary information:

Supplementary figures 24-27, and supplementary Tables 11-15.

The Methods section was updated to describe the DFT methodology and periodic structures considered.

The following sentence was added in abstract:

Computational chemistry reveals that spin polarization of distant Pt atoms mitigates the structural changes of FeN₄ sites upon adsorption of oxygenated species atop Fe, especially in high-spin state

The following sentence was added in conclusions:

Computational chemistry reveals how distant platinum objects can stabilize the structure of FeN₄ moieties by spin polarization effects.

Furthermore, more experimental evidence for homogeneous FeN₄ site distribution in the Fe-N-C and Pt NP distribution between those sites would be helpful.

Our answer:

STEM-EDX images have now been acquired on Pt/Fe-N-C_{Aero}.

Our action:

The below sentence describing the STEM-X-EDS was added in the main text, and the figure below was added in the supplementary information as Supp. Fig. 4:

Scanning transmission electron microscopy measurements coupled with energy dispersive X-ray spectroscopy (STEM-X-EDS) confirmed the high dispersion of Pt NPs on Fe-N-C_{Aero} (**Supplementary Fig. 4**).

Supplementary Fig. 1 | Scanning transmission electron microscopy measurements of Pt/Fe-N-C_{Aero} coupled with energy dispersive X-ray spectroscopy.

Alternatively, experimental data for different Pt wt. % could be presented supposedly resulting in a volcano plot for the durability/mass activity, which could prove that 1 wt. % Pt load is the ideal situation.

Our answer: we thank the reviewer for this question. We have now performed a study on the effect of the Pt content on the Fe-N-C_{Aero} catalyst and investigated the effect on the initial and final ORR activity in PEMFC, before/after a 50h durability test in PEMFC.

The study was performed on a same batch of Fe-N-C_{Aero} onto which we deposited on the same day (and from the same Pt polyol suspension solution) different amounts of Pt, namely 1 wt. % Pt, 0.5 wt. % Pt and 0.25 wt. % Pt. The polarization curves before/after a 50h durability test at 0.5 V in PEMFC show that below 1 wt. % Pt, the amount of Pt is not enough to fully stabilize this Fe-N-C material, and the loss in activity/performance during the 50 h test increases with decreasing Pt content (see the Figure below, in the description of 'our action'). However, even with only 0.25 wt. % Pt, the rate of loss in activity/performance during the PEMFC test is much slower when compared to Fe-N-C_{Aero} free of Pt. Since for practical application, a durability of 50 h is still regarded as a short durability test, the results indicate that 1 wt. % Pt is an adequate Pt content to consider, for this Fe-N-C_{Aero} cathode catalyst. Lowering the Pt content while keeping full stabilization may be achieved with other Fe-N-C catalysts (the Fe-N-C_{Aero} involves a pyrolysis in dilute NH₃ gas, which typically leads to high surface basicity and

hence low stability in acid medium) that have higher FeN₄ site intrinsic stability, and we are currently working on such a concept. As the Fe-N-C material is subjected to a pH 3 solution during Pt deposition, it is important to keep in mind that even if Fe-N-C is stabilized by Pt addition, it needs to withstand the exposure to pH 3 during which there is no Pt onto the Fe-N-C in the initial stage.

Our action:

We added the following text in the main manuscript:

We then studied the sensitivity of the stabilization effect on the amount of Pt added, with Pt contents of 0.25, 0.50 wt. % Pt on Fe-N-C_{Aero}. The results of 50h PEMFC tests at 0.5 V show that full stabilization is reached with 1 wt. % Pt while a performance loss is observed for lower Pt contents, the loss increasing with decreasing Pt content (**Supplementary Fig. 17**). Thus, for Pt/Fe-N-C_{Aero}, the lowest Pt amount needed for efficient stabilization is *ca.* 1 wt. %. This threshold Pt amount may differ across Fe-N-C materials.

And we added the following figure in supplementary information:

Supplementary Fig. 2 | Polarization curves in PEMFC device under an operating temperature of 80 °C comparing BoT and end-of-test (EoT, at $U = 0.5$ V for 50h) of **a)** Pt/Fe-N-C_{Aero}, **b)** Pt_{0.50}/Fe-N-C_{Aero} and **c)** Pt_{0.25}/Fe-N-C_{Aero}, comprising 1.0, 0.50 and 0.25 wt. % Pt, respectively Note that the BoT performance of Pt/Fe-N-C_{Aero} here is slightly lower than in all other graphs because the Pt deposition to reach 1.0, 0.50 and 0.25 wt. % Pt was made at the same time to avoid any biases, and the Fe-N-C_{Aero} batch had aged slightly during *ca.* 9 months shelf storage in air relative to all other PEMFC experiments reporting Fe-N-C_{Aero} or Pt/Fe-N-C_{Aero} data.

Regarding these two points I would recommend a revision before the manuscript can be published in Nature Communications.

Our answer:

We agree that the above-mentioned points were major points needing strengthening in the submitted work. Through DFT computation and additional experimental characterization we hope that we have provided the kind of additional support to our initial analyses that the reviewer expected.

The provided data support the conclusions of the authors but see my comment above.

The work meets the expected standard, and enough details are provided for the work to be reproduced.

Specific questions to the authors:

L56: What is the proposed mechanism for demetallation under inert gas since ROS are involved?

Our answer: In the present work, the AST-1 to AST-4 are indeed performed in Ar-saturated electrolyte, while the corresponding AST-5 to AST-8 are performed in O₂-saturated electrolyte. A low decrease in ORR activity is observed for AST-1 and AST-2 (performed at 25 °C) on Fe-N-C_{Aero}, only circa 20 % activity loss after 10,000 cycles. A stronger ORR activity loss of circa 45-60 % is observed on Fe-N-C_{Aero} after AST-3 and AST-4, following the same 10,000 cycles but this time at 80 °C. The deactivation mechanism at play during such AST is thus not assigned to ROS as no ORR occurs in those conditions. The ORR activity loss during cycling in such condition has been generally assigned to the demetallation from acid-unstable Fe-N₄ sites. This is also what we believe happens to the Fe-N-C_{Aero}, especially as the last synthesis step of this material includes a heat-treatment at 950 °C in 10 % NH₃ and 90 % N₂. Ammonia pyrolysis is known to increase the activity of Fe-N-C catalysts via increasing their surface basicity (highly basic N-groups formed), but resulting in compromised FeN₄ site stability in acidic medium. In addition, when changing the upper potential limit from 0.92 to 1 V (AST 3 and 4, respectively), and when doing such AST at high temperature, the impact of carbon corrosion and ensuing demetallation from FeN₄ sites as a result of neighboring carbon corrosion is likely the reason for increased activity loss observed on Fe-N-C_{Aero} after AST-4 vs AST-3.

Our action: we added the following description of the degradation mechanisms taking place during AST 1-4:

ASTs under Ar atmosphere lead to only *ca.* 20 % activity loss at 25 °C (ASTs 1-2) and to *ca.* 45-60 % activity loss at 80 °C (ASTs 3-4). The main deactivation mechanisms at play in such conditions are direct demetallation from acid-unstable FeN₄ sites and indirect demetallation as a result of carbon corrosion, especially for AST-4 combining high temperature and the high UPL value of 1.0 V. The presence of a fraction of intrinsically acid-unstable FeN₄ sites in Fe-N-C_{Aero} is logical due to its last synthesis step which includes a heat-treatment at 950 °C in 10 % NH₃ and 90 % N₂. Ammonia pyrolysis is known to increase the activity of Fe-N-C catalysts *via* increasing their surface basicity (highly basic N-groups formed), but resulting in compromised FeN₄ site stability in acidic medium.³⁵

L112/113: What was the number of particles used for particle size evaluation and what is the error in particle size?

Our answer:

We appreciate the reviewer's question and fully agree that the error in particle size is essential to ensure data reliability. The standard deviation has now been included, and the resulting values are: 1.69 ± 0.0512 nm for Pt/N-C_{Aero}; 1.65 ± 0.0472 nm for Pt/Fe-N-C_{Aero}; and 2.16 ± 0.0576 nm for Pt/C TKK. Given the very small particles size observed (~2 nm), only clearly visible particles were considered for the calculation, avoiding visible agglomerates and/or particles with poor contrast against the background. The number of particles used for size evaluation was 81 for Pt/N-C_{Aero}; 65 for Pt/Fe-N-C_{Aero}; and 35 for the commercial Pt/C from TKK.

To further confirm the reliability of this data, STEM-X-EDS measurements were also conducted for Pt/Fe-N-C_{Aero}, yielding a particle size of 1.64 ± 0.383 nm. This value is in excellent agreement with the previously determined value of 1.65 ± 0.0472 nm obtained through TEM analysis.

Our action:

The Supplementary Fig. 5 (Supplementary Fig. 2 in original version) has been modified to include particle size error values, as reproduced below.

Additionally, the new STEM-X-EDS results on Pt/Fe-N-C_{Aero} were included in the manuscript as Supplementary Fig. 4 and the below sentence describing the STEM-X-EDS was added in the main text:

Scanning transmission electron microscopy measurements coupled with energy dispersive X-ray spectroscopy (STEM-X-EDS) confirmed the high dispersion of Pt NPs on Fe-N-C_{Aero} (**Supplementary Fig. 4**).

L161: What was the reason for using H₂SO₄ instead of HClO₄ for the wet-electrochemical experiments (RDE/RRDE)? HClO₄ is considered as having a similar acidity compared to the sulfonate groups in Nafion and therefore allows to better compare results obtained by RDE with those obtained by PEFC experiments.

Our answer:

We certainly agree with the reviewer that when studying activity of ORR electrocatalysts whose dominant activity comes from crystalline Pt species, the use of HClO₄ electrolytes is recommended, to

avoid the strong adsorption by bisulfate anions which happens in RDE setup but not in PEMFC. The ORR activity for Pt-based catalysts is dramatically higher in HClO₄ than in H₂SO₄ electrolyte.

When studying Fe-N-C catalysts, a generally weak effect has been observed when comparing H₂SO₄ and HClO₄ electrolytes (<https://www.sciencedirect.com/science/article/abs/pii/S0926337318302558> ; <https://www.sciencedirect.com/science/article/abs/pii/S0013468623001214> ; <https://pubs.acs.org/doi/abs/10.1021/acscatal.8b01584> ; <https://pubs.acs.org/doi/10.1021/acscatal.8b04609>). In several cases, it was found that the ORR activity is slightly higher for a given Fe-N-C in H₂SO₄ than in HClO₄ electrolyte, with activity increase in the range of +31 to +70 % depending on individual catalysts (<https://www.sciencedirect.com/science/article/abs/pii/S0013468623001214> ; <https://www.sciencedirect.com/science/article/abs/pii/S0926337318302558>). Across a broader range of electrolytes, it was found for Fe-N-C catalysts that the activity trend increases in the order H₃PO₄ > H₂SO₄ > HClO₄ > HCl, believed to be related to the adsorption strength of the anions on the Fe cation site, in the order H₂PO₄⁻ < HSO₄⁻ < ClO₄⁻ < Cl⁻.

For studying Pt/Fe-N-C catalysts, there is thus no perfect acid electrolyte that would involve anions that bind most weakly to both Pt NPs and FeN₄ sites. Sulfuric acid is on the strong binding side for Pt crystalline objects but on the weak binding side for FeN₄ sites, while perchloric acid is on the weak binding side for Pt crystalline objects but more strongly binding to FeN₄ sites. Phosphoric acid is perhaps best for Fe-N-C, but binds very strongly to Pt.

What is nevertheless sure is that the use of H₂SO₄ vs. HClO₄ electrolyte for studying Pt/Fe-N-C will potentially decrease the relative contribution from Pt NPs. The reported difference in Pt ORR activity from H₂SO₄ vs. HClO₄ electrolyte (all other experimental conditions kept the same) is an increase of ca x1.8 to x2.7 depending on potential (0.9 or 0.85 V, respectively) (see Table 2, rows highlighted as 'this work' in U.A. Paulus et al, J. Electroanal. Chem. 495 (2001) 134-145). The ORR activity increase from H₂SO₄ vs. HClO₄ electrolyte was, in that work, even higher at 0.8 V, more than 3x (Fig. 8 in U.A. Paulus et al, J. Electroanal. Chem. 495 (2001) 134-145).

Our action:

We conducted beginning-of-life ORR activity tests in RDE in both 0.1 M H₂SO₄ and HClO₄ electrolyte for Pt/Fe-N-C_{Aero} to see what differences are observed and if the results can shed light on the ORR activity contribution from Pt nanoparticles. The mass activity at 0.8 V (or 0.9 V) was found identical (within error bar) in HClO₄ and H₂SO₄, as shown in **Fig. R1** (Tafel plot). This strongly suggests no or small contribution of Pt NPs to the overall ORR activity of the Pt/Fe-N-C material.

Fig. R1 | Electrochemical evaluation of Pt/Fe-N-C_{Aero} catalyst in different acidic environments: oxygen reduction reaction polarization curve measured in O₂-saturated electrolyte at $v = 2 \text{ mV}\cdot\text{s}^{-1}$ and $w = 1600 \text{ rpm}$, $T = 25 \text{ }^\circ\text{C}$. Catalyst loading of $400 \mu\text{g cm}^{-2}_{\text{geo}}$.

L173: Usually, a voltage of 0.9 V vs. RHE is chosen for mass activity. Why 0.8 V?

Our answer: The voltage of 0.8 V is the typical voltage (potential) used to report ORR activity of Fe-N-C catalysts in acid medium, because the initial activity is barely measurable at 0.9 V, especially after accelerated stress tests. The chosen potential of 0.8 V is thus practical to report the ORR activity, both before and after ASTs, of Fe-N-C and present Pt/Fe-N-C (with low Pt content, leading to no or low increase in activity relative to Fe-N-C).

L179: The authors should also calculate mass activity in $\text{A g}^{-1}_{\text{Pt}}$ for better comparison.

Our answer:

This type of normalization is a standard procedure for Pt-based catalysts and we thank the reviewer for raising this point. In our work, the hybrid catalysts contain an ultra-low amount of Pt NPs, approximately 1 wt. %, and numerous observations in the present work indicate that these Pt NPs are poorly ORR-active, or not ORR-active at all, and that the Fe-based sites mostly contribute to the ORR activity of the present Pt/Fe-N-C catalysts. Under these circumstances, comparing mass activity normalized by Pt ($\text{A g}^{-1}_{\text{Pt}}$) with Pt-based catalysts containing high noble metal loadings (*e.g.* > 20 wt. % Pt/C), where Pt NPs are indeed the primary (or sole) active sites, will not necessarily lead to meaningful values, as the apparent (measured activity) mainly does not come from Pt. On the other hand, this mathematical exercise can tell us something: if the ORR activity of Pt/Fe-N-C does mainly come from the Pt particles, the activity normalized by the Pt mass should be comparable to the well-known activity of Pt nanoparticles of similar size, supported on carbon (or N-doped) carbon supports.

Starting from the mass activity (MA) at 0.8 V reported in Fig. 2c of the original version of this manuscript (Fe-N-C_{Aero} and Pt/Fe-N-C_{Aero} exhibiting values of 2.71 and 3.95 $\text{A g}^{-1}_{\text{powder}}$, respectively, and Pt/N-C_{Aero} a MA value of 1.19 $\text{A g}^{-1}_{\text{powder}}$), the MA numbers at 0.8 V vs RHE can simply be multiplied by 100 to convert them from $\text{A g}^{-1}_{\text{powder}}$ to $\text{A g}^{-1}_{\text{Pt}}$, since the Pt loading is ca 1 wt. % Pt in both Pt/Fe-N-C_{Aero} and Pt/N-C_{Aero} materials. This leads to MA numbers of 119 $\text{A g}^{-1}_{\text{Pt}}$ for Pt/N-C_{Aero} and 395 $\text{A g}^{-1}_{\text{Pt}}$ for Pt/Fe-N-C_{Aero}, at 0.8 V vs. RHE. However, as the MA of ORR-active Pt is usually reported at 0.9 V vs RHE, these numbers must be converted to the expected MA at 0.9 V, and this was done assuming a Tafel slope of 75 mV/decade (the experimental Tafel slope observed for Pt/Fe-N-C_{Aero}). According to Tafel law and classical math, this leads to a conversion factor of / 21.45 to convert MA values from 0.8 V to

0.9 V, i.e. $5.5 \text{ A g}^{-1}_{\text{Pt}}$ for Pt/N-C_{Aero} at 0.9 V vs. RHE and $18.4 \text{ A g}^{-1}_{\text{Pt}}$ for Pt/Fe-N-C_{Aero} at 0.9 V vs. RHE. For the latter, one however cannot assign the whole activity of Pt/Fe-N-C_{Aero} to Pt, as the activity of Fe-N-C_{Aero} is comparable. Assuming the difference in MA values between Pt/Fe-N-C_{Aero} and Fe-N-C_{Aero} is representative of the Pt contribution towards ORR activity, ($3.95 - 2.71 = 1.24 \text{ A g}^{-1}_{\text{powder}}$), one can then calculate a possible MA contribution of Pt present in Pt/Fe-N-C_{Aero} to be ca $124 \text{ A g}^{-1}_{\text{Pt}}$ at 0.8 V vs. RHE, corresponding to $124/21.45 = 5.8 \text{ A g}^{-1}_{\text{Pt}}$ at 0.9 V vs RHE. Interestingly, this number is close to the MA of $5.5 \text{ A g}^{-1}_{\text{Pt}}$ estimated above for Pt/N-C_{Aero} at 0.9 V vs. RHE, indicating that this might be representative of the Pt specific activity in both these materials. If this estimation is correct (numerous hypotheses and neglecting the fact that adding Pt to Fe-N-C modifies the overall ORR selectivity, etc), it tells that the Pt particles present on N-C_{Aero} or Fe-N-C_{Aero} have a lower ORR activity than Pt particles in commercial Pt/C catalysts, where the Pt contents are typically > 20 wt. % Pt and the reported MA at 0.9 V vs RHE are typically $\geq 100 \text{ A g}^{-1}_{\text{Pt}}$.

Our action:

We added the following text in the main manuscript:

It is nevertheless of interest to assess the Pt mass activity of these composites. Assigning the ORR activity of Pt/N-C_{Aero} solely to Pt leads to a MA_{@0.9V} of $5.5 \text{ A g}^{-1}_{\text{Pt}}$ (see **Supplementary Note 1**). For Pt/Fe-N-C_{Aero}, one may assume the Pt contribution to be the difference ($3.95 - 2.71$) $\text{A g}^{-1}_{\text{powder}}$, leading to an estimated MA_{@0.9V} of $5.8 \text{ A g}^{-1}_{\text{Pt}}$ (0.9 V is used here to ease the comparison to literature of Pt activity). These numbers are comparable, and much lower than MA_{@0.9V} of state-of-art Pt/C, with values of $100 - 200 \text{ A g}^{-1}_{\text{Pt}}$.^{3,40} The low MA_{@0.9V} for Pt in the present composite catalysts (with 1 wt. % Pt) may be due to the localization of the Pt particles in the micropores of Fe-N-C.

And we added the supplementary note 1 to explain the way the Pt activity was estimated:

Supplementary Note 1 | Estimation of the Pt specific contribution to ORR activity in Pt/N-C_{Aero} and Pt/Fe-N-C_{Aero}

Starting from the mass activity (MA) at 0.8 V (Pt/N-C_{Aero} has a MA value of $1.19 \text{ A g}^{-1}_{\text{powder}}$), the MA numbers at 0.8 V vs RHE can simply be multiplied by 100 to convert them from $\text{A g}^{-1}_{\text{powder}}$ to $\text{A g}^{-1}_{\text{Pt}}$, since the Pt loading is ca 1 wt. % Pt. This leads to MA numbers of $119 \text{ A g}^{-1}_{\text{Pt}}$ for Pt/N-C_{Aero}, at 0.8 V vs. RHE. However, as the MA of ORR-active Pt is usually reported at 0.9 V vs RHE, it is more meaningful to convert this number to the expected MA at 0.9 V, and this was done assuming a Tafel slope of 75 mV/decade (which is also the experimental Tafel slope observed). According to Tafel law and classical math, this leads to a conversion factor of / 21.45 to convert MA values from 0.8 V to 0.9 V, i.e. $5.5 \text{ A g}^{-1}_{\text{Pt}}$ for Pt/N-C_{Aero} at 0.9 V vs. RHE

For Pt in Pt/Fe-N-C_{Aero}, assuming the difference in MA values between Pt/Fe-N-C_{Aero} and Fe-N-C_{Aero} is representative of the Pt contribution towards ORR activity, ($3.95 - 2.71 = 1.24 \text{ A g}^{-1}_{\text{powder}}$, at 0.8 V), one can then calculate a possible MA contribution of Pt present in Pt/Fe-N-C_{Aero} to be ca $124 \text{ A g}^{-1}_{\text{Pt}}$ at 0.8 V vs. RHE, corresponding to $124/21.45 = 5.8 \text{ A g}^{-1}_{\text{Pt}}$ at 0.9 V vs RHE.

L184: The reason to compare the performance to the mathematical sum of the individual performances of Fe-N-C_{aero} and Pt/N-C_{aero} is to show the synergetic effect? The authors should clarify this.

Our answer:

We thank the reviewer and agree that the reason needs to be better clarified. The comparison with the mathematical sum of the individual performances of Fe–N–C_{Aero} and Pt/N–C_{Aero} in the kinetic region of the ORR polarization curve was indeed conducted to highlight the synergetic effect within the hybrid Pt/Fe–N–C_{Aero} catalyst. This approach aimed to confirm that the combined performance of the FeN₄ sites and Pt nanoparticles in the synthesized material exceeds the sum of their separate contributions. Such a result suggests that the synergy arises not merely from an additive effect but rather from enhanced kinetics or improved selectivity (*e.g.*, favoring the 4-electron ORR pathway) due to the interaction between Pt nanoparticles and FeN₄ sites. We have clarified this by adding new text in the main MS.

Our action:

We expanded the existing L-184- 188 lines in the main original document by the following:

...further supports a synergy between Pt NPs and FeN₄ sites within this hybrid catalyst (**Supplementary Fig. 10**). The experimental curve for Pt/Fe–N–C_{Aero} shows indeed higher ORR activity than the curve obtained from the mathematical sum of the curves of Fe–N–C_{Aero} and Pt/N–C_{Aero}, showing that the activity observed for Pt/Fe–N–C_{Aero} is more than the mere sum of its FeN₄ sites and its Pt particles. The effect can hardly be explained by a simple change in Pt particle size between Pt/Fe–N–C_{Aero} and Pt/N–C_{Aero}, as the average size of Pt nanoparticles is similar (1.65 ± 0.05 and 1.69 ± 0.05 nm, see **Supplementary Fig. 5**).

L272: The calculation of the theoretical diffusion-limited current density should be shown in supplemental notes

Our answer: the line 272 relates to the discussion of the expected diffusion limited current density in the experiment of H₂O₂ electroreduction. The calculation of the theoretical diffusion limited current density of that experiment has been added as Supplemental Note 2 in the revised manuscript.

Our action:

We have added the below supplemental note

Supplementary Note 2 | Calculation of the diffusion-limited current density during HPRR measurement

The theoretical diffusion-limited current density expected during RDE measurement looking at the two-electron electro-reduction activity of H₂O₂ to H₂O is calculated according to Levich equation

$$j_L = (0.620) \times n \times F \times D^{\frac{2}{3}} \times \omega^{\frac{1}{2}} \times \nu^{(-\frac{1}{6})} \times C$$

Where j_L is in A·cm⁻², n is the number of electrons exchanged for each reactant molecule ($n = 2$ for H₂O₂ to H₂O), F is Faraday's constant (96485.3 C·mol⁻¹), D is the H₂O₂ diffusion coefficient in the electrolyte (1.3·10⁻⁵ cm²·s⁻¹),¹ ω is the angular rotation rate (167.55 rad·s⁻¹, corresponding to 1600 rpm), ν is the kinematic viscosity of the 0.1 M H₂SO₄ electrolyte (10⁻² cm²·s⁻¹)¹ and c is the H₂O₂ bulk concentration in the electrolyte (10⁻⁵ mol·cm⁻³, corresponding to 10 mM). Substituting these values in the above equation gives $j_L = 0.01845$ A cm⁻², or 18.45 mA cm⁻². This is the current density at the diffusional limit that should be obtained, if the electrode were being limited solely by the diffusion of H₂O₂.

L297: The TEM images are slightly too dark.

Our answer: We agree that in the Supplementary Fig. 12a in the original version the images were too dark and the contrast scale not optimized.

Our action: The contrast of the same TEM images has been modified and the modified figure in the revised version of the supplementary information is shown below.

Supplementary Fig. 3 | TEM micrographs of Fe-N-C_{Aero} **(a)** after acid exposure, **(b)** at EoL in low-density Fe (LD-Fe) region, **(c)** at EoL in high-density Fe (HD-Fe) region, and **(d)** Pt/Fe-N-C_{Aero} at EoL.

L338: Is the Pt NP size comparable to Pt/Fe-N-C?

Our answer:

We agree with the reviewer that this is an important information for a reliable comparison between the catalysts. The Pt particle size we estimated for Pt/Fe-N-C_{Paj} was 1.96 ± 0.0973 nm. Therefore, the platinumized hybrid catalysts Pt/Fe-N-C_{Paj} and Pt/Fe-N-C_{Aero} (1.65 ± 0.0472 nm) have a comparable average size of Pt NPs.

Our action:

We have added this information to the caption of the supplementary figure 18 showing the TEM image of Pt/Fe-N-C_{Paj}, with the modified figure caption as reproduced below:

Supplementary Fig. 18 | TEM images of **(a)** Fe-N-C_{Paj} and **(b)** Pt/Fe-N-C_{Paj}. PEMFC experiments under an operating temperature of 80 °C comparing Fe-N-C_{Paj} and Pt/Fe-N-C_{Paj} **(c)** Polarization curves

obtained at the BoT and EoT, and **(d)** chronoamperometries measured at $U = 0.5$ V. The average Pt particle size estimated for Pt/Fe–N–C_{Paj} from **(b)** was 1.96 ± 0.0973 nm.

We also added the following sentence in the main text at the 'position' corresponding to L338 in the original version:

The average Pt particle size in Pt/Fe–N–C_{Paj} was 1.96 nm, slightly higher but comparable to the one measured in Pt/Fe–N–C_{Aero} (1.65 nm). The higher Pt particle size of Pt/Fe–N–C_{Paj} may be assigned to its lower BET area ($755 \text{ m}^2\text{g}^{-1}$) compared to Pt/Fe–N–C_{Aero} ($1191 \text{ m}^2\text{g}^{-1}$).

L357: Why the change in electrolyte for this specific experiment? The authors should give a reason

Our answer:

Indeed, different acid electrolytes were used for the electrochemistry tests: (i) HClO₄ is used for GDE-ICP-MS and (ii) H₂SO₄ is used for RDE & RRDE tests. This difference is due to the specificity of the ICP-MS technique.

In ICP-MS, the plasma source consists of argon (Ar) and is used to decompose the sample, which was previously introduced in liquid form prior to analysis. The sample will be atomized and ionized under Ar plasma. The ions thus formed are then detected on the basis of their mass-to-charge ratio (m/z) using a quadrupole analyzer.

In order to calibrate the signal of Fe ($m/z=56$) during long-term measurements, an internal standard (Ge, $m/z=74$) is used. The sulfur from H₂SO₄ and Ar from the plasma may form polyatomic ions $40\text{Ar}^{34}\text{S}^+$ ($m/z=74$) and/or $38\text{Ar}^{36}\text{S}^+$ ($m/z=74$).^[1] The polyatomic ions may thus interfere with the 74 Ge^+ signal. Hence, for the online S-GDE-ICP-MS measurements, instead of 0.1 M H₂SO₄, 0.1 M HClO₄ was chosen as the electrolyte.

For FeNC, Fe dissolution has been reported as more pronounced in 0.1 M HClO₄ than in 0.1 M H₂SO₄.^[2] So the online Fe dissolution measurements were conducted in a harsher condition for iron stability, namely in 0.1 M HClO₄, than in 0.1 M H₂SO₄.

References:

[1] May TW, Wiedmeyer RH. A table of polyatomic interferences in ICP-MS. ATOMIC SPECTROSCOPY-NORWALK CONNECTICUT-. 1998 Sep 1;19:150-5.

[2] Pedersen A, Kumar K, Ku YP, Martin V, Dubau L, Santos KT, Barrio J, Saveleva V, Glatzel P, Paidi V, Li X. Operando Fe Dissolution in Fe-NC Electrocatalysts during Acidic Oxygen Reduction and Impact of Local pH Change. Energy Environ. Sci., 2024,17, 6323-6337

Our action:

The following text was added in the Methods section regarding ICP-MS online measurements:

Here, 0.1 M HClO₄ was chosen as the electrolyte, instead of 0.1 M H₂SO₄ used for R(R)DE measurements, in order to keep the $^{74}\text{Ge}^+$ signal from interfering with the noises contributed by $^{40}\text{Ar}^{34}\text{S}^+$ or $^{38}\text{Ar}^{36}\text{S}^+$ (with same m/z value of 74 as $^{74}\text{Ge}^+$) that form if the H₂SO₄ electrolyte had been chosen.

L474: Exact data about the amounts required to ensure reproducibility

Our answer:

we agree that the synthesis description lacked details and reading the previous publications on this synthesis is unpractical if readers would like to reproduce the synthesis.

Our action:

We have completed the synthesis description, as highlighted below in red.

An Fe–N–C aerogel catalyst was synthesized through a one-pot method.⁴⁹ In summary, a hydrogel composed of optimized quantities of resorcinol (99 %, Alfa Aesar), formaldehyde (37 wt % solution, Acros Organics), melamine (99 %, Acros Organics), and anhydrous FeCl₃ (98 %, Acros Organics) was first synthesized by a sol-gel process. Typical quantities used to synthesize Fe–N–C_{Aero} were 3.6 g of resorcinol, 4.0 g of melamine (leading to a molar ratio of resorcinol/melamine of 2/2), 12 mL of formaldehyde and 0.11 g of FeCl₃. This was followed by...

L510: Electric circuit?

Our answer:

We agree that the description in the Methods section of the dual reference electrode system was unclear. In addition, the connection is in parallel, not in series as was erroneously described in the original version. The setup includes two reference electrodes, one is the RHE from Gaskatel and the other is a Pt wire immersed in electrolyte. This Pt wire connected to the RHE reference electrode was used to filter the high frequency electrical noise of the electrical network and to avoid disturbing the low frequency electrochemical measurements (it acts as a high-pass filter). More details on the dual-reference system used in this work can be found in Ref. C.C. Herrmann, G.G. Perrault, A.A. Pilla, Anal. Chem., 40 (1968) 1173-1174.

Our action:

We modified the sentence in Methods section and added further description of the setup, as highlighted in red below:

A Pt-wire immersed in the electrolyte was also connected to the above-mentioned RHE reference electrode in parallel through a capacitor. This is used to filter the high frequency electrical noise of the electrical network and to avoid disturbing the low frequency electrochemical measurements (it acts as a high-pass filter). More details on the dual-reference system used in this work can be found in Ref.⁶⁰.

L522: What is the calculated Pt load in g cm⁻²_{geo}

Our answer:

The same ink formulation was used for Fe-N-C and Pt/Fe-N-C materials thus the total mass of catalyst was 400 and 100 μg_{powder} cm⁻² geometric area for RDE and RRDE measurements, respectively. If the Pt content in the total mass of Pt/FeNC hybrids is exactly 1 wt.%, then the calculated Pt loadings are 4.0 and 1.0 μg_{Pt} cm⁻² for RDE and RRDE measurements, respectively.

Our action:

The following paragraph was added in the Methods:

Since the same ink formulation is used for Fe-N-C and Pt/Fe-N-C materials, the same total mass of catalyst powder is deposited but in the case of Pt/Fe-N-C, the total mass comprises the Pt mass. If the

Pt content in the total mass of a given Pt/Fe-N-C is 1.0 wt. %, then the calculated Pt geometric loadings are 4.0 and 1.0 $\mu\text{g}_{\text{Pt}} \text{cm}^{-2}_{\text{geo}}$ for RDE and RRDE measurements, respectively.

L579: JEOL JEM2100 series TEM?

Our answer:

The complete description of the instrument is JEOL JEM-2010. We confirm it is NOT a JEOL 2100 instrument.

Our action:

The sentence in Methods section was modified as highlighted in red below:

TEM coupled to X-EDS **was** carried out with a JEOL **JEM-2010 series** TEM instrument operated at 200 kV with a resolution of 0.19 nm.

Reviewer #2

The submitted manuscript reports the stabilization mechanism of FeN₄ sites by Pt particles for oxygen reduction reaction in the acid medium. The characterization and electrochemical tests of catalysts are defined well, and essential presentation is also provided for them. Particularly, it is impressive that operando techniques are used combined with post mortem Mössbauer spectroscopy analysis. However, this manuscript might not be suitable for publication in Nature Communications due to some weaknesses described below.

1. The atomic-scale Pt-FeN₄ interaction has been well reported in the literature, and here the authors demonstrate the stabilization effect of the long-distance electronic effect. So, what's the contribution from each?

Our answer:

We appreciate the highlights of the strengths but also the perceived weaknesses of the present study by the reviewer.

Regarding the first comment above, while we cannot exclude that a minor fraction of Fe-N₄ sites are at atomic distance of a Pt nanoparticle in the materials studied in the present study, the calculations reported in Supplementary Figure 17 of the original version of the manuscript clearly show that the ratio of number of Pt nanoparticles to surface Fe-N₄ sites is $\ll 1$ (about 1/150, at 1 wt. % Pt and with Pt particle size of 1.7 nm and for the site density of 3×10^{19} FeN₄ sites g⁻¹, the latter corresponding to the case of the Fe_{0.5} material), meaning that the overwhelming majority of Fe-N₄ sites cannot be in close contact (atomic distance) with a Pt nanoparticle. Hence, we can assign the stabilization effect of Fe-N₄ sites observed in the presently studied hybrid materials to the long-distance electronic effect. As requested by another reviewer (first question of reviewer 1), theoretical calculations were performed to explore this and to propose a rational explanation for this long-distance effect. We invite reviewer 2 to read the summary and actions performed regarding DFT on long-distance effect in our reply to the first question of reviewer 1. As an additional experimental proof that there are no Fe-Pt bonds, we carried out EXAFS measurements at the Fe K-edge and the detailed analysis proves the absence of Fe-Pt bonds. The EXAFS insights are detailed on the next page.

1, continued. There is a lack of experimental evidence showing the contact of Pt nanoparticles to FeN₄ sites or the Fe-Pt coordination. The authors create a complicated structure, but not uniform with high and low Fe zones, preventing the fundamental understanding of Pt-FeN₄ sites.

Our answer:

We here want to stress that the zones of high and low density of Fe were observed only AFTER the Fe-N-C_{Aero} material (without platinum) was subjected to long-term electrochemical testing (supplementary Table 6 of the original submission, EoL column, Fe-N-C_{Aero} row). EoL means End-of-Life, i.e. after AST-7 (i.e. after 10 000 electrochemical cycles between 0.60 and 0.92 V, O₂-saturated solution of 0.1 M H₂SO₄, 80 °C).

We quote below a section of text from the originally submitted version of the manuscript:

“This suggests the transformation during AST-7 on Fe-N-C_{Aero} of a substantial fraction of FeN₄ sites into Fe clusters (likely FeO_x particles) leading to high density zones,⁴⁴ consistent with its reduced ORR activity after AST-7. In contrast, no high-density Fe zones were observed on Pt/Fe-N-C_{Aero} at EoL, indicating reduced demetallation of Fe from FeN₄ sites, in line with its doubled ORR activity at EoL (Fig. 3).”

We think the sentence/paragraph quoted above was clear and free of ambiguity. In contrast, the format of supplementary Table 6 in original version may have led to the misunderstanding, where the “N/A” input might have been interpreted differently from our intention.

Regarding the first part of the comment (“lack of experimental evidence showing the contact of Pt nanoparticles to FeN₄ sites or the Fe-Pt coordination”), all the experimental characterizations we have done (in particular high energy-resolution fluorescence-detected X-ray absorption near edge structure ; X-ray emission spectroscopy ; X-ray diffraction) concur and show the absence of Fe-Pt coordination and there is no evidence for a direct physical contact between FeN₄ sites and Pt nanoparticles, or if such a contact exist, it is an exception and the majority of FeN₄ sites are not at atomic distance of a Pt nanoparticle (see also our answer to question 1 of this reviewer).

The experimental proofs for the absence of contact between the majority of FeN₄ sites and a Pt nanoparticle is actually central to the paper, and the reason why a long-distance electronic stabilization of many Fe-N₄ sites by a single Pt nanoparticle seems the likeliest explanation for the experimental observations. Computational chemistry calculations have been performed to better support this idea (see our answer on this point to Question 1 of Reviewer 1).

Last but not least we conducted new Fe K-edge EXAFS measurement on Fe-N-C_{Aero} and Pt/Fe-N-C_{Aero}, and fitted carefully the spectra to identify more precisely the Fe coordination environment in these two materials. The results (shown in Supp. Fig. 3 and Supp. Table 3) independently demonstrate the absence (or below detection limit) of close interaction between Fe and Pt atoms (up to a distance of ca 4 Å, a distance at which Fe-Pt backscattering signal would be detected by FT-EXAFS), implying that all (or the vast majority of) Fe-N₄ active sites are not located close enough to platinum particles to allow a participative ORR mechanism directly involving both Fe-N₄ and Pt sites. The Fe-N distance and coordination number are also highly similar in Fe-N-C_{Aero} and Pt/Fe-N-C_{Aero} (Supp. Table 3) further supporting that the FeN₄ site structure is largely unmodified and that the change in selectivity/stability is due to a long -range effect of Pt nanoparticles (this long-range effect was investigated by DFT, see the added section towards the end of the revised manuscript).

Our action:

We have split the supplementary Table 6 (in original version) in two tables, Table 8 for Fe–N–C_{Aero} (high- and low-density Fe zones observed) and Table 9 for Pt/ Fe–N–C_{Aero} (no high-density Fe zone). The reformatted supporting Tables 8-9 are shown below:

Supplementary Table 8 / Fe content measured by energy-dispersive X-ray spectroscopy at different stages of AST-7 for Fe–N–C_{Aero}. ‘Fresh’ represents the pristine powder; ‘Acid exposure’ represents the electrode (RDE tip with the catalyst ink deposited) after immersion for 17 h in 0.1 M H₂SO₄ at room temperature. For EoL, different zones were observed on the sample and LD-Fe and HD-Fe refer to low-density and high-density Fe regions, respectively.

Catalyst	Atomic percentage				
	Metal	Fresh	Acid exposure	EoL	
				LD-Fe	HD-Fe
Fe-N-C _{Aero}	Fe	0.25±0.04	0.09±0.02	0.04±0.02	1.10±0.90
	Pt	0.21±0.19	0.10±0.06	0.16±0.12	N/A

Supplementary Table 9 | Fe content measured by energy-dispersive X-ray spectroscopy at different stages of AST-7 for Pt/Fe–N–C_{Aero}. ‘Fresh’ represents the pristine powder; ‘Acid exposure’ represents the electrode (RDE tip with the catalyst ink deposited) after immersion for 17 h in 0.1 M H₂SO₄ at room temperature. No high-density Fe regions were observed in this case.

Catalyst	Atomic percentage			
	Metal	Fresh	Acid exposure	EoL
Pt/Fe-N-C _{Aero}	Fe	0.21±0.04	0.07±0.03	0.07±0.05
	Pt	0.21±0.19	0.10±0.06	0.16±0.12

The following text describing the Fe K-edge EXAFS results was added in the main file in the section describing the physicochemical characterization of Pt/Fe–N–C_{Aero}:

The fitting of the Fe K-edge EXAFS spectrum of Pt/Fe–N–C_{Aero} revealed the absence of Fe-Pt interaction, and Fe-N and Fe-O coordination numbers and bond distances similar to those in Fe–N–C_{Aero} (**Supplementary Fig. 3b** and **Supplementary Table 3**) independently confirming the unmodified FeN₄ site structure in Fe–N–C_{Aero} and Pt/Fe–N–C_{Aero} and absence of PtFe alloys and other Pt-Fe bonds

2. During the MEA test, Pt has a loading of 1% of the total 4.1 mg/cm², this leads to about 0.041 mg_{Pt}cm⁻² at the cathode side. The power density shown in the manuscript could be easily obtained for a MEA with only Pt/C nanoparticle catalysts at this loading. There’s no experimental evidence with the manuscript to clarify whether the power performance obtained comes from the Pt nanoparticles or the Pt-FeN₄ sites. The manuscript compares Pt/Fe-N-C and Pt/N-C, showing that Pt/N-C has some much larger Pt nanoparticle and much lower power performance. So, does the higher activity of Pt/Fe-N-C comes from the interaction between FeN₄ sites and Pt nanoparticles, or just due to the improved dispersion and particle size distribution of Pt nanoparticles because of the FeN₄ sites?

Our answer:

To answer this very justified question, we would like to first stress again that the main point of the present paper is to demonstrate that the addition of low amounts of Pt nanoparticles really stabilizes FeN₄ sites during ORR in acid medium (as the manuscript’s title says “Evidence for the stabilization of FeN₄ sites by a low amount of Pt particles during oxygen reduction in acid medium”), and we agree with the reviewer that the addition of 1 wt. % Pt to FeNC materials still leads to non-negligible Pt

amount in the cathode and is not the final technologically desirable solution to stabilize FeN₄ sites. Yet, we hope the paper will be useful to move the field forward, since by demonstrating that Pt nanoparticles truly stabilize FeN₄ sites, it provides information that FeN₄ sites can be stabilized through post-synthesis modification of the FeNC surface, and hopefully Pt nanoparticles can in the future be replaced by a non-precious or less-precious metal (or metal-oxide, metal-nitride, metal-carbide).

Having said that, we disagree with the reviewer that the Pt/N-C_{Aero} comparative material has much larger Pt particle size than Pt particles in the Pt/Fe-N-C_{Aero} material: as shown in supplementary figure 2a and 2b of the original submission, the Pt/N-C_{Aero} has a mean size of Pt particles of 1.69 nm while the Pt/Fe-N-C_{Aero} material has a mean size of Pt particles of 1.65 nm. The difference is thus only 0.04 nm, or 0.4 Å. This can be considered within the error range, and even if it was a true difference in Pt particle size, this very small difference could certainly not explain the dramatic difference in ORR activity observed between Pt/Fe-N-C_{Aero} and Pt/N-C_{Aero} (both observed in RDE, Figure 2c of the original submission, and in PEMFC, supplementary figure 13 of original submission). For example, in RDE, the mass activity at 0.8 V of Pt/N-C_{Aero} is about 1.1 A/g_{powder} while for Pt/Fe-N-C_{Aero} it is about 4.0 A/g_{powder}. An even higher relative difference in activity is seen in PEMFC, looking at the current densities at 0.8 V cell voltage where the cell performance is mainly dictated by ORR kinetics (Pt/N-C_{Aero}: 12.3 mA·cm⁻² at BoT, 17.3 mA·cm⁻² at EoT, and Pt/Fe-N-C_{Aero}: 99.1 mA·cm⁻² at BoT, 72.5 mA·cm⁻² at EoT). As a result, we can assign in a large proportion the activity difference between Pt/Fe-N-C_{Aero} and Pt/N-C_{Aero} to the activity imparted by FeN₄ sites, present in Pt/Fe-N-C_{Aero} but absent in Pt/N-C_{Aero}.

In summary, the presence of Pt in Pt/Fe-N-C_{Aero} contributes in two ways to the overall activity : 1) the Pt particles contribute themselves to a small fraction of the activity (as represented by the Pt/N-C_{Aero} activity) , and 2) the Pt particles modify the electronic density at FeN₄ sites, which possibly improves their turnover frequency and, as demonstrated by RRDE data (figure 4a and supplementary figure 10 of original submission), improves their four-electron selectivity, which de facto will lead to a slightly higher 'current' generated by FeN₄ sites. With a drop in % H₂O₂ from 30 % to almost 0 % from Fe-N-C_{Aero} to Pt/Fe-N-C_{Aero} (for the largest peroxide % seen, at ca 0.65 V), the expected increase in ORR current density is +18 % (30% peroxide means that, for 100 O₂ molecules, 30 are converted to H₂O₂ and 70 are converted to H₂O, corresponding to a total of 340 electrons, while for 0% peroxide, the 100 O₂ molecules then correspond to a total of 400 electrons. The increase from 340 to 400 electrons leads to the 18 % increase in current that is mentioned above).

Last, to further highlight to readers that the addition of 1 wt. % Pt to Fe-N-C materials is not the ultimate goal and that similar PEMFC performance can simply be obtained with ultrathin cathode layers prepared from commercial Pt/C catalyst (with high wt. % of Pt on C, typically in the range 40-60 %), we measured and reported such a curve (comprising 40 μg_{Pt} cm⁻²), as described below in our 'actions'. It does not impact any of our findings, as the focus of the study is the demonstration that the addition of low content of Pt nanoparticles stabilizes FeN₄ sites, but we believe it is useful as it shows clearly that it still is necessary for the field to work on the replacement of such stabilizing Pt nanoparticles, but with a paradigm shift that one does not need to look for particles that are ORR-active or even active for H₂O₂ reduction, but look for particles that can increase the four-electron selectivity of FeN₄ sites. The fundamental effect of the long-range interaction of metal (Pt in present case) NPs with FeN₄ sites is studied and discussed in the DFT part implemented during revision.

Our action:

The following text was added in the main text:

This holds promise for developing strategies to rationally stabilize FeN₄ sites, by post-synthesis modification of Fe-N-C catalysts. While 1 wt. % Pt on Fe-N-C may at first sight appear to be a low

content, with PEMFC cathodes of $4 \text{ mg}_{\text{powder}} \cdot \text{cm}^{-2}_{\text{geo}}$, it still leads to a significant $40 \text{ } \mu\text{g}_{\text{Pt}} \cdot \text{cm}^{-2}_{\text{geo}}$ loading at the cathode. The comparison of the PEMFC performance obtained with a Pt/Fe–N–C_{Aero} cathode ($4 \text{ mg}_{\text{powder}} \cdot \text{cm}^{-2}_{\text{geo}}$ of Pt/Fe–N–C_{Aero}, corresponding to $40 \text{ } \mu\text{g}_{\text{Pt}} \cdot \text{cm}^{-2}_{\text{geo}}$) or with a cathode prepared from a commercial 40 wt.% Pt/C (at $40 \text{ } \mu\text{g}_{\text{Pt}} \cdot \text{cm}^{-2}_{\text{geo}}$) shows that similar activities at 0.8 V are obtained (after the break-in period needed to activate commercial Pt/C, as is well-known), but at high current density, the Pt/Fe–N–C_{Aero} cathode surpasses the ultrathin Pt/C layer (**Supplementary Fig. 21** and **Supplementary Note 5**). Nevertheless, the ultimate goal is still the stabilization of Fe–N–C catalysts by non-PGM sites, which can synergistically act with FeN₄ sites in the same way as is reported in the present work.

A supplementary figure 21 was added, as well as a supplementary Note 5 accompanying that figure, as shown below:

Supplementary Fig. 4 | BoT polarization curves for Fe–N–C_{Aero}, Pt/Fe–N–C_{Aero}, Pt/N–C_{Aero}, Pt/C, as well as EoT curves for Pt/N–C_{Aero} and Pt/C. The total cathode catalyst loading was $4 \text{ mg}_{\text{powder}} \text{ cm}^{-2}_{\text{geo}}$ for Fe–N–C_{Aero}, Pt/Fe–N–C_{Aero} and Pt/N–C_{Aero} (resulting in *ca.* $40 \text{ } \mu\text{g}_{\text{Pt}} \text{ cm}^{-2}_{\text{geo}}$ at the cathode for Pt/Fe–N–C_{Aero} and Pt/N–C_{Aero}) while the total cathode catalyst loading for the commercial Pt/C (40 wt. % Pt on Vulcan XC72) was adjusted to result in $40 \text{ } \mu\text{g}_{\text{Pt}} \text{ cm}^{-2}_{\text{geo}}$ at the cathode. The cathode ink deposition method and MEA preparation were otherwise performed identically. See **Supplementary Note 5** for a discussion of the figure results.

Supplementary Note 5 | Discussion of the PEMFC results shown in **Supplementary Fig. 21**.

Additional PEMFC measurements were carried out with a commercial Pt/C catalyst (40 wt. % Pt on Vulcan XC72) and preparing ultrathin cathode layers to result in a Pt loading of only $40 \text{ } \mu\text{g}_{\text{Pt}} \cdot \text{cm}^{-2}_{\text{geo}}$ at the cathode, as a comparison to the performance obtained with cathodes based on Pt/Fe–N–C_{Aero} and Pt/N–C_{Aero}, with a total cathode catalyst loading of $4 \text{ mg}_{\text{powder}} \cdot \text{cm}^{-2}_{\text{geo}}$, that also comprise *ca.* $40 \text{ } \mu\text{g}_{\text{Pt}} \cdot \text{cm}^{-2}_{\text{geo}}$ (1 wt. % Pt on Fe–N–C_{Aero} and N–C_{Aero} in the Pt/Fe–N–C_{Aero} and Pt/N–C_{Aero} materials). The BoT performance for Pt/C is very low, with an activity at 0.8 V as low as that of Pt/N–C_{Aero}. However, after the potential hold at 0.5 V, the performance increases dramatically and the Pt/C cathode reaches an activity at 0.8 V comparable to those of Fe–N–C_{Aero} and Pt/Fe–N–C_{Aero} at BoT. It is well known that state-of-art Pt/C cathodes require a break-in process in PEMFC during which the cell performance increases. In the present case, the current density of the Pt/C cathode increased during the first 10 h of the voltage hold at 0.5 V, then stabilized. Such a behavior is the fingerprint of active platinum cathodes. In contrast, the activity at 0.8 V and cell performance of the Pt/N–C_{Aero} cathode only

increased marginally during the same potential hold, highlighting the low ORR activity of Pt NPs deposited at the low content of 1 wt. % on the N-C_{Aero} support. We stress that the same activity at 0.8 V of the Pt/C cathode at EoT and of the Pt/Fe-N-C_{Aero} at BoT cannot be interpreted as a Pt-driven activity for the Pt/Fe-N-C_{Aero}: First, the same activity at 0.8 V is also observed for the Pt-free Fe-N-C_{Aero} cathode at BoT, inferring that the activity of Pt/Fe-N-C_{Aero} should have been significantly higher, if the Pt in Pt/Fe-N-C_{Aero} was highly active. Second, the Pt/Fe-N-C_{Aero} does not behave as the Pt/C material in the sense that it overall loses a small amount of activity and performance after the potential hold (see **Figure 5a**). The limited improvement in current density from *ca.* 900 to *ca.* 950 mA cm⁻² observed during the first 2 h of the potential hold of Pt/Fe-N-C_{Aero} (see **Figure 5b**) may however be interpreted as the activity break-in of the Pt NPs in Pt/Fe-N-C_{Aero}. If this is true, it shows that the contribution of Pt to the overall activity and performance of Pt/Fe-N-C_{Aero} is not zero, but is minor, in line with all the other electrochemical results of the present study. Last, the lower performance at high current density of the ultrathin Pt/C cathode compared to thick Fe-N-C_{Aero} and Pt/Fe-N-C_{Aero} layers may be assigned to water flooding. While the thick Fe-N-C_{Aero} and Pt/Fe-N-C_{Aero} layers are not optimum for O₂ and proton transport, the thin Pt/C layer results in a low absolute volume of pores, and the high amount of water produced by the ORR at high current density can easily flood these pores. The carbon loading (driving the cathode active layer thickness) is only 60 μg_C·cm⁻²_{geo} for the Pt/C cathode, but *ca.* 4 mg_C·cm⁻²_{geo} (neglecting the relatively low Fe and N contents in Fe-N-C) in the Fe-N-C_{Aero} and Pt/Fe-N-C_{Aero} layers. Thus, one can estimate that the Pt/C layer is *ca.* 66 times thinner than the Fe-N-C_{Aero} and Pt/Fe-N-C_{Aero} layers.

The preparation of the Pt/C cathode layer from the commercial Pt/C catalyst (40 wt. % Pt on Vulcan XC72) and with the low loading of 40 μg_{Pt}·cm⁻² has been added in the Methods section.

3. During the AST under Ar, the including of Pt results in a worse stability for test 1, 2 and 3. The authors might add some further discussions to clarify this result.

Our answer:

The reviewer is correct that upon AST-1, AST-2 and AST-3 (Argon-saturated acid electrolyte), the Pt/Fe-N-C_{Aero} catalyst experiences slightly more activity loss than the Fe-N-C_{Aero} catalyst. This trend seems real, and is opposite to the trend seen in O₂-saturated acid electrolyte. This apparently counter-intuitive result may be assigned to platinum-catalyzed carbon corrosion in such conditions. Since no ORR occurs in inert gas, there might be no stabilization benefit of adding platinum to Fe-N-C and instead, the known effect of platinum in catalyzing carbon corrosion (Electrochimica Acta, Volume 56, 2011, Pages 7541-7549) might have led to a small disadvantage in terms of stability of Pt/Fe-N-C_{Aero} relative to Fe-N-C_{Aero}. For AST-4, combining the high upper potential limit (UPL) of 1 V and high temperature of 80 °C, the platinum surface likely was in an oxidized state at the UPL, and oxidized Pt surface was reported to be inactive for catalyzing carbon corrosion (Electrochimica Acta, Volume 56, 2011, Pages 7541-7549).

Our action: we added in the main text the following paragraph:

Surprisingly, Pt/Fe-N-C_{Aero} experiences slightly more activity loss than Fe-N-C_{Aero} upon ASTs 1 to 3, which may be assigned to platinum-catalyzed carbon corrosion in such mild stressing conditions. Enhanced carbon corrosion can lead to loss of FeN₄ sites, or simply lead to the formation of oxygen functional groups on the carbon surface and ensuing decreased TOF of FeN₄ sites.¹³ For AST-4, the high UPL value and high temperature likely resulted in Pt surface oxidation at UPL. This is known to deactivate the ability of platinum to catalyze carbon corrosion,⁴⁴ and can explain the reversed trend of relative stability of Pt/Fe-N-C_{Aero} and Fe-N-C_{Aero} between AST-4 and AST 1-3.

Reviewer #3

In this manuscript, the authors present a novel approach to enhancing Fe-N-C catalyst durability by incorporating Pt nanoparticles (NPs) via a 'soft polyol method', minimizing FeN₄ active structure modifications. Unlike conventional hybrid catalysts where Pt NPs contribute to ORR activity and/or act as H₂O₂ scavengers, the authors propose that Pt NPs in Fe-N-C catalysts suppress H₂O₂ generation and reduce Fe dissolution rates, improving catalyst stability. Furthermore, the study elucidates the stabilization of specific high-activity but unstable active sites by Pt NPs. The study also reveals the stabilization of high-activity but unstable active sites by Pt NPs, offering a new perspective on Fe-N-C catalyst durability enhancement. While some aspects require further refinement, this research has the potential to make a valuable contribution suitable for publication in Nature Communications. My major criticisms to this manuscript are given in below:

1. In this study, Pt 4f XPS spectral analysis revealed electronic interactions between Pt NPs and FeN₄ sites. However, the absence of metal-support interaction effects in N-CAero raises questions. Typically, the introduction of heteroatoms such as nitrogen into carbon supports is known to alter the support's electron density, subsequently influencing the electronic state of the deposited platinum. This phenomenon leads to the expectation that Pt/N-CAero should exhibit different characteristics from Pt/C. What, then, accounts for the similarity in electronic states between Pt/N-CAero and Pt/C, while only Pt/Fe-N-CAero demonstrates a change in the electronic state of Pt?

Our answer:

We are thankful to the reviewer for having raised this question. The XPS measurements were repeated on Pt/Fe-N-C_{Aero}, Pt/N-C_{Aero} and a commercial Pt/C, and the binding energy shift of Pt 4f is now observed for both Pt/Fe-N-C_{Aero}, Pt/N-C_{Aero}, as expected from previous literature.

Our action:

The text describing the XPS results was modified to:

To explore electronic interactions between Pt NPs and FeN₄ sites, **Fig. 1d** compares the Pt_{4f} fitting for commercial Pt/C, Pt/N-C_{Aero} and Pt/Fe-N-C_{Aero}. **A considerable shift of ca. 0.8 eV towards a higher binding energy was found for Pt/N-C_{Aero} and Pt/Fe-N-C_{Aero} compared to Pt/C. This observation aligns with the known change in the electronic density of carbon matrices when doping them with nitrogen, which results in a stronger metal-support interaction with the Pt NPs. This effect explains the shift in the binding energy values obtained and has been observed regardless of the presence – or not – of FeN₄ sites in the N-doped carbon matrix.**^{30,36}

The new XPS data is shown in Fig. 1d, and fitting results in Supp. Fig. 7, Supp. Table 4-6.

2. The XPS survey spectra of Fe-N-C_{Aero} and its platinum-deposited counterpart in Supplementary Fig. 4a do not provide information on iron content. It would be beneficial to include high-resolution XPS analysis results for the Fe 2p region.

Our answer:

We believe that ICP-MS can provide a more accurate quantification of Fe present in the materials than XPS, as Fe 2p XPS spectra are typically noisy, even after long acquisition time. The Fe 2p XPS spectrum may additionally identify different Fe species in different oxidation states (expectedly a mix of Fe²⁺ and Fe³⁺), but this would not allow us differentiating iron oxides from FeN₄ sites with the central Fe in

different oxidation states. For this, ^{57}Fe Mössbauer spectrum of Fe-N-C_{Aero} is more insightful (see Supplementary Figure 1a). The ICP-MS results were reported in the original submission (quoted: “Inductively coupled plasma mass spectrometry (ICP-MS) measured Pt contents of 1.02 and 0.78 wt. % in Pt/N-C_{Aero} and Pt/Fe-N-C_{Aero}, respectively, with Fe content in Pt/Fe-N-C_{Aero} at 0.68 wt. %”, and for Fe-N-C_{Aero}: “The catalyst features an Fe content of 1.25 wt. %”)

3. The authors demonstrated electronic interactions between FeN₄ and Pt NPs through the observed up-shift in peak binding energy of the XPS 4f spectra, and changes in the electronic state of Fe as evidenced by Fe K-edge XANES analysis. However, they argue against the formation of PtFe alloys or Fe particle agglomeration based on the nearly identical XES spectra of Fe-N-CAero and Pt/Fe-N-CAero. Given that XES analysis is also sensitive to the oxidation state of elements, it is puzzling that no differences in Fe species were detected, unlike in the XANES results.

Our answer:

We thank the Referee for this comment. We note that the XANES measurements were conducted using the Fe K α line, whereas the XES measurements were performed using the Fe K β line.

In general, Fe K α XANES probes 1s \rightarrow 4p transitions, making it highly sensitive to unoccupied electronic states. This allows it to detect subtle changes in oxidation state, coordination environment, and bonding geometry. Any shifts or variations in the XANES spectra may indicate modifications in these parameters.

In contrast, Fe K β XES primarily captures transitions from 3p \rightarrow 1s, reflecting the occupied electronic states. It is particularly sensitive to the local spin state and the overall electronic configuration. If there are no significant changes in the chemical state or bonding environment, **the XES spectra may remain largely unchanged, even if the XANES spectra exhibit variations.**

Indeed, K β (3p \rightarrow 1s) X-ray emission spectroscopy (XES) has been proven to be an excellent tool for the study of the electronic structure of metal centers, due to 3p-3d exchange in the final state, and K β mainlines have been shown to reflect and correlate with both oxidation and spin states. Specifically, the K β _{1,3} feature, which marks the peak of the K β mainline, typically shifts to higher energy as the oxidation state increases. Meanwhile, the intensity of the lower-energy K β ' feature is linked to the number of unpaired electrons, making it useful for determining spin states.

While this is generally true in idealized systematic studies, covalency has a significant impact on the shape and energy of K β mainlines. As metal-ligand covalency increases, the metal 3d orbital character becomes more delocalized onto the ligands, leading to a reduction in metal 3p-3d exchange integrals, which influence the final state splitting in K β spectra. Consequently, the expected shift of the K β _{1,3} feature to higher energy with increasing oxidation state can be offset by stronger metal-ligand covalency, complicating the interpretation of oxidation state trends (see <https://doi.org/10.1021/ja504182n> , <https://doi.org/10.1021/acs.inorgchem.0c01620>). The higher sensitivity of Fe K β XANES relative to XES K β was also clear in a published study on three other Fe-N-C materials co-authored by some of the author's present study (see **figure R4** below).

In summary, XES on Fe-N-C materials can be used to track changes in the spin state of FeN_x sites, while changes in oxidation states are better resolved with XANES.

Figure 1. Fe $K\beta$ HERFD XANES (a) and XES (b) spectra recorded on Fe–N–C catalyst powders. The pre-edge region is shown in the inset of panel a.

Figure R4. Figure reproduced from ACS Appl. Energy Mater. 2023, 6, 611-616 (https://pubs.acs.org/doi/epdf/10.1021/acsaem.2c03736?ref=article_openPDF)

Our action:

Considering the referee’s comment to avoid confusion to the reader, we modified the following sentence in original manuscript:

Last, the identical Fe $K\beta$ XES spectra for Fe–N–C_{Aero} and Pt/Fe–N–C_{Aero} (**Supplementary Fig. 5**) confirmed that no detectable amount of FeN₄ sites was transformed to PtFe alloy or other Fe crystalline species, consistent with the soft Pt deposition method employed in the present study.

To

The identical Fe $K\beta$ XES spectra for Fe–N–C_{Aero} and Pt/Fe–N–C_{Aero} (**Supplementary Fig. 8**) confirmed that **the average spin state of iron in FeN₄ sites was unmodified in their resting state.**³⁸

The absence of PtFe alloy or other Fe crystalline species is better demonstrated with the added Fe K-edge EXAFS analysis of Fe–N–C_{Aero} and Pt/Fe–N–C_{Aero} (see our answer and actions related to the last question of this reviewer)

And we included the following statement in the figure caption of Supplementary figure 5:

We note that the XANES measurements were conducted using the Fe $K\alpha$ line, while the XES measurements were performed using the Fe $K\beta$ line. Although XES can generally be sensitive to oxidation state, the interpretation of oxidation state often relies on $K\alpha$ XANES, as $K\beta$ XES is primarily influenced by the spin state.

4. The observation of increased ORR activity after AST in platinum catalysts such as Pt/N-CAero, Pt/Fe-N-CPaj, and Pt/Fe0.5 warrants further investigation. A more comprehensive analysis of the changes in Pt NPs' contribution to ORR activity due to activation during the AST process is necessary. It would be valuable to examine the differences in chemical characteristics of Pt post-AST, as well as to assess performance trends following the blocking of Pt NPs in post-AST catalysts.

Our answer:

In the manuscript, the wording AST was specifically referring to the accelerated stress tests performed in RDE setup, for which the ORR activity of all materials decreased after AST, as shown in Figure 3 of the main manuscript, all normalized activity (normalized to their value before AST) values being less than 1. We understand that with AST, the reviewer refers in this question to the PEMFC durability tests.

We agree with the keen observation of the reviewer, that the ORR activity in PEMFC increased after durability test for the Pt/N-C_{Aero}, Pt/Fe-N-C_{Paj}, and Pt/Fe_{0.5} materials, as seen from the current density at 0.8 V at BoT and EoT (beginning of test and end of test, respectively), which we also report in the table below for ease of discussion.

Cathode	Current density at 0.8 V in PEMFC (mA cm ⁻²) at BoT	Current density at 0.8 V in PEMFC (mA cm ⁻²) at EoT
Pt/N-C _{Aero}	12.3	17.3
Pt/Fe-N-C _{Aero}	99.1	72.5
Pt/Fe-N-C _{Paj}	25.0	35.0
Pt/Fe _{0.5}	22.0	28.7

The increase in activity of + 5 mA cm⁻² at 0.8 V observed for Pt/N-C_{Aero} from BoT to EoT is most likely assigned to a small activation of Pt particles (either due to the Pt particle size increasing relative to the BoT value – the low average initial Pt particle size being off the optimum for ORR activity of Pt nanoparticles, for which the highest mass activity is typically reached in the range of 2-3 nm Pt particle size (<https://doi.org/10.1007/BF00348773>) – or reorganization of the Nafion ionomer / catalyst interface).

A similar activation might occur for Pt particles deposited on Fe-N-C substrates, unless the interaction of Pt particles with FeN₄ sites mitigates/stops the growth of Pt particles (as suggested in studies of high loading of Pt on Fe-N-C, <https://doi.org/10.1039/D1EE01675J>), and this might suppress the activation of small Pt nanoparticles during break-in in PEMFC.

In this case, the increase in ORR activity during the durability test (as observed for Pt/Fe-N-C_{Paj} and Pt/Fe_{0.5}) may be assigned to activation due to improved ionomer/catalyst interface during the break-in, or increased number of electrochemically accessible FeN₄ sites. Some Fe-N-C materials (with low or moderate activity) have shown an initial increase in activity in PEMFC (e.g. the catalyst labelled NC Por_0.8-1150Ar in 10.1016/j.electacta.2015.01.201).

Interestingly, there seems to be a correlation between the change in activity at 0.8 V of Pt/Fe-N-C hybrid materials during the PEMFC durability test and the ratio of D2/ (D1+D2) signals as derived from Mossbauer spectroscopy: the higher the D2/ (D1+D2) signal in the parent Fe-N-C material, the higher is the activity increase during PEMFC durability test of Pt/Fe-N-C (**Figure R5** below). The latter may be assigned to Pt NP activation, which however requires a high amount of D2 (low amount of D1) initially, in order to not be offset by the activity decrease related to the loss of some of the D1 sites that cannot be stabilized by Pt. Furthermore, the Fe-N-C_{Aero} not only has the lowest D2/ (D1+D2) ratio but its synthesis moreover involves a second pyrolysis in dilute NH₃, which is known to increase initial activity but also leads to less stable materials in PEMFC, due to high surface basicity.

Figure R5. Change in current density at 0.8 V in PEMFC from BoT to EoT as a function of the D2 / (D1 + D2) ratio derived from ⁵⁷Fe Mössbauer spectroscopy of the parent Fe-N-C materials. The change in current density is also shown for Pt/N-C_{Aero} as a comparison to Pt/Fe-N-C_{Aero}.

Our action:

We added in the main text the following text:

Moreover, comparing the BoT and EoT current density at 0.8 V, the slight improvement in ORR activity seen for Pt/Fe_{0.5}, Pt/Fe-N-C_{Paj} and Pt/N-C_{Aero} during the test can be assigned to fully stabilized FeN₄ sites combined with an activation of Pt NPs, as observed on Pt/N-C_{Aero} (**Supplementary Figs. 21-22**). While the relative contribution of the activated Pt to the overall EoT activity may be non-negligible for the less active Pt/Fe_{0.5} and Pt/Fe-N-C_{Paj} cathodes, it can contribute only to a minor extent to the EoT activity of Pt/Fe-N-C_{Aero}, the EoT activity of the latter being *ca.* 4 times higher than the EoT activity of Pt/N-C_{Aero} (**Supplementary Fig. 22**). Interestingly, there seems to be a correlation between the change in activity at 0.8 V of Pt/Fe-N-C hybrids during the PEMFC durability test and the ratio of D2/(D1+D2) signals as derived from Mössbauer spectroscopy. The higher the D2/(D1+D2) signal in the parent Fe-N-C material, the higher is the activity increase during PEMFC durability test (**Supplementary Fig. 22**). The latter may be assigned to Pt NP activation, which however requires a high amount of D2 (low amount of D1) initially, to not be offset by the activity decrease related to the loss of some of the D1 sites.

And Figure R5 above was integrated in the supplementary information (Supp. Fig. 22)

5. What is the rationale behind employing two distinct types of FeN₄ catalysts (Fe-N-C_{Aero} and Fe_{0.5}) in elucidating the interactions between FeN₄ sites and Pt NPs?

Our answer:

The present study focuses on Fe-N-C_{Aero} and the idea behind expanding the study to two other Fe-N-C catalysts (Fe_{0.5} and the commercial Fe-N-C_{Paj}, the latter also comprising FeN₄ sites) was to demonstrate that similar stabilization is observed for different Fe-N-C materials prepared by different approaches. It is well known that most pyrolyzed Fe-N-C catalysts comprise two sub-sets of FeN₄ sites, which are identified with Mössbauer spectroscopy as two different quadrupole doublets, typically labeled D1 and D2 with similar isomer shifts but distinct quadrupole splittings (see Supplementary Figure 1). As the stability of the D1 and D2 sites is different (D1 being less stable,

<https://www.nature.com/articles/s41929-020-00545-2>), and the three Fe-N-C catalysts show different relative contents of the D1 and D2 spectral components (61 % D1 and 20 % D2 for Fe-N-C_{Aero}, 52 % D1 and 48 % D2 for Fe-N-C_{Paj} and 64 % D1 and 36 % D2 for Fe_{0.5} – see Supplementary Table 1 in original version of manuscript), these 3 materials offer an interesting set of materials to verify if the Pt deposition method stabilizes them in a similar way or not.

Another important aspect is the porosity and pore size distribution of these three Fe-N-C materials. Fe-N-C_{Aero} is derived from an aerogel and involving a pyrolysis in dilute NH₃ and comprises a high amount of micropores and a low amount of mesopores. Fe_{0.5} is derived from ZIF-8 via pyrolysis in inert gas and is mainly microporous, but less than Fe-N-C_{Aero}, and contains a limited amount of mesopores. The commercial Fe-N-C_{Paj} catalyst is known to be derived from silica templating followed by HF removal of the silica (a methodology developed at Atanassov's laboratory at Univ. New Mexico), which results in very high mesoporosity, while its micropore volume is situated in-between those of Fe-N-C_{Aero} and Fe_{0.5}. The different porosities may have led to different environments or location of the 1 wt. % Pt nanoparticles and thus different effects of Pt deposition and Pt reactivity.

In summary, the aim with the use of the three Fe-N-C catalyst was to confirm if the phenomenon of Pt stabilization works for all of them, regardless of the synthesis approach and the resulting vastly different porous textures.

Our action:

We have added in the main text the following sentence, when initially describing the Pajarito commercial catalyst

... Fe-N-C_{Paj} is highly mesoporous while possessing also a significant amount of micropores (**Supplementary Fig. 1 and Supplementary Table 1**).

We have added in the main text the following sentence, when initially describing the Fe_{0.5} catalyst

...Its N₂ sorption isotherm has a similar shape to that of Fe-N-C_{Aero}, but with lower micropore volume and slightly higher mesopore volume (**Supplementary Fig. 1 and Supplementary Table 1**).

And finally, we now highlight in the revised conclusion that the different Fe-N-C catalysts have different textural properties.

This study offers novel insights on the synergy taking place between Pt NPs and FeN₄ sites and how the durability of various Fe-N-C catalysts **having different textural properties** is improved.

A supplementary figure was added, showing the isotherms, as reproduced below.

Supplementary Fig. 1 | Nitrogen sorption isotherms of Fe-N-C_{Aero}, Fe-N-C_{Paj}, and Fe_{0.5}. The BET specific surface area as well as pore volumes derived from the analysis of the adsorption branch are reported in **Supplementary Table 1**.

A supplementary Table was added to report the BET area, micro-, meso-pore and total pore volumes as shown below.

Supplementary Table 1 | Porous structure of the Fe-N-C_{Aero}, Fe-N-C_{Paj} (batch PMF-D14401, Pajarito Powder) and Fe_{0.5} powders. Specific surface area (BET), mesopore volume derived from BJH analysis of the adsorption isotherms, total pore volume and micropore volume (the latter was derived from the total pore volume minus the pore volume assessed by BJH analysis for all pores of size > 2 nm):

	BET area (m ² g ⁻¹)	Microporous volume (cm ³ g ⁻¹)	Mesoporous volume (cm ³ g ⁻¹)	Total pore volume (cm ³ g ⁻¹)
Fe-N-C _{Aero}	1191	0.46	0.02	0.51
Fe-N-C _{Paj}	755	0.25	0.85	1.28
Fe _{0.5}	455	0.19	0.04	0.25

6. In the electrode preparation for Mössbauer spectra measurements presented in Fig. 6, the experimental design ensured equal loading of Fe-N-C, enabling quantitative comparison of the absolute amount of each Fe species. In this context, it is crucial to ascertain whether the electrode

fabrication for Pt-deposited samples was adjusted to compensate for the platinum content. Specifically, was the amount of pure Fe-N-C catalyst (i.e., the Fe content) corrected by excluding the weight of platinum when preparing these electrodes?

Our answer:

While we agree that such an adjustment would have been important if high content of Pt had been deposited on Fe-N-C materials, with only 1 wt. % Pt (targeted Pt loading and also close to the real Pt content deposited, as measured by ICP-MS), such an adjustment is unnecessary. The electrode fabrication was not modified, and we always target to deposit 4 mg cm^{-2} , which represents the total mass of catalyst, including the mass of Pt in case of Pt/Fe-N-C materials. Assuming exactly 1 wt. % Fe in the starting Fe-N-C material and assuming 1 wt. % Pt was deposited on a 1 % Fe-N-C material, the table below reports the theoretically expected loadings of Fe and Pt, In the case of a Fe-N-C cathode and a Pt/Fe-N-C cathode, both having exactly a total catalyst loading of 4 mg cm^{-2} .

Catalyst	Fe content in the catalyst	Fe loading for $4 \text{ mg catalyst cm}^{-2} / \mu\text{g cm}^{-2}$	Pt content in the catalyst	Pt loading for $4 \text{ mg catalyst cm}^{-2} / \mu\text{g cm}^{-2}$
Fe-N-C	1 wt. %	40	0	0
Pt/Fe-N-C	0.99 wt. %	39.6	1 wt. %	40

The relative change expected in Fe loading from a Fe-N-C cathode to a Pt/Fe-N-C cathode derived from the same Fe-N-C material is thus, in theory, only a 1 % relative decrease (i.e. 0.01 wt. % Fe content decrease). Adjusting for this would be really challenging and would necessitate measuring the cathode loadings with a precision below 1 %, which is challenging since the cathode also contains Nafion ionomer, subject to mass gain depending on the relative humidity of the environment.

Our action:

In the Methods section, we now precise that the same ink formulation and MEA preparation method was used for a given Fe-N-C and its Pt/Fe-N-C derivative, as highlighted below in red.

The MEAs for PEMFC measurements with Fe-N-C_{Aero} or Pt/Fe-N-C_{Aero} were prepared using a procedure...

[...]

For the two other Fe-N-C catalysts and their platinized versions (Fe-N-C_{Paj}, Pt/Fe-N-C_{Paj}, Fe_{0.5} and Pt/Fe_{0.5}), the ink was composed of...

7. The study appears to lack comprehensive verification of the local structure in Fe-N-C catalysts. It would be beneficial to address whether EXAFS curve fitting analysis was conducted to confirm the presence of FeN₄ active structures in the Fe-N-C catalysts. Such analysis could provide crucial insights into the atomic-level arrangement and coordination environment of iron atoms, thereby substantiating the assumed FeN₄ configuration.

Our answer:

The ⁵⁷Fe Mössbauer spectroscopy results of the three Fe-N-C catalysts identify the typical doublets D1 and D2 with isomer shift and quadrupole splitting values typical for FeN₄ sites in different spin states and oxidation states (**Supplementary Fig. 2** and **Supplementary Table 2** of revised manuscript). Nevertheless, we agree that EXAFS analysis can provide independent confirmation for the FeN_x local coordination and the average Fe-N coordination number of $x = 4$. This analysis was already performed multiple times by us in previous studies on the ZIF-8 derived Fe-N-C (Fe_{0.5}) (for example <https://www.nature.com/articles/nmat4367>), but not on the two other Fe-N-C materials used in the present study. During the revision of the submitted manuscript, we could obtain some in-house beamtime at Synchrotron SOLEIL and carried out the EXAFS measurement and analysis of the Fe-N-C_{Aero} and Pt/ Fe-N-C_{Aero} materials. The choice was made to measure the EXAFS of Fe-N-C_{Aero} and Pt/ Fe-N-C_{Aero} instead of Fe-N-C_{Aero} and Fe-N-C_{Paj} so as to identify with EXAFS if the addition of Pt on Fe-N-C_{Aero} modifies the Fe-N coordination number and/or bond distance.

Our action:

A supplementary figure and supplementary Table showing the EXAFS analyses of Fe-N-C_{Aero} and Pt/ Fe-N-C_{Aero} were added, and are reproduced below:

Supplementary Fig. 3 | Fe K-edge EXAFS analysis of **a)** Fe-N-C_{Aero}, **b)** Pt/Fe-N-C_{Aero}. Upper panel: Fe-N, Fe-O and Fe-C $\chi^{(2)}$ two-body signals included in the fit, the total signal (orange or blue line) superimposed to the experimental one (black line). Lower panel: the fit in the Fourier transformed space.

Supplementary Table 1 | Best-fit parameters obtained from the Fe *K*-edge EXAFS analysis of Fe–N–C_{Aero} and Pt/Fe–N–C_{Aero}. R is the bond distance, σ^2 is the Debye-Waller factor, and CN is the coordination number.

	Fe–N–C _{Aero}			Pt/Fe–N–C _{Aero}		
	R(Å)	$\sigma^2(10^{-3} \text{ \AA}^2)$	CN	R(Å)	$\sigma^2(10^{-3} \text{ \AA}^2)$	CN
Fe-N	2.05 ± 0.02	7.9 ± 0.5	3.8 ± 0.3	2.05 ± 0.01	5.6 ± 0.7	3.5 ± 0.1
Fe-O	1.91 ± 0.03	5.6 ± 0.9	1.8 ± 0.2	1.88 ± 0.02	6.7 ± 1.0	2.0 ± 0.1
Fe-C	3.61 ± 0.04	30 ± 5	10 ± 1	3.46 ± 0.03	30 ± 5	8.6 ± 0.3

The following text describing the EXAFS results was added in the main file in the section describing the physicochemical characterization of Fe–N–C_{Aero}

The Fe-N bond distance and Fe-N average coordination number in Fe–N–C_{Aero} were assessed from its extended X-ray absorption fine structure (EXAFS) spectrum measured at the Fe *K*-edge (**Supplementary Fig. 3a** and **Supplementary Table 3**). The experimental EXAFS spectrum was satisfactorily fitted considering a number of N and O atoms in the first coordination sphere and C atoms in the second coordination sphere while no Fe-Pt interaction was necessary to interpret the spectrum. An average Fe-N coordination number of 3.8 ± 0.3 was found with a bond distance of $2.05 \pm 0.02 \text{ \AA}$ (**Supplementary Table 3**), confirming the FeN₄ site structure. The average coordination number for Fe-O was 1.8 ± 0.2 , suggesting that most of the FeN₄ sites in Fe–N–C_{Aero} are gas-phase accessible and adsorb either two O₂ molecules in end-on mode or one O₂ molecule in side-on mode, as previously proposed.³⁵

The following text describing the EXAFS results was added in the main file in the section describing the physicochemical characterization of Pt/Fe–N–C_{Aero}:

The fitting of the Fe *K*-edge EXAFS spectrum of Pt/Fe–N–C_{Aero} revealed the absence of Fe-Pt interaction, and Fe-N and Fe-O coordination numbers and bond distances similar to those in Fe–N–C_{Aero} (**Supplementary Fig. 3b** and **Supplementary Table 3**) independently confirming the unmodified FeN₄ site structure in Fe–N–C_{Aero} and Pt/Fe–N–C_{Aero} and absence of PtFe alloys and other Pt-Fe bonds.

In addition, the experimental methods for the EXAFS acquisition and analysis was added in the Methods section:

For EXAFS, Fe *K*-edge X-ray absorption spectra were collected at room temperature at SAMBA beamline (Synchrotron SOLEIL, France). The beamline is equipped with a Si 220 monochromator and two Pd-coated mirrors, also used to remove X-ray harmonics. The catalysts were pelletized as disks of 10 mm diameter using boron nitride as a binder. Spectra were recorded in transmission (Fe–N–C_{Aero}) and fluorescence (Pt/Fe–N–C_{Aero}) mode using ionization chambers and a 35-elements Ge detector, respectively. The EXAFS data analysis was performed with the GNXAS code, which interprets the $\chi(k)$ signal by decomposing it into a sum of *n*-body distribution functions, $\gamma(n)$, derived through multiple-scattering (MS) theory. A detailed explanation of the GNXAS theoretical approach can be found in Refs 62,63. The Fe–N–C_{Aero} and Pt/Fe–N–C_{Aero} coordination shells around Fe were modelled with Γ -like distribution functions, which depend on four parameters: the coordination number CN, the distance R, the Debye-Waller factor σ^2 , and the skewness β . Optimization was carried out also for E_0 (core ionization threshold energy) and S_0^2 .

Other changes:

- We modified the figure 1c, after realizing that the AD2/AG ratio reported in the original version were erroneous. The trends among the materials are however unchanged and so are the conclusions regarding carbon matrix structure as probed by Raman spectroscopy.
- We added Andrea Zitolo as co-author. He conducted the acquisition and fitting of the EXAFS spectra, that were requested and added during the revision
- We added Tzonka Mineva and Hazar Guesmi as co-authors. They conducted the DFT study, which is an entirely new addition relative to the original manuscript that was submitted.

Reviewer 1 attachment 1

Durability of Fe-N-C cathode catalysts in PEFC is a key topic to be addressed for further commercialization of this technology while replacing precious Pt. As mentioned by the authors of the submitted manuscript the two main factors for the rather rapid degradation of Fe-N-C are demetallation of FeN₄ sites and a reactive oxygen species (ROS) degradation path. The degradation mitigation approach presented by the authors in this manuscript is a promising approach. Compared to previously reported Pt/Fe-N-C catalysts the authors combine the rather mild condition Pt nanoparticle (Pt NP) deposition via the soft polyol method with a low Pt amount of approx. 1 wt. % resulting in a *supposedly* uniform dispersion of well-defined Pt NP.

While this method shows promising results for three different Fe-N-C species, my main concern is the evidence for the enhanced durability of the Pt/Fe-N-C catalyst. The authors present a theoretical approach by calculation of the Pt NP to FeN₄ sites ratios for different Pt NP sizes and FeN₄ site densities and suggest that the enhanced durability is due to a long-range stabilization effect and not atomic distance between Pt NP and FeN₄ sites.

To be published in Nature Communications the authors should either provide some basic theoretical calculations, which support that one Pt NP can stabilize many FeN₄ sites by long-range interaction or suggest a model, what kind of interaction is at work here and how this interaction can be established over a length of tens of atoms and simultaneously with different FeN₄ sites. Furthermore, more experimental evidence for homogeneous FeN₄ site distribution in the Fe-N-C and Pt NP distribution between those sites would be helpful. Alternatively, experimental data for different Pt wt. % could be presented supposedly resulting in a volcano plot for the durability/mass activity, which could prove that 1 wt. % Pt load is the ideal situation.

Regarding these two points I would recommend a revision before the manuscript can be published in Nature Communications.

The provided data support the conclusions of the authors but see my comment above.

The work meets the expected standard, and enough details are provided for the work to be reproduced.

Specific questions to the authors:

L56: What is the proposed mechanism for demetallation under inert gas since ROS are involved?

L112/113: What was the number of particles used for particle size evaluation and what is the error in particle size?

L161: What was the reason for using H₂SO₄ instead of HClO₄ for the wet-electrochemical experiments (RDE/RRDE)? HClO₄ is considered as having a similar acidity compared to the sulfonate groups in Nafion and therefore allows to better compare results obtained by RDE with those obtained by PEFC experiments.

L173: Usually, a voltage of 0.9 V vs. RHE is chosen for mass activity. Why 0.8 V?

L179: The authors should also calculate mass activity in A g⁻¹_{Pt} for better comparison.

L184: The reason to compare the performance to the mathematical sum of the individual performances of Fe-N-C_{aero} and Pt/N-C_{aero} is to show the synergetic effect? The authors should clarify this.

L272: The calculation of the theoretical diffusion-limited current density should be shown in supplemental notes

L297: The TEM images are slightly too dark.

L338: Is the Pt NP size comparable to Pt/Fe-N-C?

L357: Why the change in electrolyte for this specific experiment? The authors should give a reason

L474: Exact data about the amounts required to ensure reproducibility

L510: Electric circuit?

L522: What is the calculated Pt load in $\text{g cm}^{-2}_{\text{geo}}$

L579: JEOL JEM2100 series TEM?

Reviewer 1 attachment 2

“Text in quotation marks”: My comment to the original manuscript

Italics: My comment to the revisions addressing my comments to the original manuscript

Review:

“To be published in Nature Communications the authors should either provide some basic theoretical calculations, which support that one Pt NP can stabilize many FeN₄ sites by long-range interaction or suggest a model, what kind of interaction is at work here and how this interaction can be established over a length of tens of atoms and simultaneously with different FeN₄ sites.”

In their revised manuscript the authors provided DFT calculations based on the interaction of single atom Pt with periodic models of FeN₄C₁₀ and FeC₄C₁₂ periodic models to study the long-range interaction between Pt NPs and FeN₄ sites. I agree with the authors statement that while different from real Pt NP this approach is appropriate with respect to the Pt-Fe distances (7-10 Å). The DFT results presented by the authors answer my question.

“Furthermore, more experimental evidence for homogeneous FeN₄ site distribution in the Fe-N-C and Pt NP distribution between those sites would be helpful. Alternatively, experimental data for different Pt wt. % could be presented supposedly resulting in a volcano plot for the durability/mass activity, which could prove that 1 wt. % Pt load is the ideal situation.”

The authors have provided additional STEM-X-EDS images confirming a high dispersion of the Pt NPs and added experimental data for different Pt wt. % confirming that full stabilization is reached with 1 wt. % Pt. The additionally experimental data provide sufficient evidence to answer my question.

I would recommend publication of the manuscript in its revised version

The provided data support the conclusions of the authors

The work meets the expected standard, and enough details are provided for the work to be reproduced.

“Specific questions to the authors:”

My specific questions have been addressed sufficiently by the authors